**Air-water CO₂ evasion from U.S. East Coast estuaries**
Laruelle, Goulven Gildas[1*], Goossens, Nicolas[1], Arndt, Sandra[2], Cai, Wei-Jun[3] & Regnier, Pierre[1]
1 Department Geosciences, Environment and Society, Université Libre de Bruxelles, Brussels,
Belgium
2 School of Geographical Sciences, University of Bristol, Bristol, UK
3 School of Marine Science and Policy, University of Delaware, Newark, Delaware, USA
*corresponding author: goulven.gildas.laruelle@ulb.ac.be

**Abstract:**
This study presents the first regional-scale assessment of estuarine $CO_2$ evasion along the East coast
of the US (25 – 45 °N). The focus is on 42 tidal estuaries, which together drain a catchment of
697000 $km^2$ or 76 % of the total area within this latitudinal band. The approach is based on the
Carbon – Generic Estuarine Model (C-GEM) that allows simulating hydrodynamics, transport and
biogeochemistry for a wide range of estuarine systems using readily available geometric parameters
and global databases of seasonal climatic, hydraulic, and riverine biogeochemical information. Our
simulations, performed using conditions representative of the year 2000, suggest that, together, US
East coast estuaries emit 1.9 TgC $yr^{-1}$ in the form of $CO_2$, which correspond to about 40 % of the
carbon inputs from rivers, marshes and mangroves. Carbon removal within estuaries results from a
combination of physical (outgassing of supersaturated riverine waters) and biogeochemical
processes (net heterotrophy and nitrification). The $CO_2$ evasion and its underlying drivers show
important variations across individual systems, but reveal a clear latitudinal pattern characterized by
a decrease in the relative importance of physical over biogeochemical processes along a North-South
gradient. Finally, the results reveal that the ratio of estuarine surface area to the river discharge, S/Q
(which has a scale of per meter discharged water per year), could be used as a predictor of the
estuarine carbon processing in future regional and global scale assessments.

## 1 Introduction

Carbon fluxes along the land-ocean aquatic continuum are currently receiving increasing attention because of their recently recognized role in the global carbon cycle and anthropogenic $CO_2$ budget (Bauer et al., 2013; Regnier et al., 2013a; LeQuéré et al., 2014, 2015). Estuaries are important reactive conduits along this continuum, which links the terrestrial and marine global carbon cycles (Cai, 2011). Large amounts of terrestrial carbon transit through these systems, where they mix with carbon from autochthonous, as well as marine sources. During estuarine transit, heterotrophic processes degrade a fraction of the allochthonous and autochthonous organic carbon inputs, supporting a potentially significant, yet poorly quantified $CO_2$ evasion flux to the atmosphere. Recent estimates suggest that 0.15-0.25 PgC yr$^{-1}$ is emitted from estuarine systems worldwide (Borges and Abril, 2012; Cai, 2011; Laruelle et al., 2010; Regnier et al., 2013a; Laruelle et al., 2013, Bauer et al., 2013). Thus, in absolute terms the global estuarine $CO_2$ evasion corresponds to about 15% of the open ocean $CO_2$ uptake despite the much smaller total surface area.

Currently, estimates of regional and global estuarine $CO_2$ emissions are mainly derived on the basis of data-driven approaches that rely on the extrapolation of a small number of local measurements (Cai, 2011; Chen et al., 2013; Laruelle et al., 2013). These approaches fail to capture the spatial and temporal heterogeneity of the estuarine environment (Bauer et al., 2013) and are biased towards anthropogenically influenced estuarine systems located in industrialized countries (Regnier et al., 2013a). Even in the best surveyed regions of the world (e.g. Australia, Western Europe, North America or China) observations are merely available for a small number of estuarine systems. In addition, if available, data sets are generally of low spatial and temporal resolution. As a consequence, data-driven approaches can only provide first-order estimates of regional and global estuarine $CO_2$ emissions.

Integrated model-data approaches can help here, as models provide the means to extrapolate over temporal and spatial scales and allow disentangling the complex and very dynamic network of

physical and biogeochemical processes that controls estuarine $CO_2$ emissions. Over the past
decades, increasingly complex process-based models have been applied, in combination with local
data, to elucidate the coupled carbon-nutrient cycles on the scale of individual estuaries (e.g.,
O'Kane, 1980; Soetaert and Herman, 1995; Vanderborght et al., 2002; Lin et al., 2007; Arndt et al.,
2009; Cerco et al., 2010; Baklouti et al., 2011). However, the application of such model approaches
remains limited to the local scale due to their high data requirements for calibration and validation
(e.g. bathymetric and geometric information and boundary conditions), as well as the high
computational demand associated with resolving the complex interplay of physical and
biogeochemical processes on the relevant temporal and spatial scales (Regnier et al., 2013b).
Complex process-based models are thus not suitable for the application on a regional or global scale
and, as a consequence, the estuarine carbon filter is, despite its increasingly recognized role in
regional and global carbon cycling (e.g. Bauer et al., 2013), typically not taken into account in model-
derived regional or global carbon budgets (Bauer et al., 2013). The lack of regional and global model
approaches that could be used as stand-alone applications or that could be coupled to regional
terrestrial river network models (e.g. GLOBALNEWS: Seitzinger et al., 2005; Mayorga et al., 2010;
SPARROW: Schwarz et al., 2006) and continental shelf models (e.g. Hofmann et al., 2011) is thus
critical.
The Carbon-Generic Estuary Model (C-GEM (v1.0); Volta et al., 2014) has been developed with the
aim of providing such a regional/global modeling tool that can help improve existing, observationally
derived first order estimates of estuarine $CO_2$ emissions. C-GEM (v1.0) has been specifically designed
to reduce data requirements and computational demand and, thus, tackles the main impediments
for the application of estuarine models on a regional or global scale. The approach takes advantage
of the mutual dependency between estuarine geometry and hydrodynamics in alluvial estuaries
and uses an idealized representation of the estuarine geometry to support the hydrodynamic
calculations. It thus allows running steady state or fully transient annual to multi-decadal simulations
for a large number of estuarine systems, using geometric information readily available through maps
or remote sensing images. Although the development of such a regional/global tool inevitably
requires simplification, careful model evaluations have shown that, despite the geometric
simplification, C-GEM provides an accurate description of the hydrodynamics, transport and
biogeochemistry in tidal estuaries (Volta et al., 2014). In addition, the model approach was
successfully used to quantify the contribution of different biogeochemical processes for $CO_2$ air-
water fluxes in an idealized, funnel-shaped estuary forced by typical summer conditions
characterizing a temperate Western European climate (Regnier et al., 2013b). Volta et al. (2016b)
further investigated the effect of estuarine geometry on the $CO_2$ outgassing using three idealized
systems and subsequently established the first regional carbon budget for estuaries surrounding the
North Sea by explicitly simulating the six largest systems of the area (Volta et al., 2016a), including
the Scheldt and the Elbe for which detailed validation was performed.
Here, we extend the domain of application of C-GEM (v1.0) to quantify $CO_2$ exchange fluxes, as well
as the overall organic and inorganic carbon budgets for the full suite of estuarine systems located
along the entire East coast of the United States, one of the most intensively monitored regions in the
world. A unique set of regional data, including partial pressure of $CO_2$ in riverine and continental
shelf waters ($pCO_2$; Signorini et al., 2013; Laruelle et al., 2015), riverine biogeochemical
characteristics (Lauerwald et al., 2013), estuarine eutrophication status (Bricker et al., 2007) and
estuarine morphology (NOAA, 1985) are available. These comprehensive data sets are
complemented by local observations of carbon cycling and $CO_2$ fluxes in selected, individual
estuarine systems (see Laruelle et al., 2013 for a review), making the East coast of the United States
an ideal region for a first, fully explicit regional evaluation of $CO_2$ evasion resolving every major tidal
estuary along the selected coastal segment. The scale addressed in the present study is
unprecedented so far (> 3000 km of coastline) and covers a wide range of estuarine morphological
features, climatic conditions, land-use and land cover types, as well as urbanization levels. The
presented study will not only allow a further evaluation of C-GEM (v1.0), but will also provide the
first regional-scale assessment of estuarine $CO_2$ evasion along the East coast of the US (25 – 45 °N)
and will help explore general relationships between carbon cycling and $CO_2$ evasion, and readily
available estuarine geometrical parameters.
After a description of the model itself and of the dataset used to set up the simulations, a local
validation is presented which includes salinity, $pCO_2$ and pH longitudinal profiles for two well
monitored systems (the Delaware Bay and the Altamaha River Estuary). The yearly averaged rates of
$CO_2$ exchange at the air-water interface simulated by the model for 13 individual estuaries are also
compared with observed values reported in the literature. Next, regional scale simulations for 42
tidal estuaries of the eastern US coast provide seasonal and yearly integrated estimates of the Net
Ecosystem Metabolism (NEM), $CO_2$ evasion and carbon filtering capacity, CFilt. Model results are
then used to elucidate the estuarine biogeochemical behavior along the latitudinal transect
encompassed by the present study (30-45° N). Finally, our results are used to derive general
relationships between carbon cycling and $CO_2$ evasion, and readily available estuarine geometrical
parameters.

**2. Regional description and model approach**
**2.1 Observation-based carbon budget for the East coast of the United States**
The study area covers the Atlantic coast of the United States (Fig.1), from the southern tip of Florida
(25°N) to Cobscook Bay (45°N) at the US-Canada boundary. This area encompasses distinct climatic
zones and land cover types and exhibits a variety of morphologic features (Fig. 1). The region can be
subdivided into several sub-regions following a latitudinal gradient (Signorini et al., 2013). In this
study, we define three sub-regions following the boundaries suggested by the COSCAT segmentation
(Meybeck et al., 2006; Laruelle et al., 2013) and the further subdivision described in Laruelle et al.
(2015). From North to South, the regions are called North Atlantic, Mid Atlantic and South Atlantic
Regions (Fig. 1). Total carbon inputs from watersheds to US East coast estuaries (Tab. 1) have been
estimated to range from 4.0 to 10.7 Tg C $yr^{-1}$ (Mayorga et al., 2010; Shih et al., 2010; Stets and Strieg,
2012; Tian et al., 2010; Tian et al., 2012), consisting of dissolved organic carbon (DOC; ~50%),
dissolved inorganic carbon (DIC; ~40%) and particulate organic carbon (POC; ~10%). In addition, a
statistical approach has been applied to estuaries of the region to quantify organic carbon budgets
and Net Ecosystem Productivity (NEP) using empirical models (Herrmann et al., 2015).
Recent studies estimated that, along the East coast of the United States, rivers emit 11.4 TgC $yr^{-1}$ of
$CO_2$ to the atmosphere (Raymond et al., 2013), while continental shelf waters absorb between 3.4
and 5.4 TgC $yr^{-1}$ of $CO_2$ from the atmosphere (Signorini et al., 2013). A total of thirteen local, annual
mean estuarine $CO_2$ flux estimates across the air-water interface based on measurements are also
reported in the literature and are grouped along a latitudinal gradient (Tab. 2). Four of these
estimates are located in the South Atlantic region (SAR): Sapelo Sound, Doboy Sound, Altamaha
Sound (Jiang et al., 2008), and the Satilla River estuary (Cai and Wang, 1998). Three studies
investigate $CO_2$ fluxes in the mid-Atlantic Region (MAR): the York River Estuary (Raymond et al.,
2000) and the Hudson River (Raymond et al., 1997). There is also a comprehensive $CO_2$ flux study for
the Delaware Estuary published after the completion of this work (Joeseof et al., 2015). Six systems
are located in the North Atlantic region (NAR): The Great Bay, the Little Bay, the Oyster estuary, the
Bellamy estuary, the Cocheco estuary (Hunt et al., 2010; 2011), and the Parker River estuary
(Raymond and Hopkinson, 2003). The mean annual flux per unit area from these local studies is
11.7±13.1 mol C $m^{-2}$ $yr^{-1}$ and its extrapolation to the total estuarine surface leads to a regional $CO_2$
evasion estimate of 3.8 Tg C $y^{-1}$. This estimate is in line with that of Laruelle et al. (2013) for the same
region which proposes an average $CO_2$ emission rate of 10.8 mol C $m^{-2}$ $yr^{-1}$. Thus, $CO_2$ outgassing
could remove 35% to 95% of the riverine carbon loads during estuarine transit. About 75 % of the
air-water exchange occurs in tidal estuaries (2.8 Tg C $y^{-1}$) while lagoons and small deltas contribute to
the remaining 25 %. Although these simple extrapolations from limited observational data are
associated with large uncertainties, they highlight the potentially significant contribution of estuaries
to the $CO_2$ outgassing in the region. However, process-based quantifications of regional organic and
inorganic C budgets including air-water $CO_2$ fluxes for the estuarine systems along the East coast are
not available.
**2.2 Selection of estuaries**
The National Estuarine Eutrophication Assessment (NEAA) survey (Bricker et al., 2007), which uses
geospatial data from the National Oceanic and Atmospheric Administration (NOAA) Coastal
Assessment Framework (CAF) (NOAA, 1985), was used to identify and characterize 58 estuarine
systems discharging along the Atlantic coast of the United States. From this set, 42 'tidal' estuaries,
defined as a river stretch of water that is tidally influenced (Dürr et al., 2011), were retained (Fig. 1)
to be simulated by the C-GEM model, which is designed to represent such systems. Using outputs
from terrestrial models (Hartmann et al., 2009; Mayorga et al., 2010), the cumulated riverine carbon
loads for all the non-tidal estuaries that are excluded from the present study amount to 0.9 Tg C yr$^{-1}$,
which represents less than 15% of the total riverine carbon loads of the region. These 16 systems are
located in the SAR (10) and in the MAR (6).
The northeastern part of the domain (NAR, Fig. 1; Tab. 1) includes 11 estuaries along the Gulf of
Maine and the Scotian shelf, covering a cumulative surface area of 558 km². It includes drowned
valleys, rocky shores and a few tidal marshes. The climate is relatively cold (annual mean= 8°C) and
the human influence is relatively limited because of low population density and low freshwater
inputs. The mean estuarine water depth is 12.9 m and the mean tidal range is 2.8 m.
The central zone (MAR) includes 18 tidal estuaries accounting for a total surface area of 9298 km².
The Chesapeake Bay and the Delaware estuaries alone contribute more than 60% to the surface area
of the region. In this region, estuaries are drowned valleys with comparatively high river discharge
and intense exchange with the ocean. Several coastal lagoons, characterized by a limited exchange
with the ocean are located here, but are not included in our analysis. The Mid-Atlantic Region (MAR)
is characterized by a mean annual temperature of 13°C and is strongly impacted by human activities,
due to the presence of several large cities (e.g. New York, Washington, Philadelphia, Baltimore) and
intense agriculture. The mean water depth is about 4.7 m and the tidal range is 0.8 m.
The southern Atlantic region (SAR) includes 13 tidal estuaries covering a total surface area of 959
km². These systems are generally dendritic and surrounded by extensive salt marshes. The climate is
subtropical with an average annual temperature of 19°C. Land use includes agriculture and industry,
but the population density is generally low. Estuarine systems in the SAR are characterized by a
shallow mean water depth of 2.9 m and a tidal range of 1.2 m.
**2.3 Model set-up**
The generic 1D Reactive-Transport Model (RTM) C-GEM (Volta et al., 2014) is used to quantify the
estuarine carbon cycling in the 42 systems considered in this study. The approach is based on
idealized geometries (Savenije, 2005; Volta et al., 2014) and is designed for regional and global scale
applications (Regnier et al., 2013b; Volta et al., 2014, 2016a). The model approach builds on the
premise that hydrodynamics exerts a first-order control on estuarine biogeochemistry (Arndt et al.,
2007; Friedrichs and Hofmann, 2001) and $CO_2$ fluxes (Regnier et al., 2013a). The method takes
advantage of the mutual dependence between geometry and hydrodynamics in tidal estuaries
(Savenije, 1992) and the fact that, as a consequence, transport and mixing can be easily quantified
from readily available geometric data (Regnier et al., 2013a; Savenije, 2005; Volta et al., 2016b).
**2.3.1 Description of idealized geometries for tidally-averaged conditions**
Although tidal estuaries display a wide variety of shapes, they nevertheless share common
geometric characteristics that are compatible with an idealized representation (Fig. 2, Savenije,
1986; Savenije, 2005). For tidally-averaged conditions, their width B (or cross-sectional area A) can
be described by an exponential decrease as a function of distance, *x*, from the mouth (Savenije,
1986; Savenije, 2005):

$$B = B0 * \exp\left(-\frac{x}{b}\right)$$

(1)

where B (m) is the tidally averaged width, B0 (m) the width at the mouth, x (m) the distance from
the mouth (x=0) and b (m) the width convergence length (Fig. 2). The width convergence length, b, is
defined as the distance between the mouth and the point at which the width is reduced to B0 $e^{-1}$. It
is directly related to the dominant hydrodynamic forcing. A high river discharge typically results in a
prismatic channel with long convergence length (river dominated estuary), while a large tidal range
results in a funnel-shaped estuary with short convergence length (marine dominated estuary). At the
upstream boundary, the estuarine width is given by:

$$B_L = B0 * \exp\left(-\frac{L}{b}\right)$$

(2)

Where L denotes the total estuarine length (m) along the estuarine longitudinal axis.
The total estuarine surface S ($m^2$) can be estimated by integrating equation (1) over the estuarine
length:

$$S = \int_0^L B \, dx = b * B0 * \left(1 - \exp\left(-\frac{L}{b}\right)\right)$$

(3)


The width convergence length is then calculated from B0, $B_L$, L and the real estuarine surface area
(SR) by inserting equation (2) in equation (3):

$$b = \frac{SR}{B0 - BL}$$

(4)

SR is calculated for each system using the SRTM water body data (Fig. 3a), a geographical dataset
encoding high-resolution worldwide coastal outlines in a vector format (NASA/NGA, 2003). While
such a database exists for a well monitored region such as the East coast of the US, resorting to
using the idealized estuarine surface area (S) is necessary in many other regions. The longitudinal
mean, tidally averaged, depth h (m), is obtained from the National Estuarine Eutrophication
Assessment database (Bricker et al., 2007).
Using this idealized representation, the estuarine geometry can be defined by a limited number of
parameters: the width at the mouth ($B_0$), the estuarine length (L), the estuarine width at the
upstream limit ($B_L$) and the mean depth h. These parameters can be easily determined from local
maps or Google Earth using Geographic Information Systems (GIS) or obtained from databases
(NASA/NGA, 2003).
**2.3.2 Hydrodynamics, transport and biogeochemistry**
Estuarine hydrodynamics are described by the one-dimensional barotropic, cross-sectionally
integrated mass and momentum conservation equations for a channel with arbitrary geometry
(Nihoul and Ronday, 1976; Regnier et al., 1998; Regnier and Steefel, 1999):
$$r_s \frac{\partial A}{\partial t} + \frac{\partial Q}{\partial x} = 0 \tag{5}$$

$$\frac{\partial U}{\partial t} + U \frac{\partial U}{\partial x} = -g \frac{\partial \zeta}{\partial x} - g \frac{U|U|}{C_z^2 H} \tag{6}$$

where:
*t*       time                                                    [s]
*x*       distance along the longitudinal axis                    [m]
*A*       cross-section area $A = H \cdot B$                       [m$^2$]
*Q*       cross-sectional discharge $Q = A \cdot U$               [m$^3$ s$^{-1}$]
*U*       flow velocity $Q / A$                                   [m s$^{-1}$]
$r_s$     storage ratio $r_s = B_s / B$                          [-]
$B_s$      storage width                                   [m]
$g$       gravitational acceleration                    [m s$^{-2}$]
$\xi$       elevation                                      [m]
$H$      total water depth $H = h + \xi(x,t)$          [m]
$C_z$      Chézy coefficient                            [m$^{1/2}$ s$^{-1}$]
The coupled partial differential equations (Eqs. (5) and (6)) are solved by specifying the elevation
$\xi_0(t)$ at the estuarine mouth and the river discharge $Q_r(t)$ at the upstream limit of the model domain.
The one-dimensional, tidally-resolved, advection-dispersion equation for a constituent of
concentration $C(x,t)$ in an estuary can be written as (e.g. Pritchard, 1958):
$$\frac{\partial C}{\partial t} + \frac{Q}{A}\frac{\partial C}{\partial x} = \frac{1}{A}\frac{\partial}{\partial x}\left(AD\frac{\partial C}{\partial x}\right) + P \tag{7}$$

where $Q(x,t)$ and $A(x,t)$ denote the cross-sectional discharge and area, respectively and are provided
by the hydrodynamic model (eq. 5 and 6). $P(x,t)$ is the sum of all production and consumption
process rates affection the concentration of the constituent. The effective dispersion coefficient $D$
(m$^2$ s$^{-1}$) implicitly accounts for dispersion mechanisms associated to sub-grid scale processes (Fischer,
1976; Regnier et al., 1998). In general, $D$ is maximal near the sea, decreases upstream and becomes
virtually zero near the tail of the salt intrusion curve (Preddy, 1954; Kent, 1958; Ippen and Harleman,
1961; Stigter and Siemons, 1967). The effective dispersion at the estuarine mouth can be quantified
by the following relation (Savenije, 1986):
$$D_0 = 26 \cdot (h_0)^{1.5} \cdot (N \cdot g)^{0.5} \tag{8}$$

where $h_0$ (m) is the tidally-averaged water depth at the estuarine mouth and $N$ is the dimensionless
Canter Cremers' estuary number defined as the ratio of the freshwater entering the estuary during a
tidal cycle to the volume of salt water entering the estuary over a tidal cycle (Simmons, 1955).
$N = \frac{Q_b \cdot T}{P}$                    (9)
In this equation, $Q_b$ is the bankfull discharge (m$^3$ s$^{-1}$), T is the tidal period (s) and P is the tidal prism
(m$^3$). For each estuary, N can thus be calculated directly from the hydrodynamic model. The
variation in $D$ along the estuarine gradient can be described by Van der Burgh's equation (Savenije,

266    1986):

$$\frac{\partial D}{\partial x} = -K \frac{Q_r}{A}$$                    (10)
where $K$ is the dimensionless Van der Burgh's coefficient and the minus sign indicates that $D$
increases in downstream direction (Savenije, 2012). The Van der Burgh's coefficient is a shape factor
that has values between 0 and 1 (Savenije, 2012), and is a function of estuarine geometry for tidally
average conditions. Therefore, each estuarine system has its own characteristic $K$ value, which
correlates with geometric and hydraulic scales (Savenije, 2005). Based on a regression analysis
covering a set of 15 estuaries, it has been proposed to constrain $K$ from the estuarine geometry
(Savenije, 1992):
$$K = 4.32 \cdot \frac{h_0^{0.36}}{B_0^{0.21} \cdot b^{0.14}} \quad \text{with} \quad 0 < K < 1$$                    (11)
Reaction processes $P$ considered in C-GEM comprise aerobic degradation, denitrification,
nitrification, primary production, phytoplankton mortality and air-water gas exchange for $O_2$ and $CO_2$
(Fig. 4 and Tab. 3). These processes and their mathematical formulation are described in detail in
Volta et al. (2014) and Volta et al. (2016a).
The non-linear partial differential equations for the hydrodynamics are solved by a finite difference
scheme following the approach of (Regnier et al., 1997; Regnier and Steefel, 1999) and
(Vanderborght et al., 2002). The timestep Δt is 150s and the grid size Δx is constant along the
longitudinal axis of the estuary. The grid size default value is 2000 m, but can be smaller for short
length estuaries to guarantee a minimum of 20 grid points within the computational domain.
Transport and reaction terms are solved in sequence within a single timestep using an operator
splitting approach (Regnier et al., 1997). The advection term in the transport equation is integrated
using a third-order accurate total variation diminishing (TVD) algorithm with flux limiters (Regnier et
al., 1998), ensuring monotonicity (Leonard, 1984), while a semi-implicit Crank-Nicholson algorithm is
used for the dispersion term (Press et al., 1992). These schemes have been extensively tested using
the CONTRASTE estuarine model (e.g. Regnier et al., 1998; Regnier and Steefel, 1999; Vanderborght
et al., 2002) and guarantee mass conservation to within <1%. The reaction network (including
erosion-deposition terms when the constituent is a solid species), is numerically integrated using the
Euler method (Press et al., 1992). The primary production dynamics, which takes into account the
combined effects of nutrients limitation and light attenuation in the water column induced by its
background turbidity and SPM concentration, requires vertical resolution of the photic depth. The
latter is calculated according to the method described in Vanderborght et al. (2007). This method
assumes an exponential decrease of the light in the water column (Platt et al., 1980), which is solved
using a Gamma function.
**2.4 Boundary and forcing conditions**
Boundary and forcing conditions are extracted from global databases and global model outputs that
are available at 0.5° resolution. Therefore, C-GEM simulations are performed at the same resolution
according to the following procedure. First, 42 coastal cells corresponding to tidal estuaries are
identified in the studied area (Fig. 1). If the mouth of an estuary is spread over several 0.5° grid cells,
those cells are regrouped in order to represent a single estuary (e.g. Delaware estuary), and
subsequently, a single idealized geometry is defined as described above. The model outputs
(Hartmann et al. , 2009; Mayorga et al., 2010) and databases (Antonov et al., 2010; Garcia et al.,
2010a; Garcia et al., 2010b) used to constrain our boundary conditions are representative of the
year 2000.
For each resulting cell, boundary and forcing conditions are calculated for the following periods:
January-March; April-June; July-September and October-December. This allows for an explicit
representation of the seasonal variability in the simulations.
**2.4.1 External forcings**
Transient physical forcings are calculated for each season and grid cell using monthly mean values of
water temperature (World Ocean Atlas: Antonov et al. 2010; Locarini et al., 2010) and seasonal
averaged values for wind speed (Cross-Calibrated-Multi-Platform (CCMP) Ocean Surface Wind
Vector Analyses project (Atlas et al., 2011)). Mean daily solar radiation and photoperiods (corrected
for cloud coverage using the ISCCP Cloud Data Products, Rossow and Schiffer, 1999) are calculated
depending on latitude and day of the year using a simple model (Brock, 1981).
**2.4.2 Riverine discharge, concentrations and fluxes**
River discharges are extracted from the UNH/GRDC runoff dataset (Fekete et al., 2002). These
discharges represent long-term averages (1960-1990) of monthly and annual runoff at 0.5 degree
resolution. The dataset is a composite of long-term gauging data, which provides average runoff for
the largest river basins, and a climate driven water balance model (Fekete et al., 2002). Total runoff
values are then aggregated for each watershed at the coarser 0.5 degree resolution (Fig. 3b). Next,
seasonal mean values (in $m^3 s^{-1}$) are derived in order to account for the intra-annual variability in
water fluxes. Based on annual carbon and nutrients inputs from the watersheds (Mg $y^{-1}$), mean
annual concentrations (mmol $m^{-3}$) are estimated for each watershed using the UNH/GRDC annual
runoff ($km^3 y^{-1}$). Mean seasonal concentrations are then calculated from the seasonally resolved
river water fluxes of a given sub-region.
Annual inputs of dissolved organic carbon (DOC), particulate organic carbon (POC) and inorganic
nutrients are derived from the globalNEWS2 model (Mayorga et al., 2010). Global NEWS is a spatially
explicit, multi-element (N, P, Si, C) and multi-form global model of nutrient exports by rivers. In a
nutshell, DOC exports are a function of runoff, wetland area, and consumptive water use (Harrison
et al., 2005). No distinction is made between agricultural and natural landscapes, since they appear
to have similar DOC export coefficients (Harrison et al., 2005). Sewage inputs of OC are ignored in
GlobalNEWS, because their inclusion did not improve model fit to data (Harrison et al., 2005). POC
exports from watersheds are estimated using an empirical relationship with Suspended Particulate
Matter (SPM; Ludwig et al., 1996). Inorganic nitrogen (DIN) and phosphorus (DIP) fluxes calculated
by GlobalNEWS depend on agriculture and tropical forest coverage, fertilizer application, animal
grazing, sewage input, atmospheric N deposition and biological N fixation (Mayorga et al., 2010). The
inputs of dissolved silica (DSi) are controlled by soil bulk density, precipitation, slope, and presence
of volcanic lithology (Beusen et al., 2009).
The DIN speciation is not provided by the GlobalNEWS2 model. The $NH_4$ and $NO_3$ concentrations are
therefore determined independently on the basis of an empirical relationship between ammonium
fraction (NH4/DIN ratio) and DIN loads (Meybeck, 1982). Dissolved Oxygen (DO) concentrations are
extracted from the water quality criteria recommendations published by the United States
Environmental Protection Agency (EPA, 2009). The same source is used for phytoplankton
concentrations, using a chlorophyll-a to phytoplankton carbon ratio of 50 gC (gChla)$^{-1}$ (Riemann et
al., 1989) to convert the EPA values to carbon units used in the present study.
Inputs of dissolved inorganic carbon (DIC) and total Alkalinity (ALK) are calculated from values
reported in the GLORICH database (Hartmann et al., 2009). For each watershed, seasonal mean
values of DIC and ALK concentrations are estimated from measurements performed at the sampling
locations that are closest to the river-estuary boundary. The spatial distribution of annual inputs of
TOC=DOC+POC, DIC, and TC=TOC+DIC from continental watersheds to estuaries are reported in Fig.
5a, 5c and 5d, respectively. The contribution of tidal wetlands to the TOC inputs is also shown (Fig.
5b). Overall, the TC input over the entire model domain is estimated at 4.6 Tg C yr[-1], which falls in
the lower end of previous reported estimations (Najjar et al. 2012).

**2.4.3 Inputs from tidal wetlands**
The DOC input of estuarine wetlands (Fig. 5b) scales to their fraction, W, of the total estuarine and is
calculated using the GlobalNEWS parameterization:

$$Y\_DOC = \frac{[(E\_C_{wet} * W) + E\_C_{dry} * (1 - W)] * R^a * Q_{act}}{Q_{nat}} \qquad (12)$$


$$\frac{Y\_DOC_{wet}}{Y\_DOC} = \frac{E\_C_{wet} * W}{E\_C_{wet} * W + E\_C_{dry} * (1 - W)} \qquad (13)$$


where Y_DOC is the DOC yield (kg C km$^{-2}$ y$^{-1}$) calculated for the entire watershed, Y_DOC$_{wet}$ is the
estimated DOC yield from wetland areas (kg C km$^{-2}$ y$^{-1}$), Q$_{act}$/Q$_{nat}$ is the ratio between the measured
discharge after dam construction and before dam construction, E_C$_{wet}$ and E_C$_{dry}$ (kg C km$^{-2}$ y$^{-1}$) are
the export coefficients of DOC from wetland and non-wetland soils, respectively. W is the
percentage of the land area within a watershed that is covered by wetlands, R is the runoff (m y$^{-1}$)
and *a* is a unit-less calibration coefficient defining how non-point source DOC export responds to
runoff. The value of a is set to 0.95, consistent with the original GlobalNEWS -DOC model of Harrison
et al. (2005). The carbon load Y_DOC$_{wet}$ is then exported as a diffuse source along the relevant
portions of estuary. The estuarine segments receiving carbon inputs from tidal wetlands are
identified using the National Wetlands Inventory of the U.S. Fish and Wildlife Service (U.S. Fish and
Wildlife Service, 2014). The inputs from those systems are then allocated to the appropriate grid cell
of the model domain using GIS. The flux calculated is an annual average that is subsequently
partitioned between the four seasons as a function of the mean seasonal temperature, assumed to
be the main control of the wetland-estuarine exchange. This procedure reflects the observation that
in spring and early summer, DOC export is small as a result of its accumulation in the salt marshes
induced by the high productivity (Dai and Wiegert, 1996), (Jiang et al., 2008). In late summer and fall,
the higher water temperature and greater availability of labile DOC contribute to higher bacterial
remineralization rates in the intertidal marshes (Cai et al., 1999; Middelburg et al., 1996; Wang and
Cai, 2004), which induce an important export. This marsh production-recycle-export pattern is
consistent with the observed excess DIC signal in the offshore water (Jiang et al. 2013). DIC export
from tidal wetlands is neglected here because it is assumed that OC is not degraded before reaching
the estuarine realm. Although this assumption may lead to an overestimation of OC export from
marshes and respiration in estuarine water, it will not significantly affect the water $pCO_2$ and
degassing in the estuarine waters because mixing is faster than respiration.
**2.4.4 Concentrations at the estuarine mouth**
For each estuary, the downstream boundary is located 20 km beyond the mouth to minimize the
bias introduced by the choice of a fixed concentration boundary condition to characterize the ocean
water masses (e.g. Regnier et al., 1998). This approach also reduces the influence of marine
boundary conditions on the simulated estuarine dynamics, especially for all organic carbon species
whose concentrations are fixed at zero at the marine boundary. This assumption ignores the
intrusion of marine organic carbon into the estuary during the tidal cycle but allows focusing on the
fate of terrigenous material and its transit through the estuarine filter. DIC concentrations are
extracted from the GLODAP dataset (Key et al., 2004), from which ALK and pH are calculated
assuming $CO_2$ equilibrium between coastal waters and the atmosphere. The equilibrium value is
computed using temperature (WOA2009, Locarnini et al., 2010) and salinity (WOA2009, Antonov et
al. (2010)) data which vary both spatially and temporally. The equilibrium approach is a reasonable
assumption because differences in partial pressure $\Delta pCO_2$ between coastal waters and the
atmosphere are generally much smaller (0-250 µatm (Signorini et al., 2013)) than those reported for
estuaries ($\Delta pCO_2$ in the range 0-10000 µatm (Borges and Abril, 2012)). Salinity, DO, $NO_3$, DIP and DSi
concentrations are derived from the World Ocean Atlas (Antonov et al., 2010; Garcia et al., 2010a;
Garcia et al., 2010b). $NH_4$ concentrations are set to zero in marine waters. For all variables, seasonal
means are calculated for each grid cell of the boundary.

**2.5 Biogeochemical indicators**
The model outputs (longitudinal profiles of concentration and reaction rates) are integrated in time
over the entire volume or surface of each estuary to produce the following indicators of the
estuarine biogeochemical functioning (Regnier et al., 2013b): the mean annual Net Ecosystem
Metabolism (*NEM*), the air-water $CO_2$ flux (*FCO$_2$*), the carbon and nitrogen filtering capacity (*CFilt*
and *NFilt*) and their corresponding element budgets. The *NEM* (molC $y^{-1}$) (Caffrey, 2004; Odum,
1956) is defined as the difference between net primary production (*NPP*) and total heterotrophic
respiration (*HR*) at the system scale:

$$NEM = \int_0^{365} \int_0^L [NPP(x,t) - R_{aer}(x,t) - R_{den}(x,t)] * B(x) * H(x,t)\ dx\ dt$$

(14)


where *NPP* is the Net Primary Production (mol C $m^{-3}$ $y^{-1}$), $R_{aer}$ the aerobic degradation of organic
matter (in mol C $m^{-3}$ $y^{-1}$) and $R_{den}$ the denitrification (in mol C $m^{-3}$ $y^{-1}$) (see Volta et al., 2014 for
detailed formulations). *NEM* is thus controlled by the production and decomposition of
autochthonous organic matter, by the amount and degradability of organic carbon delivered by
rivers and tidal wetlands and by the export of terrestrial and in-situ produced organic matter to the
adjacent coastal zone. Following the definition of *NEM*, the trophic status of estuaries can be net
heterotrophic (NEM<0) when *HR* exceeds *NPP* or net autotrophic (NEM>0), when *NPP* is larger than
*HR* because the burial and export of autochthonous organic matter exceeds the decomposition of
river-borne material.
The $FCO_2$ (mol C $y^{-1}$) is defined as:

$$FCO_2 = \int\limits_0^{365} \int\limits_0^{L} RCO_2(x,t) * B(x) \, dx \, dt \qquad (15)$$


$$RCO_2(x,t) = -v_p(x,t)\left([CO_{2(aq)}](x,t) - K_0(x,t) * P_{CO2}(x,t)\right) \qquad (16)$$


where $RCO_2$ (molC $m^{-2}$ $y^{-1}$) is the rate of exchange in $CO_2$ at the air-water interface per unit surface
area, $v_p$ is the piston velocity (m $y^{-1}$) and is calculated according to Regnier et al. (2002) to account
for the effect of current velocity and wind speed, [*CO2(aq)*] is the concentration of $CO_2$ in the
estuary (mol $m^{-3}$), $K_0$ is Henry's constant of $CO_2$ in sea water (mol $m^{-3}$ $atm^{-1}$) and $P_{CO2}$ is the
atmospheric partial pressure in $CO_2$ (atm).
The carbon filtering capacity (in %) corresponds to the fraction of the river-borne supply that is lost
to the atmosphere and is defined here as the ratio of the net outgassing flux of $CO_2$ and the total
inputs of C, e.g. total carbon expressed as the sum of inorganic and organic carbon species, both in
the dissolved and particulate phases.
$CFilt = \dfrac{FCO_2}{\int_0^{365} Q*[TC]_{riv} \, dt} * 100 \qquad (17)$
where *[TC]$_{riv}$* denote the total concentrations of C in the riverine inputs.
Fluxes per unit area for *FCO₂* and *NEM*, noted $\overline{FCO_2}$ and $\overline{NEM}$, respectively, are defined in mol C $m^{-2}$
$y^{-1}$ and are calculated by dividing the integrated values calculated above by the (idealized) estuarine
surface *S*:
$$\overline{NEM} = \frac{NEM}{S} * 1000 \qquad\qquad (18)$$
$$\overline{FCO_2} = \frac{FCO_2}{S} * 1000 \qquad\qquad (19)$$
Seasonal values for the biogeochemical indicators are calculated using the same formula as above,
but calculate the integral over a seasonal rather than annual timescale (i.e. 3 months).

**2.6 Model-data comparison**
C-GEM has been specifically designed for an application on a global/regional scale requiring the
representation of a large number of individual and often data-poor systems. Maximum model
transferability and minimum validation requirements were thus central to the model design process
and the ability of the underlying approach in reproducing observed dynamics with minimal
calibration effort has been extensively tested. The performance C-GEM's one-dimensional
hydrodynamic and transport models using idealized geometries have been evaluated for a number
of estuarine systems exhibiting a wide variety of shapes (Savenije, 2012). In particular, it has been
shown that the estuarine salt intrusion can be successfully reproduced using the proposed modeling
approach (Savenije 2005; Volta et al., 2014; 2016b). In addition, C-GEM's biogeochemistry has also
been carefully validated for geometrically contrasting estuarine system in temperate climate zones.
Simulations for the Scheldt Estuary (Belgium and the Netherlands), a typical funnel-shaped estuary,
were validated through model-data and model-model comparison (Volta et al., 2014; Volta et al.,
2016a). Furthermore, simulations for the Elbe estuary (Germany), a typical prismatic shape estuary
that drains carbonate terrains and, thus, exhibits very high pH was validated against field data (Volta
et al., 2016a). In addition, C-GEM carbon budgets have been compared budget derived from
observations for 6 European estuaries discharging in the North Sea (Volta et al., 2016a). Although C-
GEM has been specifically designed and tested for the type of regional application presented here,
its transferability from North Sea to US East Coast estuaries was further evaluated by assessing its
performance in two East Coast estuaries. First, the hydrodynamic and transport model was tested
for the Delaware Bay (MAR). The model was forced with the monthly, minimal and maximal
observed discharge at Trenton over the period between 1912 and 1985 (UNH/GRDC Database,
GRDC, 2014). Simulated salinity profiles are compared with salinity observations from January,
February, May and June (the months with the highest number of data entries), which were extracted
from the UNH/GRDC Database. Figure 6 shows that the model captures both the salinity intrusion
length and the overall shape of the salinity profile well. In addition, the performance of the
biogeochemical model and specifically its ability to reproduce pH and $pCO_2$ profiles was evaluated by
a model-data comparison for both the Delaware Bay (MAR) in July 2003 and the Altamaha river
estuary (SAR) in October 1995. Similar to Volta et al., 2016a, the test systems were chosen due to
their contrasting geometries. The Delaware Bay is a marine dominated system characterized by a
pronounced funnel shape, while the Altamaha River has a prismatic estuary characteristic of river
dominated systems (Jiang et al., 2008). Monthly upstream boundary conditions for nutrients, as well
as observed pH data and calculated $pCO_2$ are extracted from datasets described in (Sharp, 2010) and
(Sharp et al., 2009) for the Delaware and in Cai and Wang (1998), Jiang et al. (2008) and (Cai et al.,
1998) for the Altamaha river estuary. The additional forcings and boundary conditions are set
similarly to the simulation for 2000 (see Tab. 2, 3, 4, 5, 6 in SI). Figure 7 shows that measured and
simulated pH values are in good agreement with observed pH and observation-derived calculations
of $pCO_2$. In the Delaware Bay, a pH minimum is located around km 140 and is mainly caused by
intense nitrification sustained by large inputs of $NH_4$ from the Philadelphia urban area, coupled to an
intense heterotrophic activity. Both processes lead to a well-developed $pCO_2$ increase in this area
(Fig. 7c). Overall, the longitudinal $pCO_2$ profile of the Delaware estuary is characterized by values
close to equilibrium with the atmosphere in the widest section of the Delaware Bay (near the
estuarine mouth and throughout the 40 first kilometers of the system) and values above 1200 µatm
at kilometer 150 and beyond, where characteristic salinities are below 5. Although the profile
presented here is simulated using boundary conditions representative of July 2003 and no $pCO_2$ data
were available for validation for this period, a recent study by Joesoef et al. (2015) reports a similar
longitudinal $pCO_2$ profile in July 2013. For the Altamaha river estuary, pH steadily increases from
typical river to typical coastal ocean values (Fig. 7b). In addition, both observations and model
results reveal that outgassing is very intense in the low-salinity region with more than a 5 fold
decrease in $pCO_2$ between salinity 0 and 5 (Fig. 7d).
While such local validations allow assessing the performance of the model for a specific set of
conditions, the purpose of this study is to capture the average biogeochemical behavior of the
estuaries of the eastern coast of the US. Therefore, in addition to the system-specific validation,
published annually averaged $FCO_2$ estimates for 12 tidal systems located within the study area
collected over the 1994-2006 period are compared to simulated $FCO_2$ for conditions representative
of the year 2000. Overall, simulated $FCO_2$ are comparable to values reported in the literature (Tab.
2). Although significant discrepancies are observed at the level of individual systems, the model
captures remarkably well the overall behaviors of estuaries along the East coast of the US in term of
intensity of $CO_2$ evasion rate. The model simulates low $CO_2$ efflux (< 5 mol C m$^{-2}$ yr$^{-1}$) for the 6
systems were such conditions have been observed, while the 5 systems for which the $CO_2$ evasion
exceeds 10 mol C m$^{-2}$ yr$^{-1}$ are the same in the observations and in the model runs. The discrepancies
at the individual system level likely result from a combination of factors, including the choice of
model processes and there parametrization, the uncertainties in constraining boundary conditions
and the limited representability of instantaneous and local observations.
**3 Results and discussion**
**3.1 Spatial variability of estuarine carbon dynamics**
Figure 8 presents the spatial distribution of simulated mean annual $\overline{FCO_2}$ and -$\overline{NEM}$ (Fig. 8a), as well
as $FCO_2$ and -$NEM$ (Fig. 8b). In general, mean annual $\overline{FCO_2}$ are about 30% larger than mean annual
$\overline{NEM}$, with the exception of six estuaries situated in the North of the coastal segment. Overall, the
$\overline{NEM}$ is characterized by smaller system to system variability compared to the $\overline{FCO_2}$ in all regions. In
addition, Fig. 8 reveals distinct differences across the three coastal segments and highlights the
important influence of the estuarine geometry and residence time, as well as the latitudinal
temperature gradient on estuarine carbon cycling.
Overall, $\overline{FCO_2}$ values are the lowest in the NAR (mean flux = 17.3 ± 16.4 mol C m$^{-2}$ y$^{-1}$; surface
weighted average = 23.1 mol C m$^{-2}$ y$^{-1}$), consistent with previously reported very low values for small
estuaries surrounding the Gulf of Maine (Hunt et al., 2010; 2011; Tab. 2). In contrast, $\overline{NEM}$ reveals a
regional minimum in the NAR (-51.2 ± 16.6 mol C m$^{-2}$ y$^{-1}$; surface weighted average = -52.8 mol C m$^{-2}$
y$^{-1}$). The MAR is characterized by intermediate values for $\overline{FCO_2}$, with a mean flux of 26.3 ± 34.6 mol
C m$^{-2}$ y$^{-1}$ (surface weighted average =11.1 mol C m$^{-2}$ y$^{-1}$) and lowest values for $\overline{NEM}$ (-15.1 ± 14.2 mol
C m$^{-2}$ y$^{-1}$; surface weighted average =-7.4 mol C m$^{-2}$ y$^{-1}$). This region also shows the largest variability
in $CO_2$ outgassing compared to the NAR and SAR, with the standard deviation exceeding the mean
$\overline{FCO_2}$ , and individual estimates ranging from 3.9 mol C m$^{-2}$ y$^{-1}$ to 150.8 mol C m$^{-2}$ y$^{-1}$. This variability
is mainly the result of largely variable estuarine surface areas and volumes. Some of the largest East
coast estuaries (e.g. Chesapeake and Delaware Bays), as well as some of smallest estuaries (e.g. York
River and Hudson River estuaries, Raymond et al., 1997; 2000), are located in this region (Tab. 2 and
4). The maximum values of 150.8 mol C m$^{-2}$ y$^{-1}$ simulated in the MAR are similar to the highest $FCO_2$
reported in the literature (132.3 mol C m$^{-2}$ y$^{-1}$ for the Tapti estuary in India; Sarma et al., 2012). The
SAR is characterized by the highest mean $\overline{FCO_2}$ (46.7 ± 33.0 mol C m$^{-2}$ y$^{-1}$; surface weighted average
= 40.0 mol C m$^{-2}$ y$^{-1}$) and intermediate $\overline{NEM}$ (-36.8 ± 24.7 mol C m$^{-2}$ y$^{-1}$; surface weighted average = -
31.2 mol C m$^{-2}$ y$^{-1}$).
The NAR is characterized by a regional minimum in $\overline{FCO_2}$ , and only contributes 4.6% to the total
$FCO_2$ of the East coast of the US, owing to the small cumulative surface area available for gas
exchange in its 10 estuarine systems. In contrast, the 18 MAR estuaries, with their large relative
contribution to the total regional estuarine surface area, account for as much as 70.1% of the total
outgassing. Because of their smaller cumulated surface area compared to those of the MAR, the 14
SAR estuaries account for merely 25.3% of the total outgassing despite their regional maximal $\overline{FCO_2}$.
A similar, yet slightly less pronounced pattern emerges for the $\overline{NEM}$. The NAR, MAR and SAR
respectively contribute 13.7%, 60.7% and 25.6% to the total regional net ecosystem metabolism. The
comparatively larger relative contribution of the NAR to the total *NEM* as compared to the total
*FCO₂* can be explained by the importance of the specific aspect ratio for NEM. A larger ratio of
estuarine width B0 and convergence length b corresponds to a more funnel shaped estuary while a
low ratio corresponds to a more prismatic geometry (Savenije, 2005; Volta et al., 2014). In the NAR,
estuaries are generally characterized by relatively narrow widths and deep-water depths, thus
limiting the potential surface area for gas exchange with the atmosphere. However, the relative
contribution of each region to the total regional *NEM* and *FCO₂* is largely controlled by estuarine
surface area. Figure 9 illustrates the cumulative *NEM* (a) and *FCO₂* (b) as a function of the cumulative
estuarine surface areas. The disproportionate contribution of large estuaries from the MAR
translates into a handful of systems (Chesapeake and Delaware Bays and the main tributaries of the
former, in particular) contributing to roughly half of the regional *NEM* and *FCO₂*, in spite of relatively
low individual rates per unit surface area. However, the smallest systems (mostly located in the NAR
and SAR) nevertheless still contribute a significant fraction to the total regional *NEM* and *FCO₂*. The
27 smallest systems merely account for less than 10% of the total regional estuarine surface area,
yet contribute 38% and 29% to the total regional *NEM* and *FCO₂*, respectively (Fig. 9). This
disproportioned contribution can be mainly attributed to their high individual $\overline{FCO_2}$ and $\overline{NEM}$. This
is illustrated by the average simulated $\overline{FCO_2}$ for all 27 smallest systems (calculated as the sum of
each estuarine $CO_2$ outgassing per unit surface area divided by the total number of estuarine
systems) which is significantly higher (30.2 mol C m$^{-2}$ y$^{-1}$) than its surface weighted average (14 mol C
m$^{-2}$ y$^{-1}$). Thereby accounting for the disproportionate contribution of very large systems (calculated
as the sum of each estuarine $CO_2$ outgassing divided by the total estuarine surface area across the
region).
Following the approach used in Regnier et al. (2013b), the contribution of each biogeochemical
process to $FCO_2$ is assessed by evaluating their individual contribution to DIC and ALK changes taking
into account the local buffering capacity of an ionic solution when TA and DIC are changing due to
internal processes, but ignoring advection and mixing (Zeebe and Wolf-Gladrow 2001). In the
present study, we quantify the effect of the NEM on the $CO_2$ balance, which is almost exclusively
controlled by aerobic degradation rates because the contributions of denitrification and NPP to the
net ecosystem balance are small. Nitrification, a process triggered by the transport and/or
production of $NH_4$ in oxygenated waters, favors outgassing through its effect on pH, which shifts the
acid-base equilibrium of carbonate species and increases the $CO_2$ concentration. The contribution of
supersaturated riverine waters to the overall estuarine $CO_2$ dynamics is calculated as difference
between all the other processes creating or consuming $CO_2$.. Figure 10a presents the contribution of
the annually integrated $NEM$, nitrification and evasion of supersaturated, DIC enriched riverine
waters to the total outgassing for each system, as well as for individual regions of the domain. The
calculation of these annual values is based on the sum of the seasonal fluxes. Model results reveal
that, regionally, the $NEM$ supports about 50% of the estuarine $CO_2$ outgassing, while nitrification and
riverine DIC inputs sustain about 17% and 33% of the $CO_2$ emissions, respectively. The relative
significance of the three processes described above shows important spatial variability. In the NAR,
oversaturated riverine waters and NEM respectively sustain 50% and 44% of the outgassing within
the sub-region, while nitrification is of minor importance (6%). In the MAR, the contribution of
riverine DIC inputs is significantly lower (~30%) and the main contribution to the outgassing is $NEM$
(~50%); nitrification accounting for slightly less than 20% of the outgassing. In the SAR, the riverine
contribution is even lower (~20%), and the outgassing is mainly attributed to the $NEM$ (~55%) and
nitrification (~25%). Therefore, although the model results reveal significant variability across
individual systems, a clear latitudinal trend in the contribution to the total $FCO_2$ emerge from the
analysis; the importance of oversaturated riverine water decreasing from North to South, while $NEM$
and nitrification increase along the same latitudinal gradient. The increasing relative importance of
estuarine biogeochemical processes over riverine DIC inputs as drivers of $FCO_2$ along the North-
South gradient is largely driven by increasing temperatures from North to South, especially in the
SAR region (Tab. SI 1).
Contrasting patterns across the 3 regions can also be observed with respect to carbon filtering
capacities, *CFilt* (Fig. 10b). In the NAR, over 90% of the riverine carbon flux is exported to the coastal
ocean. However, in the MAR, the high efficiency of the largest systems in processing organic carbon
results in a regional *CFilt* that exceeds 50%. This contrast between the NAR and the MAR and its
potential implication for the carbon dynamics of the adjacent continental shelf waters has already
been discussed by Laruelle et al. (2015). In the NAR, short estuarine residence results in a much
lower removal of riverine carbon by degassing compared to the MAR. Laruelle et al. (2015)
suggested that this process could contribute to the weaker continental shelf carbon sink adjacent to
the NAR, compared to the MAR. In the SAR, most estuaries remove between 40% and 65% of the
carbon inputs. The high temperatures observed and resulting accelerated biogeochemical process
rates in this region favor the degradation of organic matter and contribute to increase the estuarine
filtering capacity for carbon. However, in the SAR, a large fraction of the OC loads is derived from
adjacent salt marshes located along the estuarine salinity gradients, thereby reducing the overall
residence time of OC within the systems. The filtering capacity of the riverine OC alone, which
transits through the entire estuary, would thus be higher than the one calculated here. As a
consequence, highest C retention rates are expected in warm tidal estuaries devoid of salt marshes
or mangroves (Cai, 2011).
**3.2 Seasonal variability of estuarine carbon dynamics**
Carbon dynamics in estuaries of the US East coast not only show a marked spatial variability, but also
vary on the seasonal timescale. Table 5 presents the seasonal distribution of *NEM* and $FCO_2$ for each
sub-region. In the NAR, a strong seasonality is simulated for the *NEM* and the summer period
contributes more than a third to the annually integrated value. The outgassing reveals a lower
seasonal variability and is only slightly higher than summer outgassing during fall and lower during
spring. In the MAR, summer contributes more to the *NEM* (>28% of the yearly total) than any other
season, but seasonality is less pronounced than in the NAR. Here, $FCO_2$ is largest in winter and
particularly low during summer. In the SAR, summer accounts for 30 % of the NEM, while spring
contributes 21 %. $FCO_2$ is relatively constant throughout the year suggesting that seasonal variations
in carbon processing decrease towards the lower latitudes in the SAR. This is partly related to the
low variability in river discharge throughout the year in lower latitudes (Tab. SI1). In riverine
dominated systems with low residence times, such as, for instance, the Altamaha River estuary, the
$CO_2$ exchange at the air-water interface is mainly controlled by the river discharge because the time
required to degrade the entire riverine organic matter flux exceeds the transit time of OC through
the estuary. Therefore, the riverine sustained outgassing is highest during the spring peak discharge
periods. In contrast, the seasonal variability in $FCO_2$ in long-residence, marine-dominated systems
with large marsh areas (e.g. Sapelo and Doboy Sound) is essentially controlled by seasonal
temperature variations. Its maximum is reached during summer when marsh plants are dying and
decomposing, as opposed to spring when marshes are in their productive stage (Jiang et al., 2008).
These contrasting seasonal trends have already been reported for different estuarine systems in
Georgia, such as the Altamaha Sound, the Sapelo Sound and the Doboy Sound (Cai, 2011). At the
scale of the entire East coast of the US, the seasonal trends in *NEM* reveal a clear maximum in
summer and minimal values during autumn and winter. The seasonality of $FCO_2$ is much less
pronounced because the outgassing of oversaturated riverine waters throughout the year
contributes to a large fraction of the $FCO_2$ and dampens the effect of the temperature dependent
processes (*NEM* and denitrification). In our simulations, the competition between temperature and
river discharge is the main driver of the seasonal estuarine carbon dynamics is. When discharge
increases, the carbon loads increase proportionally and the residence time within the system
decreases, consequently limiting an efficient degradation of organic carbon input fluxes. In warm
regions like the SAR, the temperature is sufficiently high all year round to sustain high C processing
rates and this explains the reduced seasonal variability in NEM.

**3.3 Regional carbon budget: a comparative analysis**
The annual carbon budget for the entire East coast of the US is summarized in Fig. 11a. The total
carbon input to estuaries along the East coast of the US is 4.6 Tg C $y^{-1}$, of which 42% arrives in
organic form and 58% in inorganic form**.** Of this total input, saltmarshes contribute 0.6 Tg C $yr^{-1}$,
which corresponds to about 14% of the total carbon loads and 32% of the organic loads in the
region. The relative contribution of the saltmarshes to the total carbon input increases towards low
latitudes and is as high as 60% in the SAR region. Model results suggest that 2.7 Tg C $y^{-1}$ is exported
to the continental shelf (25% as TOC and 75% as DIC), while 1.9 Tg C $y^{-1}$ is emitted to the
atmosphere. The overall carbon filtering capacity of the region thus equals 41% of the total carbon
entering the 42 estuarine systems (river + saltmarshes). Because of the current lack of a benthic
module in C-GEM, the water column carbon removal occurs entirely in the form of $CO_2$ outgassing
and does not account for the potential contribution of carbon burial in sediments. The estimated
estuarine carbon retention presented here is thus likely a lower bound estimate. Reported to the
modeled surface area of the region, the total $FCO_2$ of 1.9 Tg C $y^{-1}$ translates into a mean air water
$CO_2$ flux of about 14 mol C $m^{-2}$ $y^{-1}$. This value is slightly higher than the estimate of 10.8 mol C $m^{-2}$ $y^{-1}$
calculated by Laruelle et al., (2013) on the basis of local $\overline{FCO_2}$ estimates assumed to be
representative of yearly averaged conditions (see section 2.1). The latter was calculated as the
average of 13 annual $\overline{FCO_2}$ values reported in the literature (Tab. 2), irrespective of the size of the
systems. This approach is useful and widely used to derive regional and global carbon budgets
(Borges et al., 2005; Laruelle et al., 2010; Chen et al., 2013). However, it may lead to potentially
significant errors (Volta et al., 2016a) due to the uncertainty introduced by the spatial interpolation
of local measurements to large regional surface areas, while useful and widely used to derive
regional and global carbon budgets.
Regional C budgets are sparse. To our knowledge, the only other published regional assessment of
the estuarine carbon and $CO_2$ dynamics comes from a relatively well studied region: the estuaries
flowing into the North Sea in Western Europe (Fig. 11b). This budget was calculated using a similar
approach (Volta 2016a) and thus provides an ideal opportunity for a comparative assessment of C
cycling in these regions. However, it is important to note that there are also important differences in
the applied model approaches and those differences should be taken into account when comparing
the derived budgets. In particular, the NW European study is based on a simulation of the 6 largest
systems only (Elbe, Scheldt, Thames, Ems, Humber and Weser), accounting for about 40% for the
riverine carbon loads of the region. It assumes that the intensity of carbon processing and evasion in
all other smaller estuaries discharging into the North Sea (16 % of the carbon loads) can be
represented by the average of the 6 largest system simulation results. In addition, the Rhine-Meuse
system, which alone accounts for 44% of the carbon riverine inputs of the region, was treated as a
passive conduit with respect to carbon due to its very short freshwater residence time (Abril et al.,
2002). The contribution of saltmarshes to the regional carbon budget was also ignored because their
total surface area is much smaller than along the US East coast (Regnier et al., 2013b). Another
important difference is the inclusion of seasonality in the present study while the budget calculated
for the North Sea is derived from yearly average conditions (Volta et al., 2016a).
Overall, although both regions receive similar amounts of C from rivers (4.6 Tg C y$^{-1}$ and 5.9 Tg C y$^{-1}$
for the East coast of the US and the North Sea, respectively), they reveal significantly different C
filtering capacities. While the estuaries of the East coast of the US filter 41% of the riverine TC loads,
those from the North Sea only remove 8% of the terrestrial-derived material. This is partly due to the
large amounts of carbon transiting through the 'passive' Rhine-Meuse system. The regional filtering
capacity is higher (15%) when this system is excluded from the analysis. However, even when
neglecting this system, significant differences in filtering efficiencies between both regions remain.
$FCO_2$ from the North Sea estuaries (0.5 Tg C $y^{-1}$) is significantly lower than the 1.9 Tg C $y^{-1}$ computed
for the East coast of the US. The reason for the lower evasion rate in NW European estuaries is
essentially twofold. First, the total cumulative surface area available for gas exchange is significantly
lower along the North Sea, in spite of comparable flux densities calculated using the entire estuarine
surface areas of both regions (14 mol C $m^{-2}$ $y^{-1}$ and 23 mol C $m^{-2}$ $y^{-1}$ for the East coast of the US and
the North Sea, respectively). Second, although the overall riverine carbon loads are comparable in
both regions (Fig. 11), the ratio of organic to inorganic matter input is much lower in the North Sea
area because of the regional lithology is dominated by carbonate rocks and mixed sediments that
contain carbonates (Dürr et al., 2005; Hartmann et al., 2012). As a consequence, TOC represents less
than 20% of the riverine loads and only 10% of the carbon exported to the North Sea. In both
regions, however, the increase of the inorganic to organic carbon ratio between input and output is
sustained by a negative NEM (Fig. 11). Although the ratios themselves may significantly vary from a
region of the world to the other as evidenced by these two studies, a NEM driven increase of the
inorganic fraction within carbon load along the estuarine axis is consistent with the global estuarine
carbon budget proposed by Bauer et al. (2013). In the East coast of the US, the respiration of riverine
OC within the estuarine filter is partly compensated by OC inputs from marshes and mangroves in
such a way that the input and export IC/OC ratios are closer than in the North Sea region.
**3.4 Scope of applicability and model limitations**
Complex multidimensional models are now increasingly applied to quantitatively explore carbon and
nutrient dynamics along the land-ocean transition zone over seasonal and even annual timescales
(Garnier et al., 2001; Arndt et al., 2007, 2009; Arndt and Regnier, 2007; Mateus et al., 2012).
However, the application of such complex models remains limited to individual, well-constrained
systems due their high data requirements and computational demand resulting from the need to
resolve important physical, biogeochemical and geological processes on relevant temporal and
spatial scales. The one-dimensional, computationally efficient model C-GEM has been specifically
designed to reduce data requirements and computational demand and to enable regional/global
scale applications (Volta et al., 2014, 2016a). However, such a low data demand and computational
efficiency inevitably requires simplification. The following paragraphs critically discuss these
simplifications and their implications.
*Spatial resolution*
Here, C-GEM is used with a 0.5° spatial resolution. While this resolution captures the features of
large systems, it is still very coarse for relatively small watersheds, such as those of the St. Francis
River, Piscataqua River, May River or the Sapelo River. For instance, the 5 estuaries reported by Hunt
et al. (2010, 2011, see section 2.6) are all small systems covered by the same watershed at a 0.5°
resolution. Only watersheds whose area spans several grid cells can be properly identified and
represented (i.e. Merrimack or Penobscot with 6 and 9 cells, respectively).

*Hydrodynamic and Transport Model*
C-GEM is based on a theoretical framework that uses idealized geometries and significantly reduces
data requirements. These idealized geometries are fully described by three, easily obtainable
geometrical parameters (B, B0, H). The model thus approximates the variability of estuarine width
and cross-section along the longitudinal axis through a set of exponential functions. A
comprehensive sensitivity study (Volta et al., 2014) has shown that integrated process rates are
generally sensitive to changes in these geometrical parameters because of their control on estuarine
residence times. For instance, Volta et al. (2014) demonstrated that the NEM, is particularly sensitive
to the convergence length. Similarly, the use of constant depth profile may lead to variations of
about 10% in NEM (Volta et al., 2014). Nevertheless, geometrical parameters are generally easy to
constrain, especially well-monitored regions such as the US east coast. Here, all geometrical
parameters are constrained on the basis of observed estuarine surface areas and average water
depths. In addition, the model also accounts for the slope of the estuarine channel. This approach
ensures that simulated estuarine surface areas, volumes and, thus, residence times are in good
agreement with those of the real systems and minimizes uncertainties associated to the physical set-
up.
In addition, the one-dimensional representation of the idealized estuarine systems does not resolve
two- or three-dimensional circulation features induced by complex topography and density driven
circulation. While C-GEM performs well in representing the dominant longitudinal gradients, its
applicability to branched systems or those with aspect ratios for which a dominant axis is difficult to
identify (e.g. Blackwater estuary, UK; Pearl River estuary, China; Tagus estuary, Portugal; Bay of
Brest, France) is limited.
*Biogeochemical Model*
Although the reaction network of C-GEM accounts for all processes that control estuarine $FCO_2$
(Borges and Abril, 2012; Cai, 2011), several, potentially important processes, such as benthic-pelagic
exchange processes, phosphorous sorption/desorption and mineral precipitation, a more complex
representation of the local phytoplankton community, grazing by higher trophic levels, or multiple
reactive organic carbon pools are not included. Although these processes are difficult to constrain
and their importance for $FCO_2$ is uncertain, the lack of their explicit representations induces
uncertainties in Cfilt. In particular, the exclusion of benthic processes such as organic matter
degradation and burial in estuarine sediments could result in an underestimation of Cfilt. However,
because very little is known on the long term fate of organic carbon in estuarine sediments, setting
up and calibrating a benthic module proves a difficult task. Furthermore, to a certain degree model
parameters (such as organic matter degradation and denitrification rate constant) implicitly account
for benthic dynamics. We nonetheless acknowledge that, by ignoring benthic processes and burial in
particular, our estimates for the estuarine carbon filtering may be underestimated, particularly in
the shallow systems of the SAR.
Biogeochemical model parameters for regional and global applications are notoriously difficult to
constrain (Volta et al., 2016b). Model parameters implicitly account for processes that are not
explicitly resolved and their transferability between systems is thus limited. In addition, published
parameter values are generally biased towards temperate regions in industrialized countries (Volta
et al., 2016b). A first order estimation of the parameter uncertainty associated to the estuarine
carbon removal efficiency (CFilt) can be extrapolated from the extensive parameter sensitivity
analyses carried out by Volta et al. (2014, 2016b). These comprehensive sensitivity studies on end-
member systems have shown that the relative variation in Cfilt when a number of key
biogeochemical parameters are varied by two orders of magnitude varies by ±15 % in prismatic
(short residence time on order of days) to ±25 % in funnel-shaped (long residence time) systems.
Thus, assuming that uncertainty increases linearly between those bounds as a function of residence
time, an uncertainty estimate can be obtained for each of our modelled estuary. With this simple
method, the simulated regional Cfilt of 1.9 Tg C yr-1 would be associated with an uncertainty range
comprised between 1.5 and 2.2 Tg C yr$^{-1}$. Our regional estuarine $CO_2$ evasion estimate is thus
reported with moderate confidence. Furthermore, in the future, this uncertainty range could be
further constrained using statistical methods such as Monte Carlo simulations (e.g. Lauerwald et al.,

783    2015).

*Boundary Conditions and Forcings*
In addition, simulations are only performed for climatological means over the period 1990-2010
without resolving interannual and secular variability. Boundary conditions and forcings are critical as
they place the modelled system in its environmental context and drive transient dynamics. However,
for regional applications, temporally resolved boundary conditions and forcings are difficult to
constrain. C-GEM places the lower boundary condition 20 km from the estuarine mouth into the
coastal ocean and the influence of this boundary condition on simulated biogeochemical dynamics is
thus limited. At the lower boundary condition, direct observations for nutrients and oxygen are
extracted from databases such as the World Ocean Atlas (Antonov et al., 2010). However, lower
boundary conditions for OC and $pCO_2$ (zero concentration for OC and assumption of $pCO_2$
equilibrium at the sea side) are simplified. This approach does not allow addressing the additional
complexity introduced by biogeochemical dynamics in the estuarine plume (see Arndt et al., 2011).
Yet, these dynamics only play a secondary role in the presented study that focuses on the role of the
estuarine transition zone in processing terrestrial-derived carbon.
Constraining upper boundary conditions and forcings is thus more critical.  Here, C-GEM is forced by
seasonally-averaged conditions for Q, T, and radiation. To date, GlobalNEWS only provide yearly-
averaged conditions for a number of upper boundary conditions (Seitzinger et al., 2005; Mayorga et
al., 2010), representative of the year 2000. Simulations are thus only partly transient (induced by
seasonality in Q, T and radiation) and do not resolve short-lived events such as storms or extreme
drought conditions. In addition, direct observations of upper boundary conditions are rarely
available, in particular over seasonal or annual timescales. For the US East Coast estuaries, direct
observations are only available for $O_2$, Chlorophyll-a, DIC and Alk. For DIC and alkalinity boundary
conditions are constrained by calculating the average concentration over a period of about three
decades. In addition, observational data are extracted at the station closest to the model's upper
boundary, which might be still located several kilometres upstream or downstream of the model
boundary. Upper boundary conditions of POC, DOC, DIN, DIP, DSi are extracted from GlobalNews
and thus model-derived. As a consequence, our results are thus intimately dependent on the
robustness of the GlobalNEWS predictions. These values are usually only considered robust
estimates for watersheds larger than ~10 cells (Beusen et al., 2005), which only correspond to 13 of
the 42 estuaries modelled in this study.
*Model-data comparison*
The generic nature of the applied model approach renders a direct validation of model results on the
basis of local and instantaneous observational data (e.g. longitudinal profiles) difficult. In particular
the applications of seasonally/annually averaged or model-deduced boundary conditions, which are
likely not representative of these long-term average conditions, do not lend themselves well to
comparison with punctual measurements. Therefore, model performance is evaluated on the basis
of spatially aggregated estimates (e.g. regional $FCO_2$ estimates based on local measurements) rather
than system-to-system comparisons with longitudinal profile from specific days. However, note that
the performance of C-GEM has been intensively tested by specific model-data comparisons for a
number of different systems (e.g. Volta et al., 2014, 2016a) and we are thus confident of its
predictive capabilities.
Despite the numerous simplifying assumptions inevitably required for such a regional assessment of
carbon fluxes along the land-ocean continuum, the presented approach does nevertheless provide
an important step forward in evaluating the role of land-ocean transition systems in the global
carbon cycle. It provides a first robust estimate of carbon dynamics based on a theoretically well-
founded and carefully tested, spatially and temporally resolved model approach. This approach
provides novel insights that go beyond those gained through traditionally applied zero-salinity
method or box model approaches. In addition, it also highlights critical variables and data gaps and
thus helps guide efficient monitoring strategies.
**3.5 Towards predictors of the estuarine carbon processing**
The mutual dependence between geometry and transport in tidal estuaries and, ultimately, their
biogeochemical functioning (Savenije, 1992; Volta et al., 2014) allows relating easily extractable
parameters linked to their shape or their hydraulic properties to biogeochemical indicators. In this
section, we explore the relationships between such simple physical parameters and indicators of the
estuarine carbon processing $\overline{NEM}$, $\overline{FCO_2}$ and *CFilt*. In order to account for the effect of temperature
on C dynamics, -$\overline{NEM}$ and $\overline{FCO_2}$ are also normalized to the same temperature (arbitrarily chosen to
be 0 degree). These normalized values are obtained by dividing -$\overline{NEM}$ and $\overline{FCO_2}$ by a $Q_{10}$ function
*f(T)* (see Volta et al., 2014). This procedure allows accounting for the exponential increase in the rate
of several temperature dependent processes contributing to the NEM (i.e. photosynthesis, organic
carbon degradation…). Applying the same normalization to $-\overline{NEM}$ and $\overline{FCO_2}$ is a way of testing how
intimately linked *NEM* and *FCO$_2$* are in estuarine systems. Indeed linear relationships relating one to
the other have been reported (Mayer and Eyre, 2012). The three indicators are then investigated as
a function of the ratio between the estuarine surface *S* and the seasonal river discharge *Q*. The
surface area is calculated from the estuarine width and length, as described by equation 2, in order
to use a parameter which is potentially applicable to other regions for which direct estimates of the
real estuarine surface area is not available. Since the fresh water residence time of a system is
obtained by dividing volume by river discharge, the S/Q ratio is also intimately linked to residence
time. Here, we choose to exclude the estuarine depth from the analysis because this variable cannot
be easily quantified from maps or remote sensing images and would thus compromise the
applicability of a predictive relationship on the global scale. However, from dimensional analysis, S/Q
can be viewed as a water residence time normalized to meter depth of water. As shown by equation
3, *S* only requires constraining *B0* and width convergence length *b*, two parameters that can readily
be extracted from the Google Earth engine. Global database of river discharges, as for instance
RivDIS (Vörösmarty et al., 1996) are also available in such a way that the S/Q ratio can potentially be
extracted for all estuaries around the globe.
Figure 12a reveals that small values of S/Q are associated with the most negative $\overline{NEM}$ / $f(T)$. The
magnitude of the $\overline{NEM}$ then exponentially decreases with increasing values of S/Q. Estuaries
characterized by small values of *S/Q* are mainly located in the NAR sub-region and correspond to
small surface area, and thus short residence time systems. It is possible to quantitatively relate -
$\overline{NEM}$ /$f(T)$ and S/Q through a power law function (y = 25.85 x$^{-0.64}$ with a r$^2$ = 0.82). The coefficient
of determination remains the same when excluding estuaries from the NAR region and the equation
itself is not significantly different, although those estuaries on their own do not display any
statistically significant trend (Tab. 6). The decrease in the intensity of the net ecosystem metabolism
in larger estuaries (Fig 8), characterized by high S/Q ratios, can be related to the extensive
consumption of the organic matter pool during its transit through the estuarine filter. However,
when reported to the entire surface area of the estuary, larger systems (with high values of S/Q) still
reveal the most negative surface integrated *NEM* (Fig. 12b). It can also be noted that some estuaries
from the NAR region display very low values of –*NEM*. These data points correspond to fall and
winter simulations for which the temperature was relatively cold (<5 °C) and biogeochemical
processing was very low.
The overall response of $\overline{FCO_2}/f(T)$ to S/Q is comparable to that of $-\overline{NEM}/f(T)$ (Fig. 12c), with
lower values of $\overline{FCO_2}$ observed for high values of S/Q. However, for S/Q < 3 days m$^{-1}$, the $\overline{FCO_2}$
values are very heterogeneous and contain many, low $\overline{FCO_2}$ outliers from the NAR region. These
data points generally correspond to low water temperature conditions which keep $pCO_2$ low, even if
the system generates enough $CO_2$ internally via NEM. Thus, the well-documented correlation
between $\overline{NEM}$ and $\overline{FCO_2}$ (Maher and Eyre, 2012) does not seem to hold for systems with very short
residence times. For systems with S/Q > 3 days m$^{-1}$, we obtain a regression *FCO$_2$* = -0.64 x *NEM* + 5.96
with a r$^2$ of 0.46, which compares well with the relation *FCO$_2$* = -0.42 x *NEM* + 12 proposed by Maher
and Eyre (2012) who used 24 seasonal estimates from small Australian estuaries. However, our
results suggest that this relationship cannot be extrapolated to small systems such as those located
in the NAR. Figure 12d, which reports non-normalized *FCO$_2$* reveals a monotonous increase of *FCO$_2$*
with S/Q. This suggests that, unlike the *NEM* for which the normalization by a temperature function
allowed explaining most of the variability; *FCO$_2$* is mostly controlled by the water residence time
within the system. Discharge is the main $FCO_2$ driver in riverine dominated systems, while
interactions with marshes are driving the outgassing in marine dominated systems surrounded by
marshes. Net aquatic biological production (NEM being negative or near 0) in large estuaries (with
large S/Q) is another important reason for low $FCO_2$ in such systems. For example, despite the higher
$CO_2$ degassing flux in the upper estuary of the Delaware, strong biological $CO_2$ uptake in the mid-bay
and near zero NEM in the lower bay result in a much lower $FCO_2$ for the entire estuary (Joesoef et al.
2015).   In systems with S/Q < 3 days m$^{-1}$, the short residence time prevents the excess $CO_2$ of
oversaturated water from being entirely exchanged with the atmosphere and simulations reveal that
the estuarine waters are still oversaturated in $CO_2$ at the estuarine mouth. Thus, the inorganic
carbon, produced by the decomposition of organic matter, is not outgassed within the estuary but
exported to the adjacent continental shelf waters. This result is consistent with the observation-
based hypothesis of Laruelle et al. (2015) for the NAR estuaries. As a consequence of the distinct
behavior of short residence time systems, the coefficient of determination of the best-fitted power
law function relating $\overline{FCO_2}$ and S/Q is only significant if NAR systems are excluded (y = 31.64 x$^{-0.58}$
with a r$^2$ = 0.70). This thus suggests that such relationships (as well as that proposed by Maher and
Eyre, 2012) cannot be applied to any system but only those for which S/Q>3 day m$^{-1}$.
Finally, Fig. 12e reports the simulated mean seasonal carbon filtering capacities as a function of the
depth normalized residence time. Not surprisingly, and in overall agreement with previous studies
on nutrient dynamics in estuaries (Nixon et al., 1996), the carbon filtering capacity increases with
S/Q. The best statistical relation between *CFilt* and S/Q is obtained when including all 3 regions,
resulting in r$^2$ = 0.70 (y = 40.64 log$_{10}$(x) + 11.84). Very little C removal occurs in systems with S/Q < 1
day m$^{-1}$. For systems characterized by longer depth-normalized residence times, *CFilt* increases
regularly, and reaches 100% for S/Q > 100 day m$^{-1}$. Such high values are only observed for very large
estuaries from the MAR region (Delaware and Chesapeake Bays); the majority of our systems had an
*S/Q* range between 1 and 100 day m$^{-1}$.  The quantitative assessment of estuarine filtering capacities
is further complicated by the complex interplay of estuarine and coastal processes. Episodically,
marked spatial variability in concentration gradients near the estuarine mouth may lead to a reversal
of net material fluxes from coastal waters into the estuary (Regnier at al., 1998; Arndt et al. 2011).
Our results show that this feature is particularly significant for estuaries with a large width at the
mouth and short convergence length (funnel shaped or 'Bay type' systems). These coastal nutrient
and carbon inputs influence the internal estuarine C dynamics and lead to filtering capacities that
can exceed 100%. This feature is particularly significant in summer, when riverine inputs are low and
the marine material is intensively processed inside the estuary.
Previous work investigated the relationship between fresh water residence time and nutrient
retention (Nixon et al., 1996; Arndt et al., 2011; Laruelle, 2009). These studies, however, were
constrained by the scarcity of data. For instance, the pioneering work of Nixon et al. (1996) only
relied on a very limited number (<10) of quite heterogeneous coastal systems, all located along the
North Atlantic. Here, our modeling approach allows us to generate 168 (42 x 4) data points, each
representing a system-scale biogeochemical behavior. Together, this database spans the entire
spectrum of estuarine settings and climatic conditions found along the East coast of the US. In
addition, the ratio S/Q used as master variable for predicting temperature normalized -$\overline{NEM}$, $\overline{FCO_2}$
and *CFilt* only requires a few easily accessible geometric parameters (B0, b and L) and an estimate of
the river discharge. While it is difficult to accurately predict $\overline{FCO_2}$ for small systems such as those
located in the NAR region, the relationships found are quite robust for systems in which S/Q > 3 days
m$^{-1}$. Most interestingly, *CFilt* values reveal a significant correlation with S/Q and could be used in
combination with global riverine carbon delivery estimates such as GlobalNews 2 (Mayorga et al.,
2010) to constrain the estuarine $CO_2$ evasion and the carbon export to the coastal ocean at the
continental and global scales.
**4. Conclusions**
This study presents the first complete estuarine carbon budget for the East coast of the US using a
modeling approach. The structure of the model C-GEM relies on a restricted number of readily
available global datasets to constrain boundary conditions and limits the number of geometrical and
physical parameters to be constrained. Our simulations predict a total $CO_2$ outgassing of 1.9 Tg C y$^{-1}$
for all tidal estuaries of the East coast of the US. This quantification accounts for the seasonality in
estuarine carbon processing as well as for distinct individual behaviors among estuarine types
(marine or river dominated). The total carbon output to the coastal ocean is estimated at 2.7 TgC y$^{-1}$,
and the carbon filtering capacity with respect to riverine, marshes and mangrove inputs is thus on
the order of 40%. This value is significantly higher than the recently estimated C filtering capacity for
estuaries surrounding the North Sea using a similar approach (Volta et al., 2016a), mainly because
the surface area available for gas exchange and the draining lithology limits the $CO_2$ evasion in the
NW European systems. At the regional scale of the US East coast estuaries, net heterotrophy is the
main driver (50%) of the $CO_2$ outgassing, followed by the ventilation of riverine supersaturated
waters entering the estuarine systems (32%) and nitrification (18%). The dominant mechanisms for
the gas exchange and the resulting carbon filtering capacities nevertheless reveal a clear latitudinal
pattern, which reflects the shapes of estuarine systems, climatic conditions and dominant land-use
characteristics.
Our model results are used to derive predictive relationships relating the intensity of the area-based
Net Ecosystem Metabolism ($\overline{NEM}$), air-water $CO_2$ exchange ($\overline{FCO_2}$) and the carbon filtering capacity
(*CFilt*) to the depth normalized residence time, expressed as the ratio of the estuarine surface area
to the river discharge. In the future, such simple relationships relying on readily available geometric
and hydraulic parameters could be used to quantify carbon processing in areas of the world devoid
of direct measurements. However, it is important to note that such simple relationships are only
valid over the range of boundary conditions and forcings explored and may not be applicable to
conditions that fall outside of this range. In regions with better data coverage, such as the one
investigated here, our study highlights that the regional-scale quantification, attribution, and
projection of estuarine biogeochemical cycling are now at reach.
**5. Acknowledgements**
G. G. Laruelle is Chargé de recherches du F.R.S.-FNRS at the Université Libre de Bruxelles. The
research leading to these results has received funding from the European Union's Horizon 2020
research and innovation programme under the Marie Sklodowska-Curie grant agreement No 643052
(C-CASCADES project). The authors thank V. L. Mulder for her thorough reading of the manuscript
upon submission.

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

**Table 1:** Estimates of total annual riverine input from watersheds to estuaries (Tg C yr$^{-1}$). The ranges
are based on Stets and Striegl (2012), Global NEWS (Mayorga et al. 2010), Hartmann et al. (2009),
SPARROW (Shih et al. 2010) and DLEM (Tian et al. 2010, 2012). Modified from Najjar et al. 2012.

| | DIC | DOC | POC | TOTAL |
|---|---|---|---|---|
| NAR | 0.2-0.8 | 0.3-2.1 | 0.1-0.2 | 0.6-3.1 |
| MAR | 1.4-1.8 | 0.5-2.3 | 0.1-0.3 | 2.0-4.4 |
| SAR | 0.4-1.4 | 0.9-1.6 | 0.1-0.2 | 1.4-3.2 |
| TOTAL | 2.0-4.0 | 1.7-6.0 | 0.3-0.7 | 4.0-10.7 |




**Table 2**: Published local annually averaged estimates of $\overline{FCO_2}$ in mol C m$^{-2}$ yr$^{-1}$ for estuaries along the
East coast of the US.**"**

| Name | Lon | Lat | $\overline{FCO_2}$ Observed. | Modeled | Reference |
|------|-----|-----|-----------|---------|-----------|
| Altamaha Sound | -81.3 | 31.3 | 32.4 | 72.7 | Jiang et al. (2008) |
| Bellamy | -70.9 | 43.2 | 3.6 | 3.9 | Hunt et al. (2010) |
| Cocheco | -70.9 | 43.2 | 3.1 | 3.9 | Hunt et al. (2010) |
| Doboy Sound | -81.3 | 31.4 | 13.9 | 25.7 | Jiang et al. (2008) |
| Great Bay | -70.9 | 43.1 | 3.6 | 3.9 | Hunt et al. (2011) |
| Little Bay | -70.9 | 43.1 | 2.4 | 3.9 | Hunt et al. (2011) |
| Oyster Bay | -70.9 | 43.1 | 4 | 3.9 | Hunt et al. (2011) |
| Parker River estuary | -70.8 | 42.8 | 1.1 | 3.9 | Raymond and Hopkinson (2003) |
| Sapelo Sound | -81.3 | 31.6 | 13.5 | 20.6 | Jiang et al. (2008) |
| Satilla River | -81.5 | 31 | 42.5 | 25.7 | Cai and Wang (1998) |
| York River | -76.4 | 37.2 | 6.2 | 8.1 | Raymond et al. (2000) |
| Hudson River | -74 | 40.6 | 13.5 | 15.5 | Raymond et al. (1997) |



**Table 3:** State variables and processes explicitly implemented in CGEM.

| State variables | | |
| --- | --- | --- |
| **Name** | **Symbol** | **Unit** |
| Suspended Particulate Mater | SPM | $gL^{-1}$ |
| Total Organic Carbon | TOC | µM C |
| Nitrate | $NO_3$ | µM N |
| Ammonium | $NH_4$ | µM N |
| Phosphate | DIP | µM P |
| Dissolved Oxygen | DO | µM $O_2$ |
| Phytoplankton | Phy | µM C |
| Dissolved Silica | dSi | µM Si |
| Dissolved Inorganic Carbon | DIC | µM C |
| **Biogeochemical reactions** | | |
| **Name** | **Symbol** | **Unit** |
| Gross primary production | GPP | µM C $s^{-1}$ |
| Net primary production | NPP | µM C $s^{-1}$ |
| Phytoplankton mortality | M | µM C $s^{-1}$ |
| Aerobic degradation | R | µM C $s^{-1}$ |
| Denitrification | D | µM C $s^{-1}$ |
| Nitrification | N | µM N $s^{-1}$ |
| $O_2$ exchange with the atmosphere | $FO_2$ | µM $O_2$ $s^{-1}$ |
| $CO_2$ exchange with the atmosphere | $FCO_2$ | µM C $s^{-1}$ |
| SPM erosion | $E_{SPM}$ | $gL^{-1}$ $s^{-1}$ |
| SPM deposition | $D_{SPM}$ | $gL^{-1}$ $s^{-1}$ |



**Table 4:** Yearly averaged surface area (*S*), fresh water discharge (*Q*), residence time (*Rt*), *FCO₂* and *NEM* of all simulated estuaries.

| long degrees | lat degrees | S km² | Q m³s⁻¹ | Rt days | $\overline{FCO_2}$ mol C m⁻² yr⁻¹ | $\overline{NEM}$ mol C m⁻² yr⁻¹ | FCO₂ 10⁶ mol C yr⁻¹ | NEM 10⁶ mol C yr⁻¹ |
|---|---|---|---|---|---|---|---|---|
| **NAR** | | | | | | | | |
| -67.25 | 44.75 | 7 | 38.5 | 15 | 3.7 | -37.4 | 27 | -270 |
| -67.25 | 45.25 | 12 | 73.6 | 15 | 6.0 | -56.7 | 71 | -666 |
| -67.25 | 45.25 | 12 | 73.6 | 15 | 13.8 | -56.6 | 162 | -666 |
| -67.75 | 44.75 | 3 | 68.5 | 4 | 6.7 | -63.5 | 23 | -221 |
| -68.25 | 44.75 | 14 | 69.5 | 19 | 4.1 | -56.2 | 58 | -791 |
| -68.75 | 44.75 | 89 | 309.9 | 23 | 27.4 | -58.2 | 2431 | -5163 |
| -69.75 | 44.25 | 50 | 626.6 | 5 | 32.3 | -74.4 | 1607 | -3703 |
| -70.25 | 43.75 | 3 | 25.8 | 10 | 2.1 | -21.0 | 7 | -71 |
| -70.75 | 41.75 | 288 | 103.6 | 958 | 5.0 | -4.0 | 1428 | -1146 |
| -70.75 | 42.25 | 63 | 210.7 | 40 | 16.2 | -32.9 | 1025 | -2081 |
| -70.75 | 42.75 | 17 | 105.8 | 3 | 56.3 | -69.0 | 943 | -1155 |
| **MAR** | | | | | | | | |
| -70.75 | 43.25 | 31 | 29.9 | 11 | 21.6 | -37.4 | 662 | -1146 |
| -71.25 | 41.75 | 257 | 28.2 | 808 | 3.9 | -2.5 | 997 | -650 |
| -71.75 | 41.25 | 21 | 112.4 | 4 | 35.2 | -32.6 | 726 | -672 |
| -72.75 | 40.75 | 20 | 25.4 | 62 | 30.7 | -21.1 | 623 | -430 |
| -72.75 | 41.25 | 10 | 142.5 | 2 | 150.8 | -36.9 | 1578 | -386 |
| -72.75 | 41.75 | 55 | 476.6 | 3 | 55.9 | -45.7 | 3088 | -2523 |
| -73.25 | 40.75 | 19 | 26.8 | 56 | 31.4 | -28.4 | 608 | -550 |
| -74.25 | 40.75 | 1192 | 608.2 | 126 | 15.5 | -11.8 | 18432 | -14047 |
| -75.25 | 38.25 | 399 | 80.5 | 172 | 13.9 | -5.0 | 5558 | -2016 |
| -75.25 | 38.75 | 354 | 31.8 | 357 | 7.5 | -3.0 | 2659 | -1076 |
| -75.25 | 39.75 | 1716 | 499.0 | 221 | 10.0 | -7.8 | 17072 | -13439 |
| -75.75 | 39.25 | 224 | 18.3 | 434 | 7.5 | -2.9 | 1685 | -640 |
| -76.25 | 39.25 | 3427 | 717.1 | 352 | 8.1 | -5.1 | 27646 | -17352 |
| -76.75 | 37.25 | 586 | 272.3 | 74 | 15.0 | -10.4 | 8810 | -6084 |
| -76.75 | 37.75 | 154 | 36.3 | 163 | 10.7 | -6.6 | 1654 | -1023 |
| -76.75 | 39.25 | 59 | 71.2 | 29 | 48.6 | -34.6 | 2862 | -2038 |
| -77.25 | 38.25 | 206 | 30.2 | 268 | 6.1 | -3.3 | 1265 | -676 |
| -77.25 | 38.75 | 568 | 259.2 | 118 | 16.7 | -10.8 | 9488 | -6134 |
| **SAR** | | | | | | | | |
| -78.25 | 34.25 | 48 | 167.4 | 7 | 122.5 | -62.4 | 5916 | -3015 |
| -79.25 | 33.25 | 47 | 56.3 | 42 | 43.4 | -36.5 | 2056 | -1728 |
| -79.25 | 33.75 | 45 | 291.4 | 8 | 85.1 | -78.7 | 3843 | -3551 |
| -79.75 | 33.25 | 25 | 33.8 | 15 | 37.9 | -32.8 | 956 | -828 |
| -80.25 | 32.75 | 25 | 31.0 | 50 | 48.8 | -42.5 | 1214 | -1057 |
| -80.25 | 33.25 | 92 | 75.5 | 61 | 62.7 | -61.2 | 5769 | -5625 |
| -80.75 | 32.25 | 71 | 21.1 | 182 | 12.9 | -7.0 | 918 | -501 |
| -80.75 | 32.75 | 164 | 63.1 | 95 | 20.6 | -11.5 | 3372 | -1879 |
| -81.25 | 31.75 | 92 | 71.7 | 45 | 25.7 | -20.9 | 2361 | -1926 |
| -81.25 | 32.25 | 130 | 379.8 | 11 | 51.7 | -39.2 | 6732 | -5097 |
| -81.75 | 30.75 | 34 | 18.7 | 61 | 17.5 | -14.7 | 602 | -505 |
| -81.75 | 31.25 | 130 | 17.7 | 294 | 5.5 | -4.0 | 713 | -523 |
| -81.75 | 31.75 | 56 | 350.5 | 4 | 72.7 | -67.4 | 4068 | -3770 |

**Table 5:** Seasonal contribution to $FCO_2$ and *NEM* in each the sub-region. The seasons displaying the
highest percentages are indicated in bold. Winter is defined as January, February and March, Spring
as April, May and June and so on…

| Region | $S$ km$^2$ | *NEM* mol C y$^{-1}$ | winter % | spring % | summer % | fall % | $FCO_2$ mol C y$^{-1}$ | winter % | spring % | summer % | fall % |
|--------|-----|----------------------|------|------|--------|------|------------------------|------|------|--------|------|
| NAR | 558 | -16.3 10$^9$ | 14.7 | 21.2 | **37.0** | 27.2 | 7.2 10$^9$ | 26.3 | 18.9 | 26.5 | **28.3** |
| MAR | 9298 | -72.2 10$^9$ | 21.9 | 25.9 | **28.8** | 23.4 | 108.3 10$^9$ | **29.8** | 23.3 | 20.7 | 26.2 |
| SAR | 959 | -30.5 10$^9$ | 24.6 | 20.9 | **30.3** | 24.2 | 39.2 10$^9$ | 26 | 23.4 | **27** | 23.6 |



**Table 6:** Regressions and associated coefficient of determination between the depth normalized
residence time (S/Q) and $-\overline{NEM} / f(T)$, $\overline{FCO_2} / f(T)$ and *CFilt*.

| Region | $-\overline{NEM}/f(T)$ | $\overline{FCO_2}/f(T)$ | *CFilt* |
|---|---|---|---|
| **NAR** | $y = 27.84\ x^{-0.17}$ | $y = 6.07\ x^{0.00}$ | $y = 15.08\ \log_{10}(x) + 4.86$ |
|  | $r^2 = 0.11$ | $r^2 = 0.00$ | $r^2 = 0.40$ |
| **MAR** | $y = 26.03\ x^{-0.63}$ | $y = 34.36\ x^{-0.58}$ | $y = 40.46\ \log_{10}(x) + 9.60$ |
|  | $r^2 = 0.86$ | $r^2 = 0.68$ | $r^2 = 0.70$ |
| **SAR** | $y = 28.36\ x^{-0.71}$ | $y = 32.82\ x^{-0.66}$ | $y = 23.19\ \log_{10}(x) + 43.71$ |
|  | $r^2 = 0.76$ | $r^2 = 0.80$ | $r^2 = 0.46$ |
| **MAR + SAR** | $y = 25.85\ x^{-0.64}$ | $y = 31.64\ x^{-0.58}$ | $y = 33.30\ \log_{10}(x) + 24.88$ |
|  | $r^2 = 0.82$ | $r^2 = 0.70$ | $r^2 = 0.57$ |
| **NAR + MAR + SAR** | $y = 28.98\ x^{-0.66}$ | $y = 12.98\ x^{-0.33}$ | $y = 40.64\ \log_{10}(x) + 11.84$ |
|  | $r^2 = 0.82$ | $r^2 = 0.30$ | $r^2 = 0.70$ |



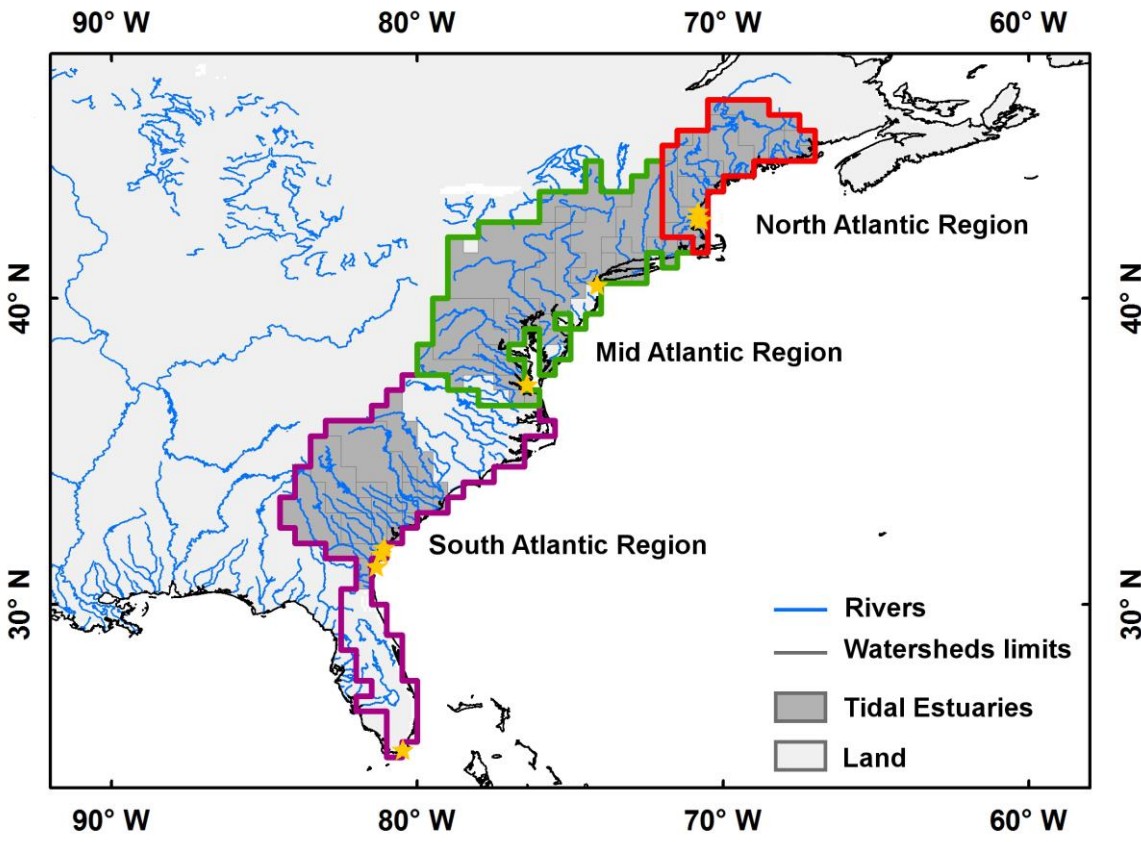

**Figure 1**: Limits of the 0.5 degrees resolution watersheds corresponding to tidal estuaries of the East coast of the US. 3 sub-regions are delimited with colors and orange stars represent the location of previous studies.

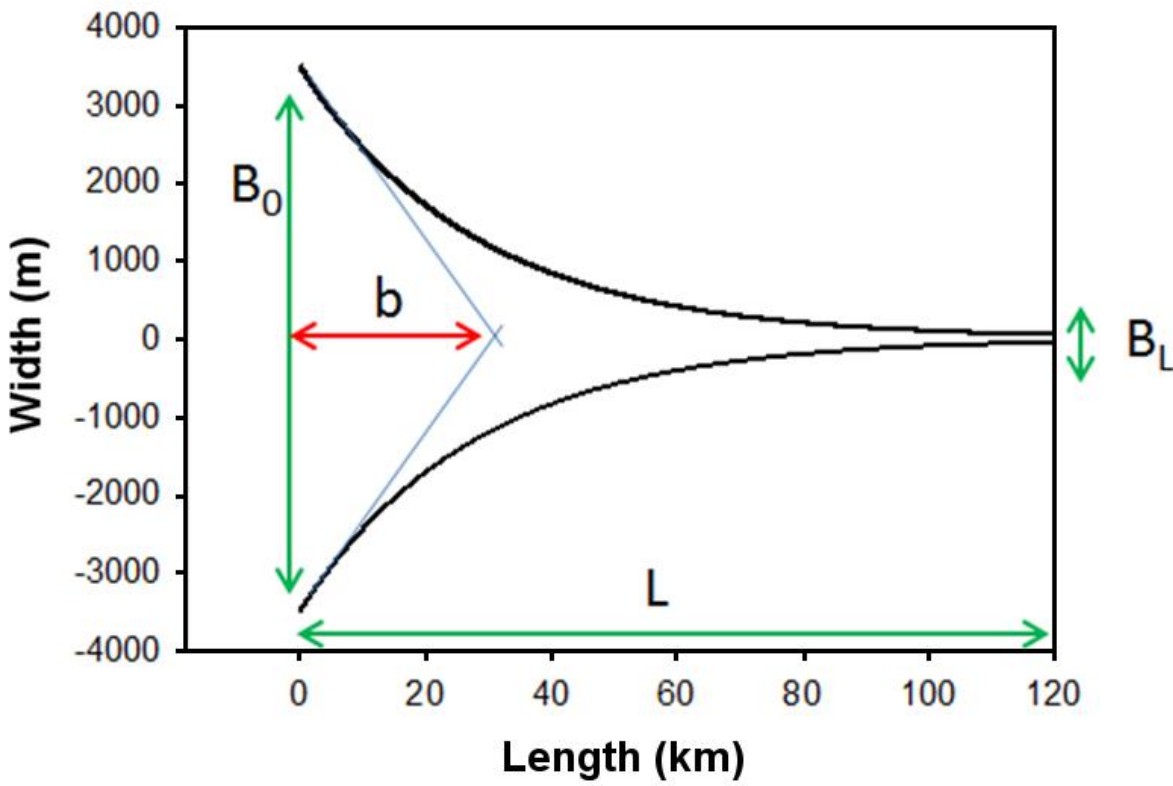


**Figure 2:** Idealized estuarine geometry and main parameters. Parameters indicated by green arrows
are measured, b is calculated. See section 2.3.1 for further details.


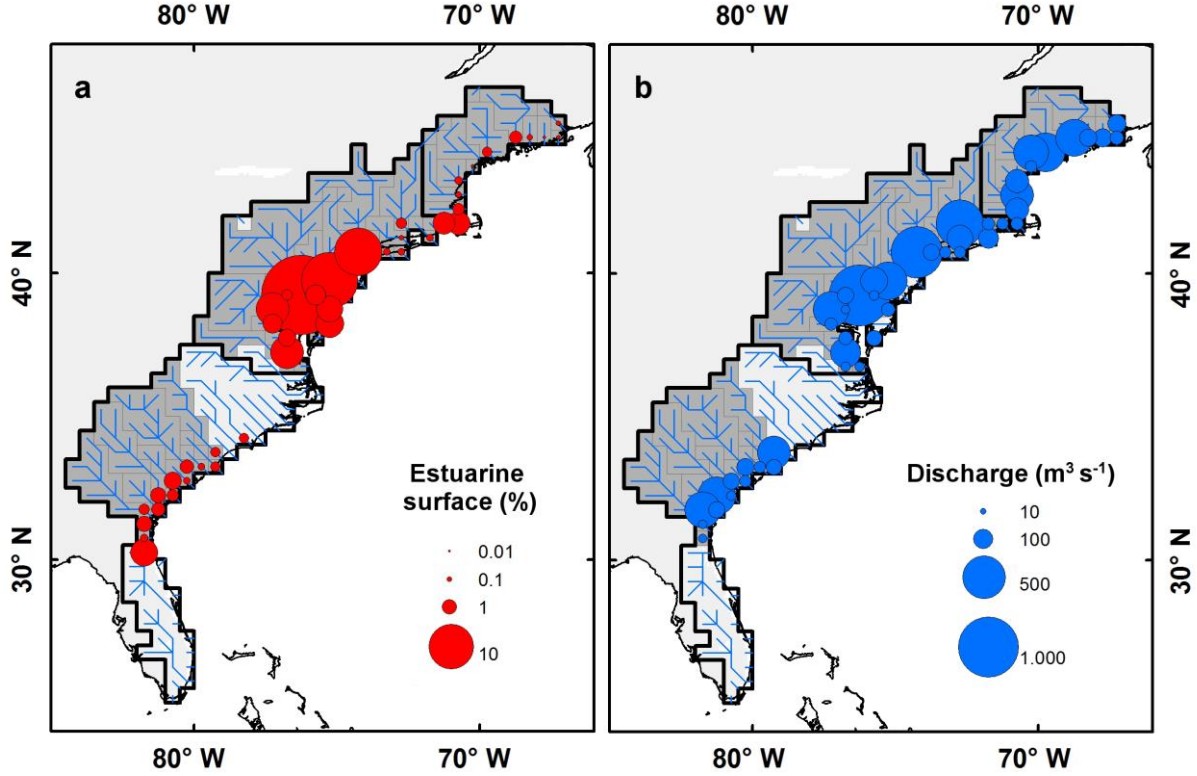


**Figure 3:** Estuarine surface area (a) and mean annual freshwater discharge (b) for each tidal estuary
of the East coast of the US. Estuarine surface area are expressed as percentage of the entire surface
area of the region (19830 km$^2$)


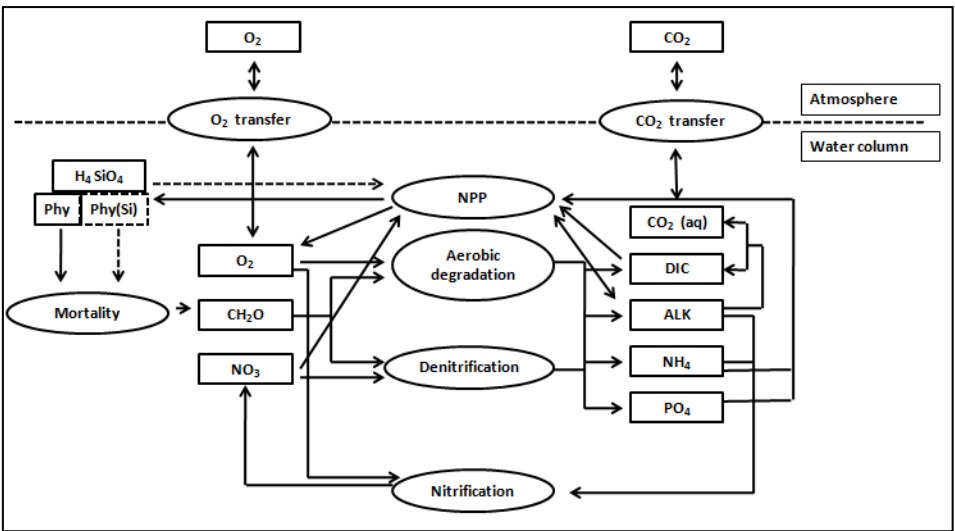

**Figure 4:** Conceptual scheme of the biogeochemical module of C-GEM used in this study. State-variables and processes are represented by boxes and oval shapes, respectively. Modified from Volta et al., 2014.


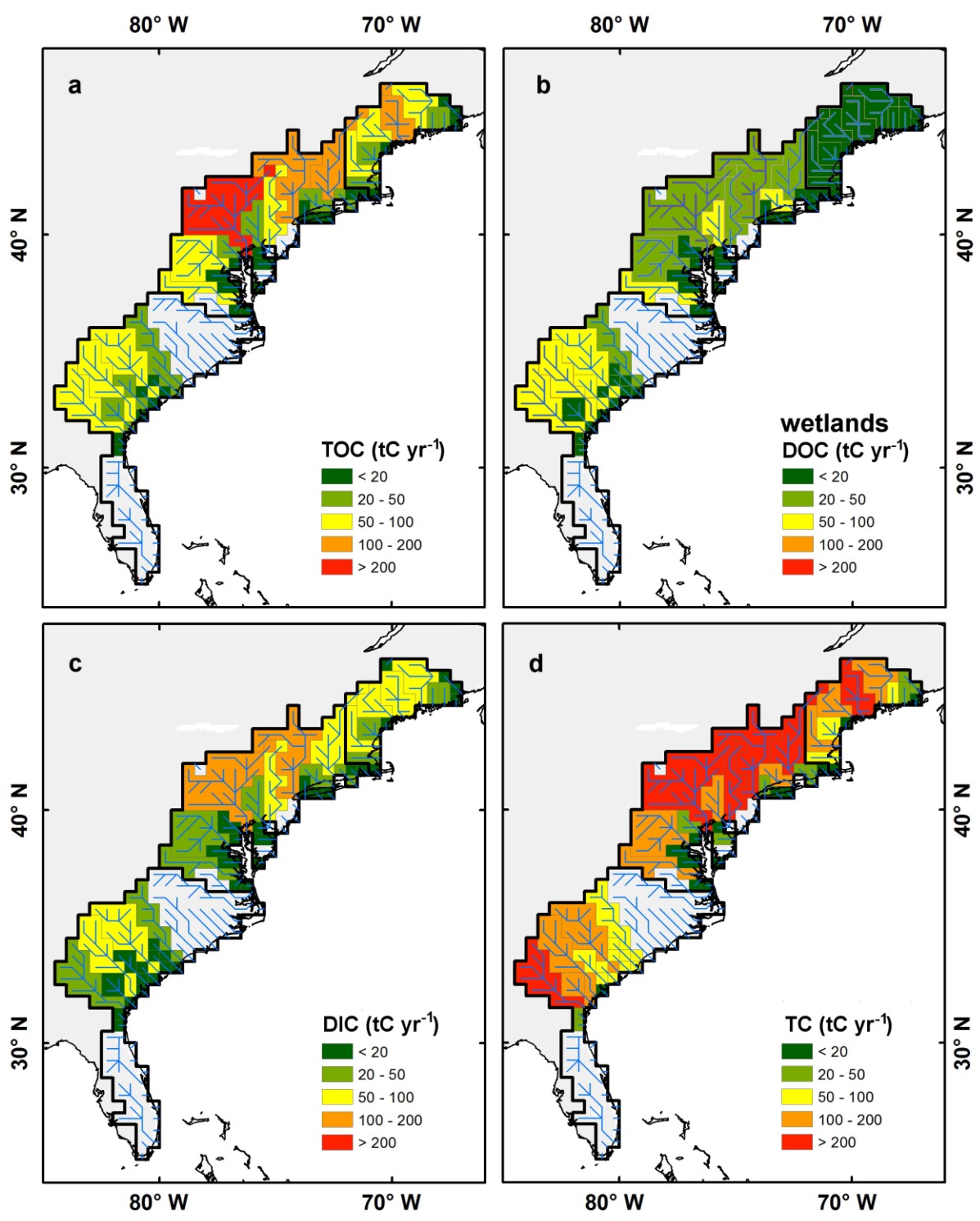


**Figure 5:** Annual river carbon loads of TOC (a), annual DOC fluxes from wetlands (b), annual river
carbon loads of DIC (c) and annual TC fluxes (d). All fluxes are indicated per watershed.


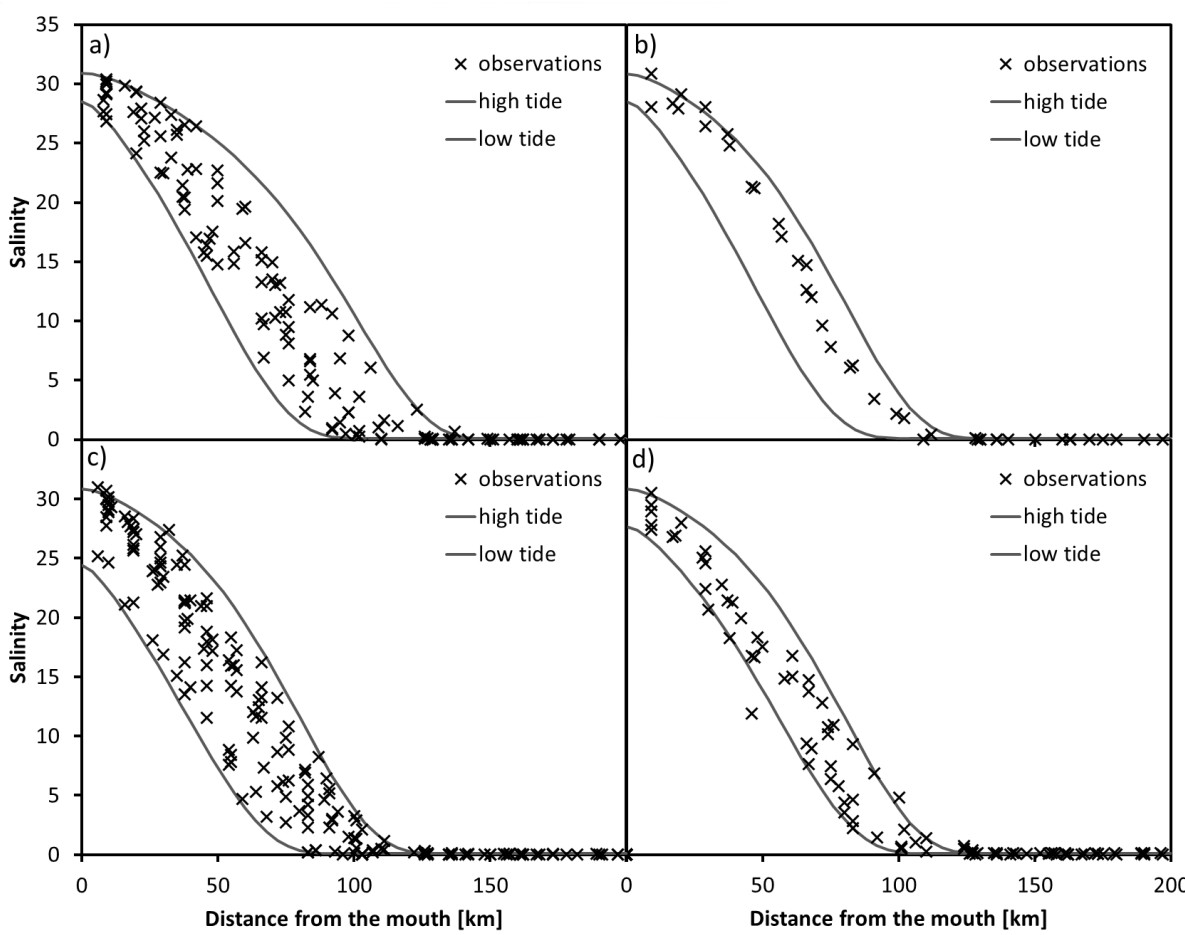


**Figure 6.** Modeled (lines) and measured (crosses) salinities in the Delaware Bay estuary for January

(a), February (b), May (c), June (d). The two lines correspond to high and low tides.


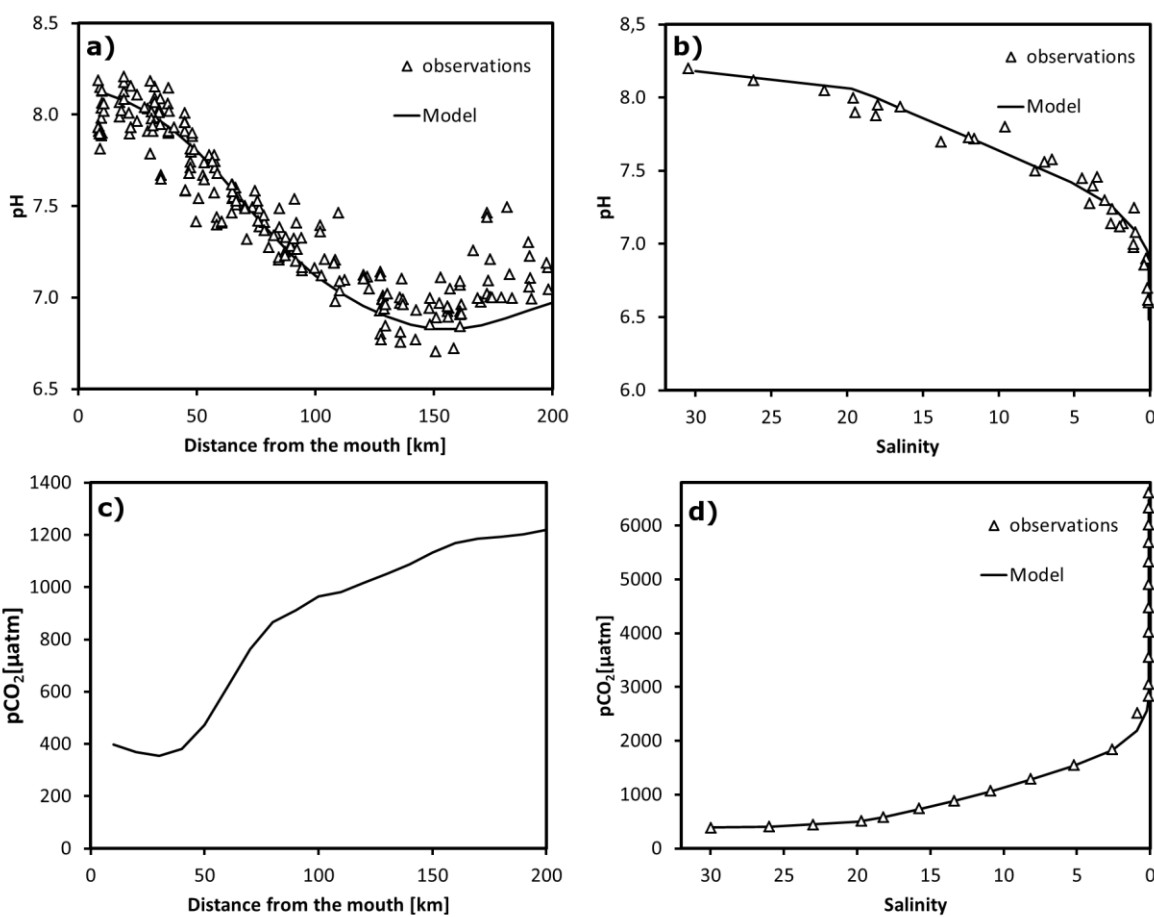


**Figure 7.** Longitudinal profiles of pH (top) and pCO$_2$ (bottom) for the Delaware Bay (left) and
Altamaha river estuary (right).


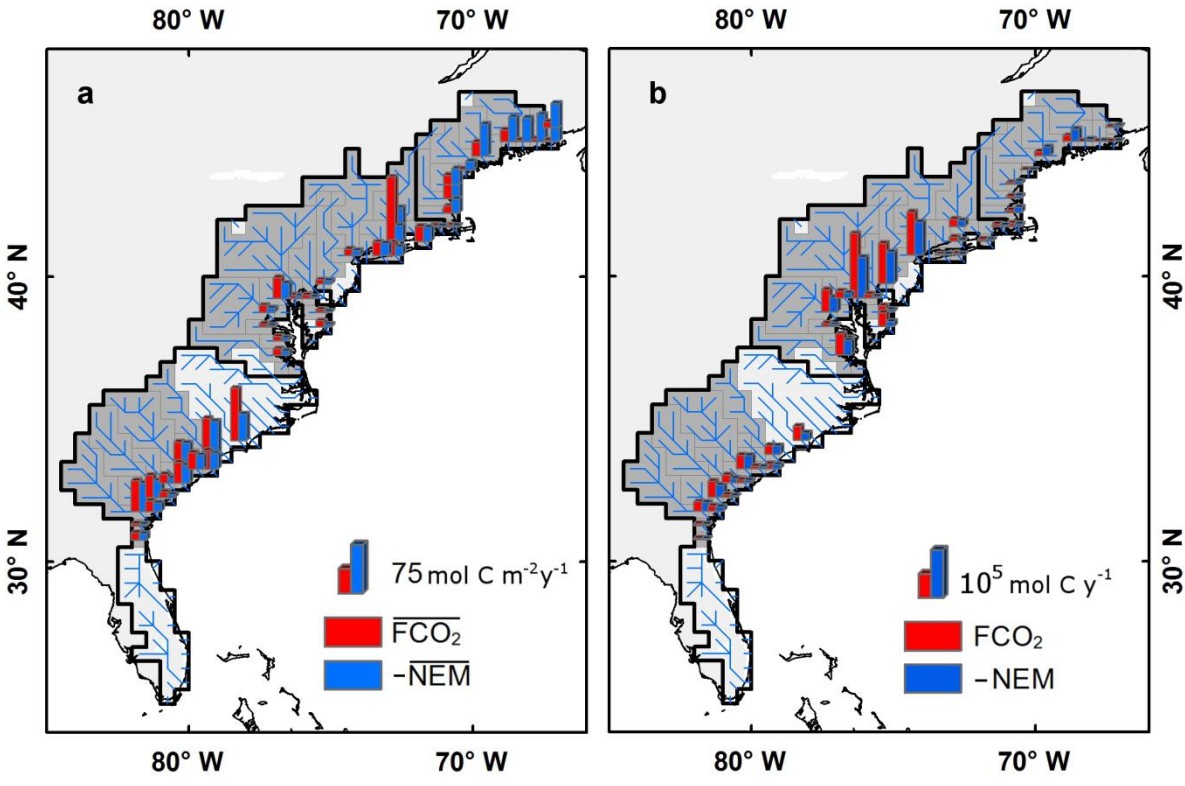


**Figure 8:** Spatial distribution of spatially averaged value (a) and integrated value (b) of mean annual
*FCO₂* (red) and *-NEM* (blue) along the East coast of the US. On panel a, the notation with overbars
($\overline{FCO_2}$ and $-\overline{NEM}$) represents rates per unit surface. For the sake of the comparison with $\overline{FCO_2}$, Fig.
8 displays $-\overline{NEM}$ because the model predicts that all estuaries in this region are net heterotrophic.

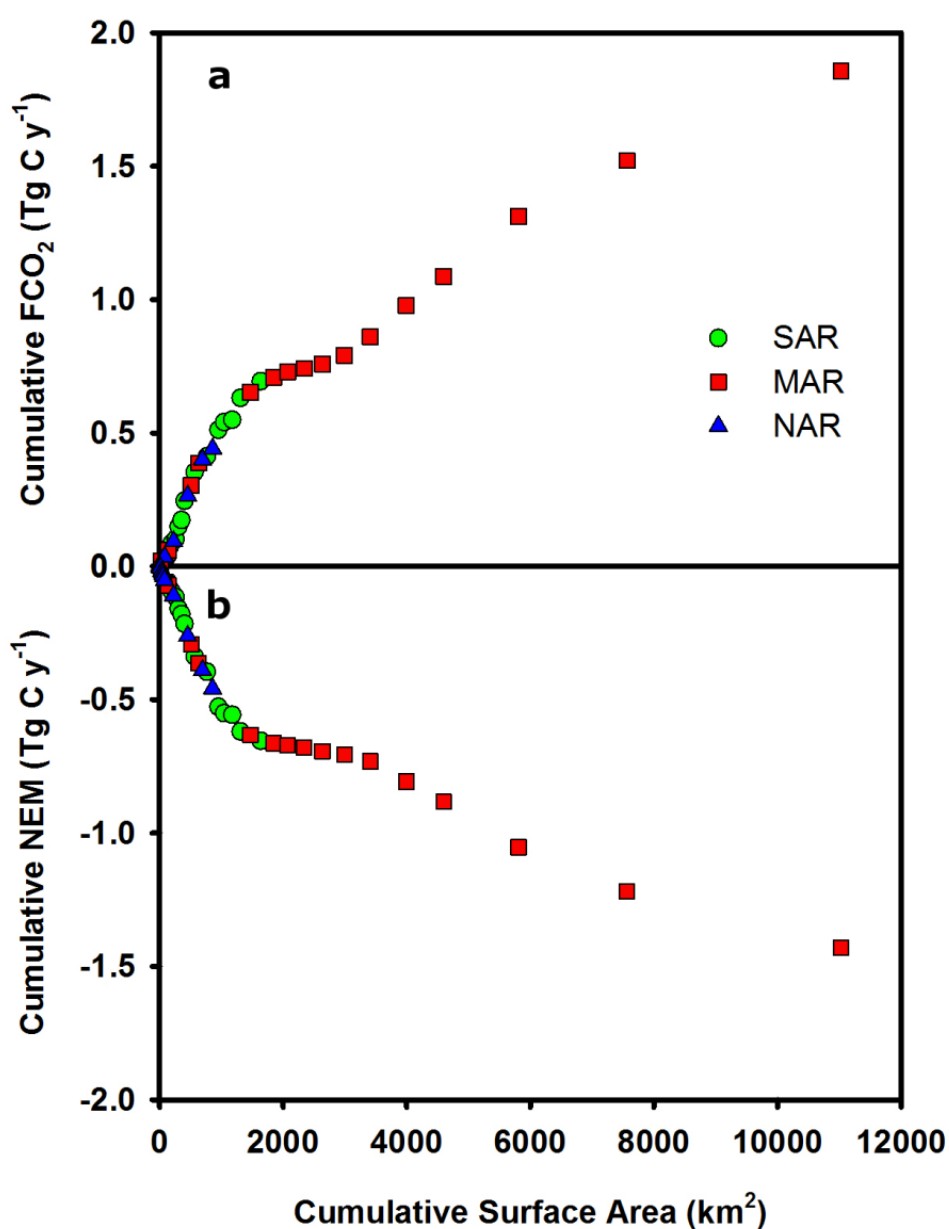


**Figure 9:** The Cumulative *FCO₂* (a) and *NEM* (b) as functions of the cumulative estuarine surface area. Systems are sorted by increasing surface area.




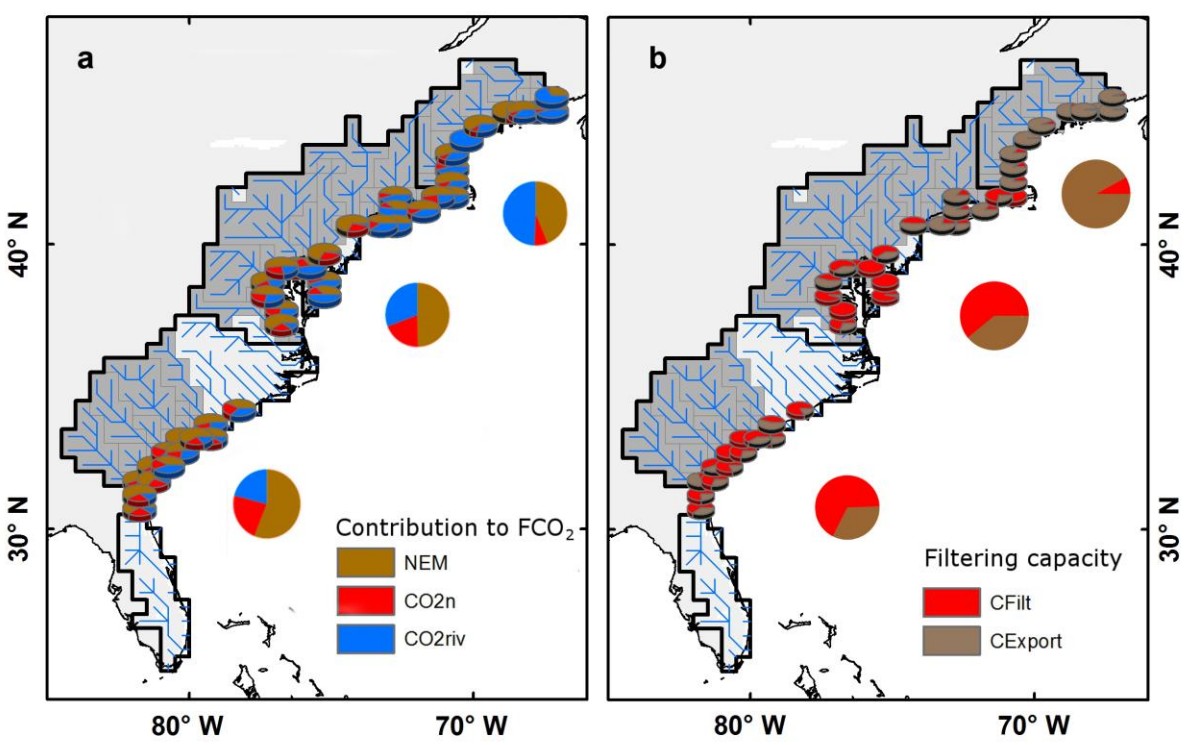


**Figure 10:** Contribution of *NEM*, nitrification and riverine waters super-saturated waters to the mean annual *FCO₂* (a). Spatial distribution of mean annual carbon filtration capacities (*CFilt*) and export (*CExport*) along the East coast of the US (b).


## a) Eastern US coast

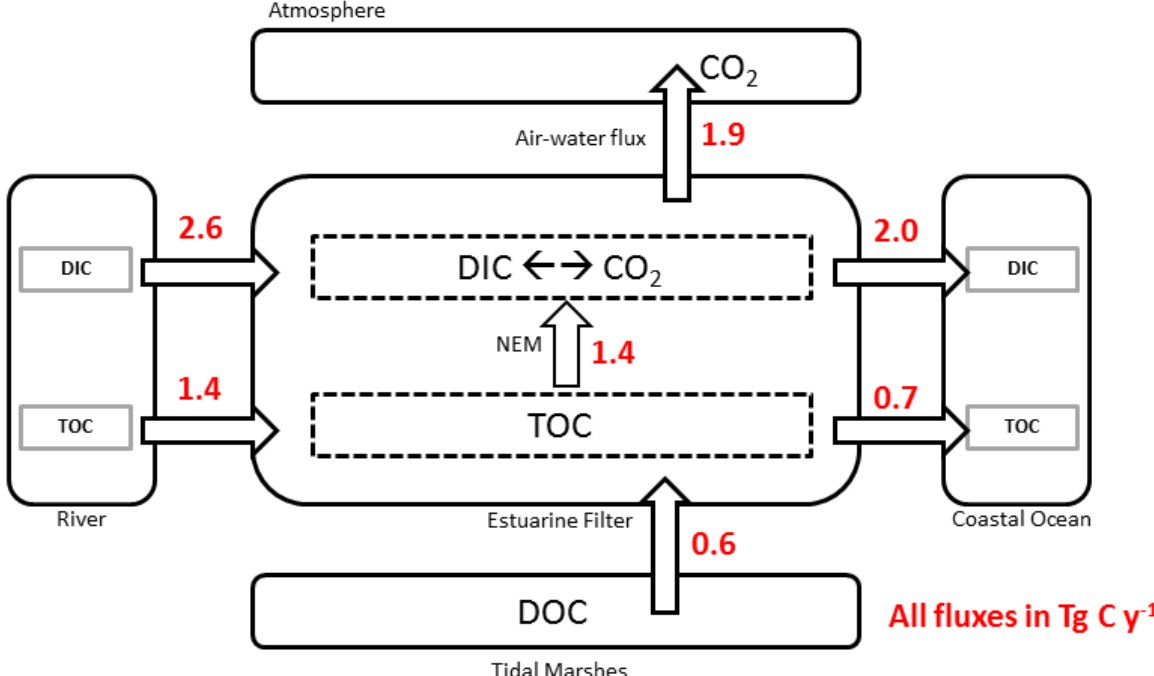

## b) North Sea coast

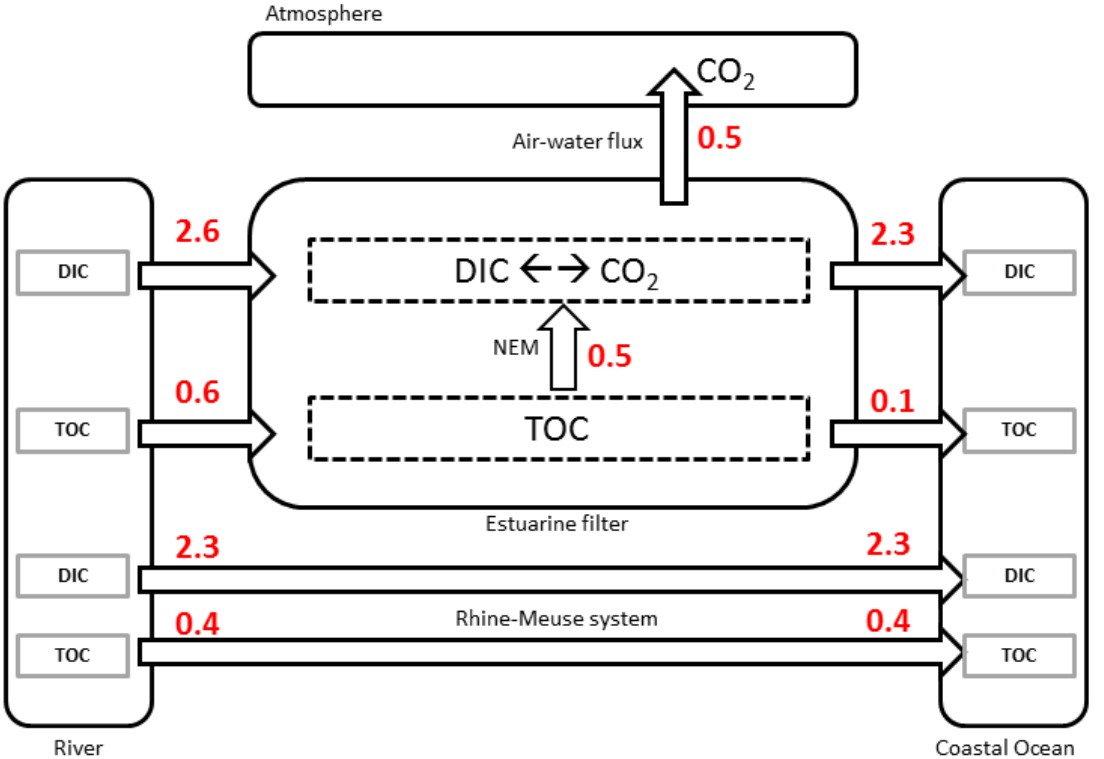

**Figure 11:** Annual carbon budget of the estuaries of the East coast of the US (a) and of the coast of the North Sea (b, modified from Volta et al., 2016a).

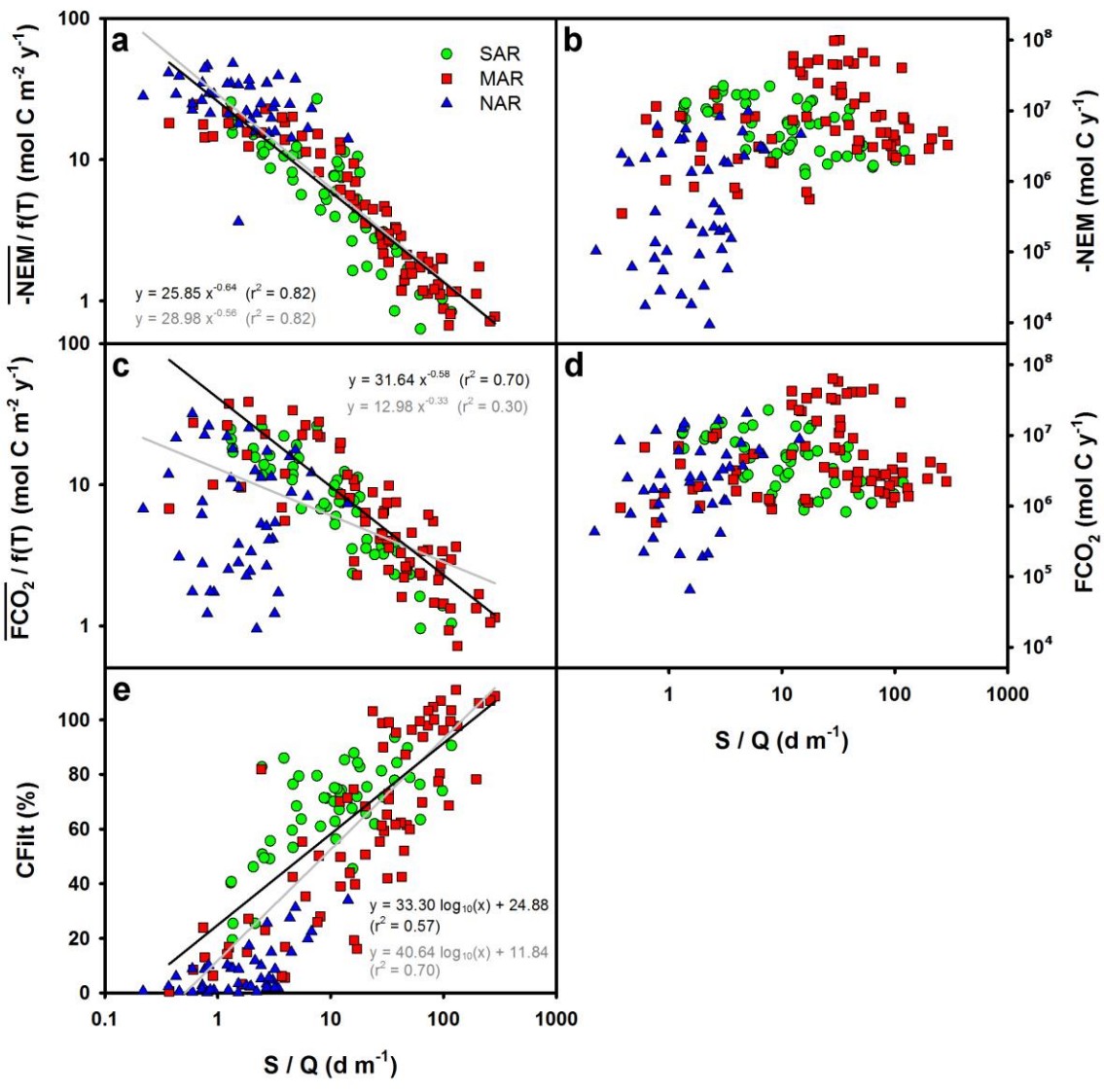


**Figure 12:** System scale integrated biogeochemical indicators expressed as functions of the depth

normalized residence time expressed as the ratio of the estuarine surface S and the river discharge Q

for all seasons. Panels b, d and e represent -NEM, $FCO_2$ and CFilt, respectively. Panels a and c

represent -NEM, $FCO_2$ normalized by a temperature $Q_{10}$ function. Black lines are the best fitted linear

regressions obtained using all the point. Grey lines are best fit using only the estuaries from the MAR

and SAR regions.

1377