# Peer review of "Air-water CO₂ evasion from U.S. East Coast estuaries"

_Biogeosciences, 2016_

## Referee Comment (RC1) · Anonymous Referee #1 · 4 Sep 2016

After reviewing this manuscript, I do not know that weather I should believe the estimated results for the mean annual net ecosystem metabolism (NEM), FCO2, and CFilt or not, because it is a big issue for air-sea interaction. I am confusing that the authors tried to use a simple 1D model coupled with Global News model in current study. How did they do? Many assumptions should be made to compromise the estimated results for air-sea CO2 evasion. The authors should clarify many assumptions in their study. In the model, many parameters should be set up to simulate the state variables shown in Table 2. How did the authors select those parameters? The parameters and values should be listed clearly. In 2.6 Model-data comparison, the description of this subsection is very poor. The authors described the model validation for other estuaries in the Europe. How did the authors validate the model for the study areas (U.S. east coast estuaries)? I would like to see the model validation in the study areas to convince me

the model is capable and suitable to be used in U.S. east coast estuaries.

---

## Referee Comment (RC2) · Anonymous Referee #2 · 8 Nov 2016

Overall statements

The manuscript "Air-water CO2 evasion from U.S. East Coast estuaries" by Goossens, N., Gildas, L.G., Arndt, S., Regnier, P. gives valuable estimates on the main biogeo-chemical fluxes of the estuaries along the US east coast. The authors model 43 tidal estuaries and subdivide the results into 3 different latitudinal zones showing distinct differences which appear reasonable. The problem is that the reader has to accept these "black box results" even though all the details of the different estuaries should be available. I will pinpoint the problems and possible ways to resolve them:

* The data preparation for the 43 estuaries is not transparent and reproducible. Please prepare a table in which all details for each estuary are inserted (like Volta et al., 2016a, Tab. 1). If this table appears too large, put it into the Appendix (supplemental data).

Please use lat/lon positions of the mouth and estuary names if possible.

* The validation chapter only refers to applications elsewhere. Please validate the model for at least one estuary in each latitudinal zone like Volta et al., 2016a did it for some North Sea estuaries.

* You used some arguable boundary conditions and forcing functions: The Alkalinity near the mouth, DIC and Alkalinity from GLORICH positions closest to the river boundary or older discharge estimates. I know that it is difficult to put this all together in a reasonable way. But the reader should get knowledge about the sensitivities of the model in relation to estimates of boundary or forcing data. Please show how the model reacts on changes in these data. In the detailed statements I will show in which context such studies should be done.

As different input parameters are means over several years, the time span of validity of the results should be defined.

The introduction reads rather as an advertising text. Give, for example, details about the structure of the ms. A question, which could be tackled, is whether global models miss estuarine processes (Line 39).

Detailed statements

L14/15 Write 697.000 km2

L19 For which time period?

L19 Only CO2, or also other gases including carbon?

L25 the results

L100 Make a full sentence: For a review see Laruelle et al. (2013)

L107/109/124 unify "Fig. x" -> all over the text

L125 give lat/lon for these stations or enlarge Fig. 1 and indicate individual stations.

L146ff 47 stationed were simulated. This contradicts the number of 43 (abstract). 15+47 is not 64 as I would expect from this sentence.

L151 Do you have a reference for this?

L152 Tab. x not table x (all over the text)

L169 2.9 m (use space)

L207 "These parameters were determined through.."?

L210 are described

L226 Use C only for concentration

L227 You mean eqs (5) and (6)?

L233 You mean eqs (5) and (6)?

L239 Use only English peer reviewed references

L241 Define N by an equation

L260 Omit brackets

L262 2000 m (use space)

L272 273 please give a more detailed description here

L276 For which year? Or are climatological or mean values used?

L289 cloud coverage: Which is the origin of this data?

L293 Are there no recent data available?

L320 You mean 50 g C (g Chla)-1 ?

L332 339 W is not consistently defined. Is it percentage or surface area?

L341 give definition of "a"

L355 358 It seem that you use this argument twice. Here a sensitivity analysis would help.

L364 I doubt that zero concentration for org C is appropriate at open sea boundary. Often org C is transported from the open sea into the estuaries were it is degradated. Please substantiate this assumption.

L374 domain -> boundary

L377 Why longitudinal profiles? You mean at right angles with the river flow?

L383 How large do you estimate the error when neglecting degradation or burial in bottom sediments? A sensitivity test could help.

L408 Fluxes

L430 Boundary conditions and forcings differ from European settings. Show validations for American estuaries.

L443 a regional minimum

L440-456 Give these numbers in a table and discuss the most relevant ones.

L457-462 The percentages should sum up to 100%

L466 What do you mean with "aspect ratio"?

L479ff Why do the small estuaries show higher mean values?

L485 Give more details about the assumptions made to calculate the partitioning for Fig. 8a. Were seasonal partitioning combined to overall partitioning?

L489 Give more details about the different partitioning in the different zones here

L508 Where is Table S1?

L577 budgets

L630 The normalization of NEM by a Q10 value appears reasonable. The normalization of FCO2 by a Q10 value must be justified. I'm not convinced of the latter normalization.

L660 ff Here it becomes obvious that f(t) cannot be applied to FCO2.

L668 whom -> who

L677 In this case the assumption of pCO2 in equilibrium with the atmosphere at the lower boundary contradicts the case "still oversaturated waters .."

L682 No link to Fig. 10d ?

L739 You really mean "prediction"? Not "projection"?

L740 As your model is rather based on empirical relations than on first principles, I expect that changed systems due to climate shifts and consequences can change your basic relationship. Please include this aspect in a more careful outlook.

L1021 7(4), 1271-1295

L1045 give units and if possible your own values.

L1052 definition of winter (DJF)?

L1103 The caption must be understandable alone.

L1105 Separate: "black lines .. using all points" " grey lines are best fit only for .."

---

## Author Comment (AC1) · 22 Jan 2017

Please, find our answers, the updated manuscript (with track changes) and supplementary material in the zip archive in attachment.

Please also note the supplement to this comment:
http://www.biogeosciences-discuss.net/bg-2016-278/bg-2016-278-AC1-supplement.zip

---

## Author Response (AR1)

**Answer to Reviewer #1**

After reviewing this manuscript, I do not know that weather I should believe the estimated results for the mean annual net ecosystem metabolism (NEM), FCO2, and CFilt or not, because it is a big issue for air-sea interaction. I am confusing that the authors tried to use a simple 1D model coupled with Global News model in current study. How did they do? Many assumptions should be made to compromise the estimated results for air-sea CO2 evasion. The authors should clarify many assumptions in their study. In the model, many parameters should be set up to simulate the state variables shown in Table 2. How did the authors select those parameters? The parameters and values should be listed clearly.

**We agree with the reviewer that a better representation of carbon dynamics through the quantification of the Net Ecosystem Metabolism, $CO_2$ outgassing and carbon filtration in estuarine systems is a critical issue for air-sea interaction. This topic is a particularly pressing matter at the regional scale due to the difficulty of deriving consistent regional budgets from the upscaling of rare local measurements performed in morphologically complex and profoundly heterogeneous systems (Borges and Abril, 2011; Laruelle et al., 2013; Regnier et al., 2013). On the modeling side, the set-up of a reliable reactive transport model able to realistically capture the estuarine carbon dynamics generally proves a very costly endeavor in terms of data requirement to constrain the model (i.e. bathymetric data, boundary conditions, climatic forcing…) and in terms of time necessary to develop such model and run it (see e.g. Garnier et al., 2001; Huret et al., 2005; Arndt et al., 2011; Mateus et al., 2012). The model presented here is thus developed as a compromise, as it is currently the only one capable of running regional scale simulations with limited data and computation needs without sacrificing too much to oversimplification (as done when using box models to represent estuarine systems, Gordon et al., 1996). It follows from several studies published over the past few years (Regnier et al., 2013; Volta et al., 2014, 2016a, 2016b) that led to the development of a 1 dimensional generic estuarine model for tidal systems (Volta et al., 2014) forced by a set of generic parameters compiled from an unprecedented literature review (Volta et al., 2016a). This model was successfully applied and validated on several European estuaries (the Scheldt and Elbe, in particular, see Volta et al., 2016a&b) as well as at the regional scale of the North Sea, using a strategy similar to that presented here (Volta et al., 2016b). This strategy involved the use of the same boundary conditions as those used here for the east coast of the US. That is, the outputs of the global river model GLOBALNEWS and the global river carbon database GloRiCH to constrain upstream boundary conditions and the use of the World Ocean Atlas to specify the downstream boundary conditions. In other words, the model described in our**

manuscript has precisely been designed to produce regional estuarine carbon budgets using the outputs of GlobalNEWS as boundary conditions and was already successfully used for similar purpose in another region.

As a consequence of the reviewer's skepticism and following numerous precise suggestions form the other reviewer, we have substantially modified the manuscript to better describe and justify our methodology, its underlying assumptions and potential limitations. We have also made the set-up of our simulations more transparent to secure reproducibility of our model results. In particular, the updated version of the manuscript now contains:

- A substantially modified introduction that puts our study into a more precise context and provides an improved description of the structure of the manuscript.

- Numerous additions to the model description section in order to clarify and substantiate all the assumptions on which our model relies on (i.e. calculation of boundary conditions, period of simulation, choice of databases, etc…), and which together, describe in much more detail the set-up of our simulations.

- 6 comprehensive tables (presented in the supplementary information) and which contain all physical forcings (i.e. estuarine geometry, wind speed, temperature…) and boundary conditions (nutrients and carbon concentrations, pH, alkalinity…).

- A new section (3.3. Scope of applicability and model limitations) which reflects on the strength and weaknesses of our modeling strategy in light of the current state of knowledge available to constrain a model such as ours. In particular, the adequacy of our approach to tackle regional scale modeling, the set-up of boundary condition with available databases and the quantification of the model's uncertainty are addressed in this section.

In 2.6 Model-data comparison, the description of this subsection is very poor. The authors described the model validation for other estuaries in the Europe. How did the authors validate the model for the study areas (U.S. east coast estuaries)? I would like to see the model validation in the study areas to convince me the model is capable and suitable to be used in U.S. east coast estuaries.

We understand the reviewer's concern about the limited validation of our model within the study area. This issue was also pointed out by reviewer #2. We thus expanded extensively section 2.6 to confront the annual $CO_2$ outgassing predicted by our model with 13 published estimates derived from direct measurements performed in estuaries located along the East coast of the US (Table 1). In addition, we provide a validation of our hydrodynamic model using several seasonal longitudinal salinity profiles in the Delaware Bay as well a validation of our biogeochemical model for two estuaries (the Delaware Bay and the Altamaha estuary). These additional simulations reveal that C-GEM is able to properly represent a $pCO_2$ (Delaware Bay) and both pH and $pCO_2$ longitudinal profiles along the estuarine gradient (Altamaha). Also, in the new section 3.3 (Scope of applicability and model limitations), a paragraph discusses the issue of representativeness of the model's performance through local punctual validations in the case of regional simulations including numerous small systems for which the data that would be required to perform a local validation are simply inexistent.

We hope that all these modifications will convince the reviewer of the usefulness and relevance of our study and modelling strategy.

References:

Arndt, S., Lacroix, G., Gypens, N., Regnier, P., and Lancelot, C.: Nutrient dynamics and phytoplankton development along an estuary-coastal zone continuum: A model study. Journal of Marine Systems, 84(3-4), 49-66, 2011.

Borges, A.V., and Abril, G.: Carbon Dioxide and Methane Dynamics in Estuaries. In: E. Wolanski and D.S. McLusky (Editors), Treatise on Estuarine and Coastal Science. Academic Press, Waltham, pp. 119–161, 2012.

Garnier, J., Servais, P., Billen, G., Akopian, M., and Brion, N.: Lower Seine River and Estuary (France) Carbon and Oxygen Budgets During Low Flow, Estuaries, 24, 964–976, 2001.

Gordon, J. D. C., Boudreau, P. R., Mann, K. H., Ong, J. E., Silvert, W. L., Smith, S. V., Wattayakorn, G., Wulff, F., and Yanagi, T.: LOICZ biogeochemical modelling guidelines. LOICZ reports & studies, 5. Texel: LOICZ, 1996.

Huret, M., Dadou, I., Dumas, F., Lazure, P., and Garcon, V.: Coupling physical and biogeochemical processes in the Rio de la Plata plume, Cont. Shelf Res., 25, 629–653, 2005.

Laruelle, G.G., Dürr, H.H., Lauerwald, R., Hartmann, J., Slomp, C.P., Goossens, N., and Regnier, P.A.G.: Global multi-scale segmentation of continental and coastal waters from the watersheds to the continental margins. Hydrol. Earth Syst. Sci., 17(5), 2029-2051, 2013.

Mateus, M., Vaz, N., and Neves, R.: A process-oriented model of pelagic biogeochemistry for marine systems. Part II: Application to a mesotidal estuary, J. Mar. Syst., 94, 90–101, 2012.

Regnier, P., Arndt, S., Goossens, N., Volta, C., Laruelle, G.G., Lauerwald, R., and Hartmann, J.: Modelling Estuarine Biogeochemical Dynamics: From the Local to the Global Scale. Aquatic Geochemistry, 19(5-6), 591-626, 2013.

Volta, C., Arndt, S., Savenije, H.H.G., Laruelle, G.G., and Regnier, P.: C-GEM (v 1.0): a new, cost-efficient biogeochemical model for estuaries and its application to a funnel-shaped system. Geosci. Model Dev., 6(4), 5645-5709, 2014.

Volta, C., Laruelle, G. G., and Regnier, P.: Regional carbon and $CO_2$ budgets of North Sea tidal estuaries, Estuarine, Coastal and Shelf Science, 176, 76-90, 2016a.

Volta, C., Laruelle, G. G., Arndt, S., and Regnier, P.: Linking biogeochemistry to hydro-geometrical variability in tidal estuaries: a generic modeling approach, Hydrol. Earth Syst. Sci., 20, 991-1025, doi:10.5194/hess-20-991-2016, 2016b.

**Answer to Reviewer #2**

Overall statements

The manuscript "Air-water CO2 evasion from U.S. East Coast estuaries" by Goossens, N., Gildas, L.G., Arndt, S., Regnier, P. gives valuable estimates on the main biogeochemical fluxes of the estuaries along the US east coast. The authors model 43 tidal estuaries and subdivide the results into 3 different latitudinal zones showing distinct differences which appear reasonable. The problem is that the reader has to accept these "black box results" even though all the details of the different estuaries should be available. I will pinpoint the problems and possible ways to resolve them:

We are grateful for the reviewer's evaluation and the constructive suggestions provided. We understand that the reviewer is mainly concerned about an apparent 'lack of transparency', as well as a seemingly weak validation of the model within the study area. Following the reviewer's recommendations, we thus substantially modified the manuscript to respond to these concerns. More specifically, we added a comparison between model-predicted annual $CO_2$ outgassing fluxes and 13 published flux estimates, derived from direct measurements in local estuaries to section 2.6 (Model-data comparison). In addition, we also provide new validations of the hydrodynamic and biogeochemical model (section 2.6). Furthermore, we introduced a new section (section 3.4), which critically discusses the scope of applicability and model limitations.

Please find bellow a detailed answer to each comment. All our answers are written in blue and the modifications within the text are highlighted in bold and italic. In the revised manuscript, changes are tracked via Word's track changes tool.

On behalf of all co-authors,

Goulven Laruelle

* The data preparation for the 43 estuaries is not transparent and reproducible. Please prepare a table in which all details for each estuary are inserted (like Volta et al., 2016a, Tab. 1). If this table appears too large, put it into the Appendix (supplemental data).

In the revised manuscript, we now provide 5 additional, extensive tables as supplementary information, which summarize all key parameters and boundary conditions required to perform the simulations. In addition to table SI1, which already provided the estuarine surface areas, as well as fresh water discharge fluxes for all systems and seasons, these new tables provide:

Table SI2: Geometric properties of the estuary (i.e. length, width at both boundaries, depth and convergence length)

Table SI3: Upstream boundary conditions for nutrients and chlorophyll concentrations

Table SI4: Downstream boundary conditions for nutrients and chlorophyll concentrations

Table SI5: Upstream boundary conditions for the organic carbon and carbonate system (i.e. TOC, DIC, pH…)

Table SI6: Downstream boundary conditions for the organic carbon and carbonate system (i.e. TOC, DIC, pH…)

Please use lat/lon positions of the mouth and estuary names if possible.

As requested, latitudes and longitudes, as well as the names of the largest rivers are provided for each estuary in all aforementioned tables. In addition, within the main text, we now make reference, whenever possible, to the name and coordinates of the estuaries that are being discussed.

* The validation chapter only refers to applications elsewhere. Please validate the model for at least one estuary in each latitudinal zone like Volta et al., 2016a did it for some North Sea estuaries.

The general performance of C-GEM in reproducing and predicting estuarine hydrodynamics and biogeochemical cycling has been extensively tested across a large range of different estuarine systems (e.g. Volta et al., 2014, 2016, see also Savenije 2001 for the estuarine physics). Here, we extended these tests by a number of local model-data comparisons. We added a new comparison between model-predicted annual $CO_2$ outgassing fluxes and 13 published flux estimates, derived from direct local measurements to section 2.6 (Table 1). In addition, we also evaluated the performance of the hydrodynamic model by comparing simulation results with seasonal, longitudinal salinity profiles in the Delaware Bay. Furthermore, the performance of the biogeochemical model is critically evaluated by comparing simulation results with longitudinal profiles of $pCO_2$ and pH, in the Delaware Bay and the Altamaha River estuary. These additional model-data comparisons reveal that C-GEM is able to reproduce local measurements of $pCO_2$ (Delaware Bay), as well as longitudinal pH and $pCO_2$ profiles (Altamaha). Following what was done in Volta et al. (2016) with the Scheldt and the Elbe in Europe, the choice of using the Delaware Bay and the Altamaha river estuary was motivated by their contrasting geometries: The Delaware Bay is a marine dominated system characterized by a pronounced funnel shape while the Altamaha River ends with a very prismatic estuary characteristic of river dominated systems (Jiang et al., 2008). Thus, selecting these two end-members estuaries reveals the ability of C-GEM to simulate widely differing estuarine dynamics. Although we agree that performing a simulation on a system located in the Northern region would be a valuable addition, we could not find suitable a suitable set of observed nutrient and carbon boundary conditions and their corresponding longitudinal profiles. Note, however, that several flux values reported in Table 1 refer to estuaries located in this region. Finally, within the new section 3.4 (Scope of applicability and model limitations), we critically discuss the difficulties associated with 'validating' regional/global model simulations with a limited set of local, instantaneous observations, as well as the uncertainties that arise from the proposed model approach.

See updated manuscript at the end of this document

* You used some arguable boundary conditions and forcing functions: The Alkalinity near the mouth, DIC and Alkalinity from GLORICH positions closest to the river boundary or older discharge estimates. I know that it is difficult to put this all together in a reasonable way. But the reader should get knowledge about the sensitivities of the model in relation to estimates of boundary or forcing data. Please show how the model reacts on changes in these data. In the detailed statements I will show in which context such studies should be done.

As pointed out by the reviewer, designing realistic and consistent boundary conditions is a critical step in model set-up. In the present study, this task is further complicated by the necessity to find complete sets of boundary conditions for 43 tidal systems located along the eastern coast of the US. C-GEM was specifically designed with this difficulty in mind. Some of the strengths of the model are its comparatively modest data requirement and its transferability from one estuarine system to another, which has already been demonstrated on the estuaries surrounding the North Sea (Volta et al., 2016). C-GEM is also well suited to be operated in conjunction with global databases such as GlobalNEWS (Mayorga et al., 2010) because they share in common the watershed as their fundamental unit , which is essentially what was done in this study. We agree with the reviewer that some of the assumptions and choices of data sources could be better justified and critically discussed in the manuscript. We thus carefully addressed all concerns raised in the answers to the detailed statements of the reviewer. In addition, we also added an entire new section to the manuscript ('Scope of applicability and model limitations') that summarizes the strengths and weaknesses of our regional model approach.

As different input parameters are means over several years, the time span of validity of the results should be defined.

In the present study, simulations are representative for the year 2000 because some the largest datasets we rely on to constrain boundary conditions or forcings (e.g. GlobalNEWS ) are derived from models calibrated for that year. As a consequence, additional data used to constrain boundary conditions and forcing parameters was selected from the same time span. In the revised manuscript, we added a few sentences justifying our data choice and specifying the time period for which simulation results are representative. In addition, we also specify if boundary conditions/forcings are constrained on the basis of punctual measurements or averages of several years.

The introduction reads rather as an advertising text.  Give, for example, details about the structure of the ms.  A question, which could be tackled, is whether global models miss estuarine processes (Line 39).

In the previous version of our introduction, we tried to emphasize the originality and the potential of our modelling approach. We felt that it was important to stress the novelty of the approach developed here: an explicit simulation of seasonal carbon transformations and fluxes along the land-ocean continuum at a regional scale. This work builds on the study of Volta et al. (2016) which provided the first annually average estimates of estuarine carbon transformations and fluxes focusing on the estuaries surrounding the North Sea. However, we agree with the reviewer that the proportion of the introduction dedicated to the presentation of C-GEM was too long and sometime superfluous. Following the reviewer's advice, we shortened the introduction and emphasized the research questions that can be tackled with the presented model approach. We also provided a better overview of the structure of the manuscript and the studied area. Furthermore, we followed the reviewer's suggestion and shortly discuss the ability of global carbon cycle models to account for the influence of the estuarine modulator.

See updated manuscript at the end of this document

Detailed statements

L14/15 Write 697.000 km2

Done

L19 For which time period?

The sentence has been re-written to state that model simulations are representative of the year 2000.

"***Our simulations, performed using conditions representative of the year 2000, suggest that,*** together, US East coast estuaries emit 1.9 TgC yr$^{-1}$ ***in the form of $CO_2$,*** which correspond to about 40 % of the carbon inputs from rivers, marshes and mangrove"

L19 Only CO2, or also other gases including carbon?

Model simulations only account for $CO_2$ exchange at the air-water interface and the sentence has been modified to clarify this point.

"…, together, US East coast estuaries emit 1.9 TgC yr$^{-1}$ ***in the form of $CO_2$,*** which correspond to about 40 % of the carbon inputs from rivers, marshes and mangroves."

L25 the results

Done

"Finally, ***the*** results reveal that the ratio of estuarine surface area to the river discharge, S/Q…"

L100 Make a full sentence: For a review see Laruelle et al. (2013)

We added a sentence.

"These comprehensive data sets are complemented by local observations of carbon cycling and $CO_2$ fluxes in selected, individual estuarine systems, making the East coast of the United States an ideal region for a first, fully explicit regional evaluation of $CO_2$ evasion resolving every major tidal estuary along the selected coastal segment. ***An extensive review of published local estimates of $CO_2$ fluxes in estuarine systems worldwide can be found in Laruelle et al. (2013)."***

L107/109/124 unify "Fig. x" -> all over the text

On the website of Biogeosciences, the guidelines for authors states: 'The abbreviation "Fig." should be used when it appears in running text and should be followed by a number unless it comes at the beginning of a sentence, e.g.: "The results are depicted in Fig. 5. Figure 9 reveals that…".'

In the updated manuscript, we paid attention to strictly follow these rules.

L125 give lat/lon for these stations or enlarge Fig. 1 and indicate individual stations.

In the paragraph summarizing the published annual mean $FCO_2$ estimates based on measurements for Atlantic US estuaries, we replaced the reference to Figure 1 by a reference to Table 3, which provides a list of all the estuaries, as well as their respective coordinates mentioned in that section. Table 3 thus becomes Table 2 in the manuscript.

"A total of thirteen local, annual mean estuarine $CO_2$ flux estimates across the air-water interface based on measurements are also reported in the literature and are grouped along a latitudinal gradient *(Tab. 2)*. "

L146ff 47 stationed were simulated.  This contradicts the number of 43 (abstract).

Some watersheds flow into the same estuarine system and were merged to calculate boundary conditions for some of the systems (see section 2.4). The sentence the reviewer refers to mistakenly refers to the number of watersheds represented in our simulations rather than the number of estuaries actually represented. Only 43 tidal estuarine systems were simulated and the text was updated in sections 2.2, 2.3 and 3.4. The abstract, which refers to 43 estuaries, is thus still correct. We made the following corrections in the manuscript:

Line 147:

"The National Estuarine Eutrophication Assessment (NEAA) survey (Bricker et al., 2007), which uses geospatial data from the National Oceanic and Atmospheric Administration (NOAA) Coastal Assessment Framework (CAF) (NOAA, 1985), was used to identify and characterize *58* estuarine systems discharging along the Atlantic coast of the United States. From this set, *43* 'tidal' estuaries, defined as a river stretch of water that is tidally influenced (Dürr et al., 2011), were retained (fig.1) to be simulated by the C-GEM model, which is designed to represent such systems."

Line 171:

"The generic 1D Reactive-Transport Model (RTM) C-GEM (Volta et al., 2014) is used to quantify the estuarine carbon cycling in the *43* systems considered in this study."

Line 278:

"First, *43* coastal cells corresponding to tidal estuaries are identified in the studied area (Fig. 1)."

Line 568:

"The overall carbon filtering capacity of the region thus equals 41% of the total carbon entering the *43* estuarine systems (river + saltmarshes)."

15+47 is not 64 as I would expect from this sentence.
This is indeed a mistake, only 58 estuarine systems are presented (15+43, see comment above).

L151 Do you have a reference for this?

This figure was calculated using the GLOBALNEWS data (Mayorga et al., 2010) for POC and DOC combined with data from Hartmann et al. (2009) for DIC. We modified the sentence in a way that reflects the fact that we performed this calculation ourselves.

*"**Using outputs from terrestrial models (Hartmann et al., 2009; Mayorga et al., 2010), the cumulated riverine carbon loads for all the** non-tidal estuaries that are excluded from the present study **amount to 0.9 Tg C yr$^{-1}$ , which represents less than 15% of the total riverine carbon loads of the region. These 15 systems** are located in the SAR (10) and in the MAR (5)."*

L152 Tab. x not table x (all over the text)

We update the entire manuscript.

L169 2.9 m (use space)

Done

L207 "These parameters were determined through.."?

The geometric parameters we are referring to can be extracted using Geographic Information Systems (GIS). This widely used type of software allows the determination of, for example, a distance such as the width of an estuary at its mouth from a digitalized map. The manuscript has been modified to spell out GIS on its first occurrence and make clear that GIS is the tool that can be used to extract the data from local maps.

"These parameters can be easily determined **_from_ _local maps or Google Earth using Geographic Information Systems (GIS)_** or obtained from databases (NASA/NGA, 2003)".

L210 are described

Corrected

L226 Use C only for concentration

C is the commonly used symbol for Chézy's coefficient but, to avoid any confusion with concentration, we followed the reviewer's advice and use Cz to denote the Chézy's coefficient. We updated equations (6) and the text.

L227 You mean eqs (5) and (6)?

Correct. The text was modified accordingly

L233 You mean eqs (5) and (6)?

Correct. The text was modified accordingly

L239 Use only English peer reviewed references

The original Dutch reference (Van der Burgh, 1972) was replaced by Savenije (1986), which is the oldest English peer reviewed publication using Van der Burgh's equations to calculate the dispersion coefficient in estuarine systems.

"The effective dispersion at the estuarine mouth can be quantified by the following relation **_(Savenije, 1986)_**:"

L241 Define N by an equation

As stated in the text, the Canter Cremers' estuary number N corresponds to the ratio of the freshwater entering the estuary during a tidal cycle to the volume of salt water entering the estuary over a tidal cycle. We introduced a new equation (new equation 9) to define this parameter.

"...where h0 (m) is the tidally-averaged water depth at the estuarine mouth and N is the dimensionless Canter Cremers' estuary number defined as the ratio of the freshwater entering the estuary during a tidal cycle to the volume of salt water entering the estuary over a tidal cycle *(Simmons, 1955)*:

$$N = \frac{Q_b \cdot T}{P}$$

*In this equation, $Q_b$ is the bankfull discharge ($m^3\ s^{-1}$), T is the tidal period (s) and P is the tidal prism ($m^3$).* For each estuary, N can thus be calculated directly from the hydrodynamic model. "

L355 358 It seem that you use this argument twice. Here a sensitivity analysis would help.

The sentences pointed out by reviewer states that our models ignores DIC fluxes from tidal marshes because all the carbon exports from those systems come under the form of organic carbon and then justifies this assumption with the fact that very little degradation of organic matter takes place in the tidal marshes before it reaches the estuary. Considering the relatively small amounts of carbon that can be degraded within tidal marshes (and the lack of data to constrain such process) because of the short time scale of mixing processes, our assumption implicitly means that some biogeochemical processing taking place within tidal marshes may be accounted for by our estuarine model.

L364 I doubt that zero concentration for org C is appropriate at open sea boundary. Often org C is transported from the open sea into the estuaries were it is degradated.

Please substantiate this assumption.

We agree that the assumption that the open ocean is devoid of organic carbon is simplistic and that some organic carbon can enter the estuarine system from the seaward boundary. However, the focus of our study is to investigate and quantify the fate of the carbon delivered to estuaries by the riverine network. We introduced a sentence to section 2.4.4 to reflect on this limitation.

"This approach also reduces the influence of marine boundary conditions on the simulated estuarine dynamics, especially for all the organic carbon species whose concentrations are fixed at zero at the marine boundary. *This assumption ignores the intrusion of marine organic carbon into the estuary during the tidal cycle but allows focusing on the fate of terrigenous material and its transit through the estuarine filter.*"

In addition, the implications of this assumption are also discussed in a paragraph of the new section 3.4:

"*C-GEM places the lower boundary condition 20 km from the estuarine mouth into the coastal ocean and the influence of this boundary condition on simulated biogeochemical dynamics is thus limited. At the lower boundary condition, direct observations for nutrients and oxygen are extracted from databases such as the World Ocean Atlas (Antonov et al., 2014). However, lower boundary conditions for OC and $pCO_2$ (zero concentration for OC and assumption of $pCO_2$ equilibrium at the sea side) are simplified. This approach does not allow addressing the additional complexity introduced by biogeochemical dynamics in the estuarine plume (see Arndt et al., 2011). Yet, these dynamics only play a secondary role in the presented study that focuses on the role of the estuarine transition zone in processing terrestrial-derived carbon.*"

L374 domain -> boundary

Done

L377 Why longitudinal profiles? You mean at right angles with the river flow?

The term longitudinal profile is commonly used and refers to a concentration profile along the longitudinal (length) axis of the estuary, i.e. from the estuarine mouth to the river. Concentrations are thus representative of the cross-sectional average at the respective longitudinal position.

L383 How large do you estimate the error when neglecting degradation or burial in bottom sediments? A sensitivity test could help.

We agree that neglecting benthic processes is a potential limitation of our model. As the reviewer points out, organic matter degradation and burial may influence the biogeochemical of carbon in some estuaries and affect carbon retention within the system. However, because of the dynamic nature of estuarine sediments and the logistic challenges involved to sample them, direct observations and measurements of benthic processes are even more limited than those available for pelagic processes. Very little is known on the long term fate of organic carbon in estuarine sediments and its burial. Because of this lack of knowledge, benthic processes are not explicitly represented in the model. However, to a certain degree model parameters (such as organic matter degradation, denitrification rate constant) implicitly account for benthic dynamics. We acknowledge that, by ignoring benthic processes and burial in particular, our estimates for the estuarine carbon filtering may be underestimated. These considerations have been incorporated into a paragraph of the new section 3.4:

"*Although the reaction network of C-GEM accounts for all processes that control estuarine $FCO_2$ (Borges and Abril, 2012; Cai, 2011), several, potentially important processes, such as benthic-pelagic exchange processes, phosphorous sorption/desorption and mineral precipitation, a more complex representation of the local phytoplankton community, grazing by higher trophic levels, or multiple reactive organic carbon pools are not included. Although these processes are difficult to constrain and their importance for $FCO_2$ is uncertain, the lack of their explicit representations induces uncertainties in Cfilt. In particular, the exclusion of benthic processes such as organic matter degradation and burial in estuarine sediments could result in an underestimation of Cfilt. However, because very little is known on the long term fate of organic carbon in estuarine sediments, setting up and calibrating a benthic module proves a difficult task. Furthermore, to a certain degree model parameters (such as organic matter degradation and denitrification rate constant) implicitly account for benthic dynamics. We nonetheless acknowledge that, by ignoring benthic processes and burial in particular, our estimates for the estuarine carbon filtering may be underestimated, particularly in the shallow systems of the SAR.*"

In addition, although, the discussion is not centred around the role of benthic processes, a paragraph of the new section 3.4 is also dedicated the difficulty of quantifying the uncertainties of model simulations and an attempt is made using the sensitivity analysis performed by Volta et al. (2014, 2016b).

"*Biogeochemical model parameters for regional and global applications are notoriously difficult to constrain (Volta et al., 2016b). Model parameters implicitly account for processes that are not explicitly resolved and their transferability between systems is thus limited. In addition, published parameter values are generally biased towards temperate regions in industrialized countries (Volta et al., 2016b). A first order estimation of the parameter uncertainty associated to the estuarine carbon removal efficiency (CFilt) can be extrapolated from the extensive parameter sensitivity analyses carried out by Volta et al. (2014, 2016b). These comprehensive sensitivity studies on end-member systems have shown that the relative variation in Cfilt when a number of key biogeochemical parameters are varied by two orders of magnitude varies by is ±15 % in prismatic (short residence time on order of days) to ±25 % in funnel-shaped (long residence time) systems. Thus, assuming that uncertainty increases linearly between those bounds as a function of residence time, an uncertainty estimate can be obtained for each of our modelled estuary. With this simple method, the simulated regional Cfilt of 1.9 Tg C $yr^{-1}$ would be associated with an uncertainty range comprised between 1.5 and 2.2 Tg C $yr^{-1}$. Our regional estuarine $CO_2$ evasion estimate is thus reported with moderate confidence. Furthermore, in the future, this uncertainty range could be further constrained using statistical methods such as Monte Carlo simulations (e.g. Lauerwald et al., 2015).*"

L408 Fluxes

Done

L430 Boundary conditions and forcings differ from European settings. Show validations for American estuaries.

We added validations for the American estuaries. Section 2.6 (Model-data comparison) now includes a comparison between model-predicted annual $CO_2$ outgassing fluxes and 13 published flux estimates, derived from direct measurements in local estuaries to section 2.6 (Model-data comparison). In addition, we provide a validation of our hydrodynamic model using several seasonal longitudinal salinity profiles in the Delaware Bay as well a validation of our biogeochemical model on the basis of pH and $pCO_2$ profiles from two estuaries (the Delaware Bay and the Altamaha estuary). These additional simulations reveal that C-GEM is able to reproduce observed $pCO_2$ (Delaware Bay) and both pH and $pCO_2$ longitudinal profiles along the estuarine gradient (Altamaha).

"*Although C-GEM has been specifically designed and tested for the type of regional application presented here, its transferability from North Sea to US East Coast estuaries was further evaluated by assessing its performance in two East Coast estuaries. First, the hydrodynamic and transport model was tested for the Delaware Bay (MAR). The model was forced with the monthly, minimal and maximal observed discharge at Trenton over the period between 1912 and 1985 (UNH/GRDC Database). Simulated salinity profiles are compared with salinity observations from January, February, May and June (the months with the highest number of data entries), which were extracted from the UNH/GRDC Database. Fig. 6 shows that the model captures both the salinity intrusion length and the overall shape of the salinity profile well. In addition, the performance of the biogeochemical model and specifically its ability to reproduce pH and $pCO_2$ profiles was evaluated by a model-data comparison for both the Delaware Bay (MAR) in July 2003 and the Altamaha river estuary (SAR) in October 1995. Similar to Volta et al., 2016a, the test systems were chosen due to their contrasting geometries. The Delaware Bay is a marine dominated system characterized by a pronounced funnel shape, while the Altamaha River has a prismatic estuary characteristic of river dominated systems (Jiang et al., 2008). Monthly upstream boundary conditions for nutrients, as well as observed pH data and calculated $pCO_2$ are extracted from datasets described in (Sharp, 2010) and (Sharp et al., 2009) for the Delaware and in (Cai and Wang, 1998; Jiang et al., 2008) and (Cai et al., 1998) for the Altamaha river estuary. The additional forcings and boundary conditions are set similarly to the simulation for 2000 (see table 2, 3, 4, 5, 6 in SI). Fig. 7 shows that measured and simulated pH values are in good agreement with observed pH and observation-derived calculations of $pCO_2$. In the Delaware Bay, a pH minimum is located around km 140 and is mainly caused by intense nitrification sustained by large inputs of $NH_4$ from the Philadelphia urban area, coupled to an intense heterotrophic activity. Both processes lead to a well-developed $pCO_2$ increase in this area (Fig. 7b). Although no $pCO_2$ data were available for validation for the period from which boundary conditions were extracted, the simulated profile agree with $pCO_2$ measurement from July 2013 presented by Joesoef et al. (2015) with $pCO_2$ values close to equilibrium with the atmosphere in the widest section of the Delaware Bay (close to the estuarine mouth) and values above 1200 µatm at salinities below 5. For the Altamaha river estuary, pH steadily increases from typical river to typical coastal ocean values (Fig. 7b). In addition, both observations and model results reveal that outgassing is very intense in the low-salinity region with more than a 5 fold decrease in $pCO_2$ between salinity 0 and 5 (Fig 7d)."*

In addition, the new section 3.4 (Scope of applicability and model limitations) critically discusses the difficulties of validating regional/global simulations with local data:

*"**The generic nature of the applied model approach and, in particular the application of seasonally/annually averaged or model-deduced boundary conditions renders a direct validation of model results on the basis of local and instantaneous observational data (e.g. longitudinal profiles), which is likely not representative of these long-term average conditions, difficult. Therefore, model performance is evaluated on the basis of spatially aggregated estimates (e.g. regional $FCO_2$ estimates based on local measurements) rather than system-to-system comparisons with longitudinal profile from specific days. However, note that the performance of C-GEM has been intensively tested by specific model-data comparisons for a number of different systems (e.g. Volta et al., 2014, 2016a) and we are thus confident of its predictive capabilities.**"*

L443 a regional minimum

Done

L440-456 Give these numbers in a table and discuss the most relevant ones.

These results were compiled for all estuaries and seasons in supplementary table SI1 and discussed within the text of section 3.1.

L457-462 The percentages should sum up to 100%

When taking all decimals into account, the percentage values do sum up to 100%. We now provide the exact value in the new manuscript.

"In contrast, the 18 MAR estuaries, with their large relative contribution to the total regional estuarine surface area, account for ***as much as 70.1%*** of the total outgassing."

L466 What do you mean with "aspect ratio"?

Aspect ratio refers to the geometry of the estuary (which subsequently affect its biogeochemical behaviour) and more explicitly refers to the ratio between the estuarine width b0 and convergence length b. A wider, funnel shaped estuary whose dynamics are controlled by a strong marine influence while the dynamics in a narrower prismatic estuary is dominated by the river influence (Savenije, 2001).

In the text, the sentence has been modified and the meaning of the term aspect ratio has been clarified:

"The comparatively larger relative contribution of the NAR to the total NEM as compared to the total $FCO_2$ can be explained by the importance of the specific aspect ratio for NEM. ***A larger ratio of estuarine width b0 and convergence length b corresponds to a more funnel shaped estuary while a low ratio corresponds to a more prismatic geometry (Savenije, 2001; Volta et al., 2014)***"

L479ff Why do the small estuaries show higher mean values?

In large systems, the total outgassing of $CO_2$ extends over a much larger surface area. In small estuaries, the surface area acts as a limiting factor for the gas exchange with the atmosphere.

L485 Give more details about the assumptions made to calculate the partitioning for

In addition to the reference to Regnier et al. (2013), more details regarding the method used to calculate the respective contributions to the estuarine $CO_2$ outgasing (NEM, nitrification and riverine oversaturated $CO_2$) is now provided.

"***Following the approach used in Regnier et al. (2013),*** the contribution of biogeochemical process to *FCO2* is assessed by evaluating their individual contribution to DIC and ALK changes ***taking into account the local buffering capacity of an ionic solution when TA and DIC are changing due to internal processes, but ignoring advection and mixing (Zeebe and Wolf-Gladrow 2001). In the present study, we quantify the effect of the NEM on the $CO_2$ balance, which is almost exclusively controlled by aerobic degradation rates because the contributions of denitrification and NPP to the net ecosystem balance are small. Nitrification, a process triggered by the transport and/or production of $NH_4$ in oxygenated waters, favors outgassing through its effect on pH, which shifts the acid-base equilibrium of carbonate species and increases the $CO_2$ concentration. The contribution of supersaturated riverine waters to the overall estuarine $CO_2$ dynamics is calculated as difference between all the other processes creating or consuming $CO_2$.***"

Fig. 8a. Were seasonal partitioning combined to overall partitioning?

Indeed, the partitioning presented in figure 8a (and figure 8b) are calculated on the basis of the 4 seasonal fluxes for each estuary. The manuscript was edited to clarify this point:

"Fig. 8a presents the contribution of the annually integrated NEM, nitrification and evasion of supersaturated, DIC enriched riverine waters to the total outgassing for each system, as well as for individual regions of the domain. ***The calculation of these annual values is based on the sum of the seasonal fluxes.***"

L489 Give more details about the different partitioning in the different zones here

An entire paragraph following L489 fully describes the partitioning of the 3 drivers of $FCO_2$ (NEM, nitrification and riverine $CO_2$) in the 3 different zones (i.e. NAR, MAR, SAR). This paragraph used to begin on L496 of the previous version of the manuscript. We now explain and discuss the regional breakdown earlier in the text and moved the description of the influence of nitrification and NEM on $CO_2$ outgassing to L 485. Now the discussion of the contributions of NEM, nitrification and riverine $CO_2$ to FCO2 in each of the 3 sub regions directly follows the sentence in L.489, pointed out by the reviewer.

"Model results reveal that, regionally, the NEM supports about 50% of the estuarine $CO_2$ outgassing, while nitrification and riverine DIC inputs sustain about 17% and 33% of the $CO_2$ emissions, respectively. ~~*Nitrification, a process triggered by the transport and/or production of $NH_4$ in oxygenated waters, favors outgassing through its effect on pH, which shifts the acid-base equilibrium of carbonate species and increases the $CO_2$ concentration. In addition, the NEM is almost exclusively controlled by aerobic degradation rates because the contribution of denitrification and NPP to the net ecosystem balance is small.*~~ The relative significance of the three processes described above shows important spatial variability…"

L508 Where is Table S1?

Table S1 was uploaded as a supplementary table and a link to download it was included on the page from which the manuscript could be downloaded (below the PDF symbol). Attached to this reply, we provide an archive containing the updated manuscript as well as all the supplementary information

L577 budgets

Done

L630 The normalization of NEM by a Q10 value appears reasonable. The normalization of FCO2 by a Q10 value must be justified. I'm not convinced of the latter normalization.

The rationale for the normalization of $FCO_2$ by a Q10 value, using the same approach as the one used for NEM, is the fact that, in many systems, NEM and $FCO_2$ are intimately linked. For instance, Mayer and Eyre (2012) proposed a linear relationship between NEM and FCO2. Applying the same normalization to both NEM and $FCO_2$ thus allows testing if a similar relationship can be observed along the entire climatic gradient of the US East Coast.

"In this section, we explore the relationships between such simple physical parameters and indicators of the estuarine carbon processing $\overline{NEM}$, $\overline{FCO_2}$ and *CFilt*. In order to account for the effect of temperature on C dynamics, $-\overline{NEM}$ and $\overline{FCO_2}$ are also normalized to the same temperature (arbitrarily chosen to be 0 degree). These normalized values are obtained by dividing $-\overline{NEM}$ and $\overline{FCO_2}$ by a $Q_{10}$ function *f(T)* (see Volta et al., 2014). ***This procedure allows accounting for the exponential increase in the rate of several temperature dependent processes contributing to the NEM (i.e. photosynthesis, organic carbon degradation…). Applying the same normalization to -$\overline{NEM}$ and $\overline{FCO_2}$ is a way of testing how intimately linked NEM and $FCO_2$ are in estuarine systems. Indeed linear relationships relating one to the other have been reported (Mayer and Eyre, 2012).***"

L660 ff Here it becomes obvious that f(t) cannot be applied to FCO2.

Indeed, we agree with the reviewer that no clear relationship between Q10-normalized $FCO_2$ and S/Q can be observed over the entire spectrum of values of S/Q that can be found along the east coast of the US. In fact, our results clearly illustrate that a linear regression between $FCO_2$ / f(T) and S/Q only provides a good fit using estuaries located in the MAR and SAR regions. The small estuaries from the NAR region, characterized by values of S/Q < 3 d m$^{-1}$ display a significantly different behaviour. We think that it is important to point out that small estuaries show a different biogeochemical response and establishing a range of values of S/Q within which Mayer and Eyre's relationship can be reproduced justifies the use of this normalization of $FCO_2$ by a Q10 . We modified the text to clarify our approach.

"Thus, the well-documented correlation between $\overline{NEM}$ and $\overline{FCO_2}$ (Maher and Eyre, 2012) does not seem to hold for systems with very short residence times. For systems with S/Q > 3 days m$^{-1}$, we obtain a regression $FCO_2$ = -0.64 x *NEM* + 5.96 with a r$^2$ of 0.46, which compares well with the relation $FCO_2$ = -0.42 x *NEM* + 12 proposed by Maher and Eyre (2012) who used 24 seasonal estimates from small Australian estuaries. ***However, our results suggest that this relationship cannot be extrapolated to small systems such as those located in the NAR.***"

"As a consequence of the distinct behavior of short residence time systems, the coefficient of determination of the best-fitted power law function relating $\overline{FCO_2}$ and S/Q is only significant if NAR systems are excluded (y = 31.64 $x^{-0.58}$ with a $r^2$ = 0.70). *__This thus suggest that such relationships (as well as that proposed by Maher and Eyre, 2012) cannot be applied to any system but only those for which S/Q>3 day $m^{-1}$.__*"

L668 whom -> who

Done

L677 In this case the assumption of pCO2 in equilibrium with the atmosphere at the lower boundary contradicts the case "still oversaturated waters .."

In our simulations, the seaward boundary is located 20km away from the estuarine mouth and estuarine waters close to the mouths can thus be still oversaturated.

At the beginning of section 2.4.4, we state that '*For each estuary, the downstream boundary is located 20 km beyond the mouth to minimize the bias introduced by the choice of a fixed concentration boundary condition to characterize the ocean water masses (e.g. Regnier et al., 1998).*'

This assumption is also discussed in a paragraph of the new section 3.4:

"*__C-GEM places the lower boundary condition 20 km from the estuarine mouth into the coastal ocean and the influence of this boundary condition on simulated biogeochemical dynamics is thus limited. At the lower boundary condition, direct observations for nutrients and oxygen are extracted from databases such as the World Ocean Atlas (Antonov et al., 2014). However, lower boundary conditions for OC and $pCO_2$ (zero concentration for OC and assumption of $pCO_2$ equilibrium at the sea side) are simplified. This approach does not allow addressing the additional complexity introduced by biogeochemical dynamics in the estuarine plume (see Arndt et al., 2011). Yet, these dynamics only play a secondary role in the presented study that focuses on the role of the estuarine transition zone in processing terrestrial-derived carbon.__*"

L682 No link to Fig. 10d ?

We added a reference to Fig. 10d (now Fig.12d), as well as a brief discussion of the non-normalized results to the text.

"*__Figure 12d, which reports non-normalized $FCO_2$ reveals a monotonous increase of $FCO_2$ with S/Q. This suggests that, unlike the NEM for which the normalization by a temperature function allowed explaining most of the variability; $FCO_2$ is mostly controlled by the water residence time within the system.__* Discharge is the main $FCO_2$ driver in riverine dominated systems, while interactions with marshes are driving the outgassing in marine dominated systems surrounded by marshes."

L739 You really mean "prediction"? Not "projection"?

We agree that term projection is better suited and the text was updated accordingly.

"In regions with better data coverage, such as the one investigated here, our study highlights that the regional-scale quantification, attribution, and ***projection*** of estuarine biogeochemical cycling are now at reach."

L740 As your model is rather based on empirical relations than on first principles, I expect that changed systems due to climate shifts and consequences can change your basic relationship. Please include this aspect in a more careful outlook.

We agree with the remark of the reviewer stating that the domain of applicability of the relationship we found between NEM, temperature and the depth normalized estuarine residence is bound within the range of values observed within our study area. Some of these aspects are tackled in the new section 'Scope of applicability and model limitations'.

Additionally, following the reviewer's recommendation, a sentence was added in the outlook section to account for the limitations of the applicability of the relationships we designed. We would like however, to draw the attention of the reviewer on the mechanistic nature of our model. Thus, while the relationships presented in section 3.5 are indeed empirical, they stem from results produced by a model that is actually largely based on first principles.

"In the future, such simple relationships, relying on readily available geometric and hydraulic parameters could be used to quantify carbon processing in areas of the world devoid of direct measurements. ***However, it is important to note that such simple relationships are only valid over the range of boundary conditions and forcings explored and may not be applicable to conditions that fall outside of this range.*** In regions with better data coverage, such as the one investigated here, our study highlights that the regional-scale quantification, attribution, and ***projection*** of estuarine biogeochemical cycling are now at reach."

L1021 7(4), 1271-1295

The reference was updated:

"Volta, C., Arndt, S., Savenije, H. H. G., Laruelle, G. G., and Regnier, P.: C-GEM (v 1.0): a new, cost-efficient biogeochemical model for estuaries and its application to a funnel-shaped system, Geosci. Model Dev., ***7, 1271-1295, doi:10.5194/gmd-7-1271-2014***, 2014."

L1045 give units and if possible your own values.

Following the reviewer's advice, table 3 (now table 2) has been updated to include the unit of the values in the caption and the values calculated by our simulations for the selected estuaries.

**"Table 2**: Published local annually averaged estimates of $\overline{FCO_2}$ *in mol C m$^{-2}$ yr$^{-1}$* for estuaries along the East coast of the US.**"**

| Name | Lon | Lat | $\overline{FCO_2}$ Observed. | Modeled | Reference |
|---|---|---|---|---|---|
| Altamaha Sound | -81.3 | 31.3 | 32.4 | 72.7 | Jiang et al. (2008) |
| Bellamy | -70.9 | 43.2 | 3.6 | 3.9 | Hunt et al. (2010) |
| Cocheco | -70.9 | 43.2 | 3.1 | 3.9 | Hunt et al. (2010) |
| Doboy Sound | -81.3 | 31.4 | 13.9 | 25.7 | Jiang et al. (2008) |
| Great Bay | -70.9 | 43.1 | 3.6 | 3.9 | Hunt et al. (2011) |
| Little Bay | -70.9 | 43.1 | 2.4 | 3.9 | Hunt et al. (2011) |
| Oyster Bay | -70.9 | 43.1 | 4 | 3.9 | Hunt et al. (2011) |
| Parker River estuary | -70.8 | 42.8 | 1.1 | 3.9 | Raymond and Hopkinson (2003) |
| Sapelo Sound | -81.3 | 31.6 | 13.5 | 20.6 | Jiang et al. (2008) |
| Satilla River | -81.5 | 31 | 42.5 | 25.7 | Cai and Wang (1998) |
| York River | -76.4 | 37.2 | 6.2 | 8.1 | Raymond et al. (2000) |
| Hudson River | -74 | 40.6 | 13.5 | 15.5 | Raymond et al. (1997) |
| Florida Bay | -80.68 | 24.96 | 1.4 | n.a. | Dufore (2012) |

In addition, the text of the section 2.6 (Model-data comparison) has also been updated to compare observed and simulated $FCO_2$ in these 13 systems.

"***While such local validations allow assessing the performance of the model for a specific set of conditions, the purpose of this study is to capture the average biogeochemical behaviour of the estuaries of the eastern coast of the US. Therefore, in addition to the system-specific validation, published annually averaged $FCO_2$ estimates for 13 tidal systems located within the study area collected over the 1994-2006 period are compared to simulated $FCO_2$ for conditions representative of the year 2000. Overall, simulated $FCO_2$ are comparable to values reported in the literature (Tab. 2). Although discrepancies, which sometimes can significant, are observed at the level of individual systems, the model captures remarkably well the overall trend in $CO_2$ evasion rate across estuaries. The model simulates low $CO_2$ efflux (< 5 mol C m$^{-2}$ yr$^{-1}$) for the 7 systems were such conditions have been observed, while the 6 systems for which the $CO_2$ evasion exceeds 10 mol C m$^{-2}$ yr$^{-1}$ are the same in the observations and in the model runs. The discrepancy at the individual system level likely result from a combination of factors, including the choice of model processes and there parametrization, the uncertainties in constraining boundary conditions and the limited representability of instantaneous and local observed.***"

L1052 definition of winter (DJF)?

We define winter in section 2.4 as January, February and March. The definitions of the seasons are now reiterated in the table caption of table 5 to avoid any confusion:

"Table 5: Seasonal contribution to $FCO_2$ and NEM in each the sub-region. The seasons displaying the highest percentages are indicated in bold. ***Winter is defined as January, February and March, Spring as April, May and June and so on...***"

L1103 The caption must be understandable alone. & L1105 Separate: "black lines .. using all points" "grey lines are best fit only for .."

The caption was rewritten taking into account the suggestion of the reviewer:

[revised manuscript text omitted]
 | lat | $S$ | $Q$ | $Rt$ | $\overline{FCO_2}$ | $\overline{NEM}$ | $FCO_2$ | $NEM$ |
| degrees | degrees | $km^2$ | $m^3s^{-1}$ | days | $mol\ C\ m^{-2}\ yr^{-1}$ | $mol\ C\ m^{-2}\ yr^{-1}$ | $10^6\ mol\ C\ yr^{-1}$ | $10^6\ mol\ C\ yr^{-1}$ |
|---|---|---|---|---|---|---|---|---|
| **NAR** | | | | | | | | |
| -67.25 | 44.75 | 7 | 38.5 | 15 | 3.7 | -37.4 | 27 | -270 |
| -67.25 | 45.25 | 12 | 73.6 | 15 | 6.0 | -56.7 | 71 | -666 |
| -67.25 | 45.25 | 12 | 73.6 | 15 | 13.8 | -56.6 | 162 | -666 |
| -67.75 | 44.75 | 3 | 68.5 | 4 | 6.7 | -63.5 | 23 | -221 |
| -68.25 | 44.75 | 14 | 69.5 | 19 | 4.1 | -56.2 | 58 | -791 |
| -68.75 | 44.75 | 89 | 309.9 | 23 | 27.4 | -58.2 | 2431 | -5163 |
| -69.75 | 44.25 | 50 | 626.6 | 5 | 32.3 | -74.4 | 1607 | -3703 |
| -70.25 | 43.75 | 3 | 25.8 | 10 | 2.1 | -21.0 | 7 | -71 |
| -70.75 | 41.75 | 288 | 103.6 | 958 | 5.0 | -4.0 | 1428 | -1146 |
| -70.75 | 42.25 | 63 | 210.7 | 40 | 16.2 | -32.9 | 1025 | -2081 |
| -70.75 | 42.75 | 17 | 105.8 | 3 | 56.3 | -69.0 | 943 | -1155 |
| **MAR** | | | | | | | | |
| -70.75 | 43.25 | 31 | 29.9 | 11 | 21.6 | -37.4 | 662 | -1146 |
| -71.25 | 41.75 | 257 | 28.2 | 808 | 3.9 | -2.5 | 997 | -650 |
| -71.75 | 41.25 | 21 | 112.4 | 4 | 35.2 | -32.6 | 726 | -672 |
| -72.75 | 40.75 | 20 | 25.4 | 62 | 30.7 | -21.1 | 623 | -430 |
| -72.75 | 41.25 | 10 | 142.5 | 2 | 150.8 | -36.9 | 1578 | -386 |
| -72.75 | 41.75 | 55 | 476.6 | 3 | 55.9 | -45.7 | 3088 | -2523 |
| -73.25 | 40.75 | 19 | 26.8 | 56 | 31.4 | -28.4 | 608 | -550 |
| -74.25 | 40.75 | 1192 | 608.2 | 126 | 15.5 | -11.8 | 18432 | -14047 |
| -75.25 | 38.25 | 399 | 80.5 | 172 | 13.9 | -5.0 | 5558 | -2016 |
| -75.25 | 38.75 | 354 | 31.8 | 357 | 7.5 | -3.0 | 2659 | -1076 |
| -75.25 | 39.75 | 1716 | 499.0 | 221 | 10.0 | -7.8 | 17072 | -13439 |
| -75.75 | 39.25 | 224 | 18.3 | 434 | 7.5 | -2.9 | 1685 | -640 |
| -76.25 | 39.25 | 3427 | 717.1 | 352 | 8.1 | -5.1 | 27646 | -17352 |
| -76.75 | 37.25 | 586 | 272.3 | 74 | 15.0 | -10.4 | 8810 | -6084 |
| -76.75 | 37.75 | 154 | 36.3 | 163 | 10.7 | -6.6 | 1654 | -1023 |
| -76.75 | 39.25 | 59 | 71.2 | 29 | 48.6 | -34.6 | 2862 | -2038 |
| -77.25 | 38.25 | 206 | 30.2 | 268 | 6.1 | -3.3 | 1265 | -676 |
| -77.25 | 38.75 | 568 | 259.2 | 118 | 16.7 | -10.8 | 9488 | -6134 |
| **SAR** | | | | | | | | |
| -78.25 | 34.25 | 48 | 167.4 | 7 | 122.5 | -62.4 | 5916 | -3015 |
| -79.25 | 33.25 | 47 | 56.3 | 42 | 43.4 | -36.5 | 2056 | -1728 |
| -79.25 | 33.75 | 45 | 291.4 | 8 | 85.1 | -78.7 | 3843 | -3551 |
| -79.75 | 33.25 | 25 | 33.8 | 15 | 37.9 | -32.8 | 956 | -828 |
| -80.25 | 32.75 | 25 | 31.0 | 50 | 48.8 | -42.5 | 1214 | -1057 |
| -80.25 | 33.25 | 92 | 75.5 | 61 | 62.7 | -61.2 | 5769 | -5625 |
| -80.75 | 32.25 | 71 | 21.1 | 182 | 12.9 | -7.0 | 918 | -501 |
| -80.75 | 32.75 | 164 | 63.1 | 95 | 20.6 | -11.5 | 3372 | -1879 |
| -81.25 | 31.75 | 92 | 71.7 | 45 | 25.7 | -20.9 | 2361 | -1926 |
| -81.25 | 32.25 | 130 | 379.8 | 11 | 51.7 | -39.2 | 6732 | -5097 |
| -81.75 | 30.75 | 34 | 18.7 | 61 | 17.5 | -14.7 | 602 | -505 |
| -81.75 | 31.25 | 130 | 17.7 | 294 | 5.5 | -4.0 | 713 | -523 |
| -81.75 | 31.75 | 56 | 350.5 | 4 | 72.7 | -67.4 | 4068 | -3770 |

**Name** ⋯

[revised manuscript text omitted]

---

## Referee Report (RR1)

Overall statements

The revised manuscript "Air-water CO2 evasion from U.S. East Coast estuaries" by Goossens, N., Gildas, L.G., Arndt, S., Wei-Jun, C., Regnier, P. has improved substantially. The authors added several sections and additional tables which improved the text. Unfortunately there are still a lot of flaws in the manuscript. After correcting these minor bugs I suggest to accept the manuscript for publishing in Biogeosciences.

Detailed statements
There is still some confusion on the number of estuaries (43) handled by the model. Table S2, for example, includes 40 estuaries, Table 4 has 42. I know, the problem is that some rivers enter the same box. But anyway, this discrepancy in the presentation must be corrected within the entire manuscript.

In the following the line numbers refer to bg-2016-278-manuscript-version4.pdf.

L558 ff and your letter to the reviewer:
You discuss the outgassing of small and large estuaries saying that the surface area act as a limiting factor for gas exchange. Then I would expect smaller gas exchange per surface unit for smaller estuaries.

L169 NAR Fig. 1; Tab. 4 includes 11 estuaries .. 558 km2

L174 MAR (Tab. 4) 18 entries .. 9298 km2

L182 SAR (Tab. 4) 13 entries .. 959 km2

L193 Friedrichs and Hofmann2001 : Reference missing

L254 Fischer 1976: Reference missing

L295 The primary production module does not include nutrients, grazing pressure, turbidity. The only phytoplankton group is diatoms. Please comment on these shortenings.

L312 World Ocean Atlas, 2009: Reference missing

L467 UNH/GRDC Database: give a reference

L468 Fekete et al 2000: Reference missing

L473 is it July 2003 or July 2013? Compare L487

L481 I think you exchanged the display of pH upstream and downstream in the tables, as they do not fit to Fig. 7

L490 Compare pCO2 along the salinity gradient, or change text here.

L500 "Although .." Improve grammar.

L501 is it really a trend?

L502 I see only 6 systems

L503 I see only 5 systems

L507 "local observations"

L544 "B0"

L468 Savenije 2000: Reference missing

L564 "2013b"

L567 Zeebe and Wolf-Gladrow 2001: Reference missing

L717 "discuss"

L721  "watersheds"

L723 "contained by" -> "covered by"

L730 (b, B0, H)

L772 omit "is"

L789 Antonov et al., 2014: Reference missing

L802 "Chlorophyll-a"

L802 omit "and" -> "For DIC and alkalinity boundary conditions .."

L804 "are extracted"

L812-815 Improve grammar.

L841 Mayer and Eyre, 2012: Reference missing

L897 "suggests"

L967 Abril 2002 does not fit to L39

L998 Billen et al: Not called

L1062 Hartmann 2012 does not fit to L165

L1176 Paerl et al: Not called

L1262 Thieu et al: Not called

L1282 Van der Burgh: Not called

L1306 Tab. 2 omit Florida Bay as this was not modeled.

L1353 In the text you discuss Fig. 7b as Delaware (L486).

L1378 –FCO2 should be also at the y-axis

---

## Author Response (AR2)

Dear Editor,
Please find below a point by point answer to the comments provided by reviewer #2. In the
following text, the comments of the reviewer are in written back and our answers are written in
blue. All the modifications suggested by the reviewer have been carefully implemented into the
manuscript. This updated version of the manuscript is available with the 'track changes' option at
the end of this file, after our answers. In addition, following the reviewer's first request, the
supplementary tables have also been modified and re-uploaded.
On behalf of all co-authors,
Goulven Laruelle

Overall statements
The revised manuscript "Air-water CO2 evasion from U.S. East Coast estuaries" by Goossens, N.,
Gildas, L.G., Arndt, S., Wei-Jun, C., Regnier, P. has improved substantially. The authors added several
sections and additional tables which improved the text. Unfortunately there are still a lot of flaws in
the manuscript. After correcting these minor bugs I suggest to accept the manuscript for publishing
in Biogeosciences.
Detailed statements
There is still some confusion on the number of estuaries (43) handled by the model. Table S2, for
example, includes 40 estuaries, Table 4 has 42. I know, the problem is that some rivers enter the
same box. But anyway, this discrepancy in the presentation must be corrected within the entire
manuscript.
As the reviewer points out, there still were inconsistencies within the text with respect to the total
number of tidal estuaries simulated by the model. Table 2 is correct and the total number of systems
is 42. The entire manuscript has been updated accordingly and the supplementary tables (S1-S6)
have been updated to match the systems reported in table 2.
In the following the line numbers refer to bg-2016-278-manuscript-version4.pdf.
L558 ff and your letter to the reviewer:
You discuss the outgassing of small and large estuaries saying that the surface area act as a limiting
factor for gas exchange. Then I would expect smaller gas exchange per surface unit for smaller
estuaries.
For very small estuarine systems (such as those referred to in the discussion about the NAR region),
the overall surface area in the system available for the exchange between air and water is not
sufficient to transfer all the oversaturated $CO_2$ from the estuarine water to the atmosphere. In that
sense, surface area is a limiting factor to $CO_2$ outgasing because this physical limitation of the
available surface for gas exchange hampers the overall amount of $CO_2$ emitted to the atmosphere
during its transit through the estuarine filter. The emission rates per surface area however, are high
compared to other larger estuaries because the magnitude of the $pCO_2$ gradient at the air-water
interface is larger in small systems which are largely oversaturated on their entire length compared
to larger system where most of the outgasing only occurs in the upstream section of the estuary and
where the average emission rate is distributed over a much larger surface area.
L169 NAR Fig. 1; Tab. 4 includes 11 estuaries .. 558 km2
L174 MAR (Tab. 4) 18 entries .. 9298 km2
L182 SAR (Tab. 4) 13 entries .. 959 km2

The numbers of estuaries per region as well as their associated surface areas have been updated to match the value reported in table 4. In addition, the cumulated surface areas per regions are now also included in table 5.

L193 Friedrichs and Hofmann2001 : Reference missing

Both this reference and the one mentioned below were in the reference list but the line break between them was missing, making the reference look like Fischer, 2001. This has been corrected in the revised manuscript.

L254 Fischer 1976: Reference missing

Both this reference and the one mentioned above were in the reference list but the line break between them was missing, making the reference look like Fischer, 2001. This has been corrected in the revised manuscript.

L295 The primary production module does not include nutrients, grazing pressure, turbidity. The only phytoplankton group is diatoms. Please comment on these shortenings.

The primary production module used in the model is described *in extenso* in Volta et al. (2014, 2016)

and does take into account nutrients limitations, grazing pressure and the effect of SPM

concentrations on water turbidity and, thus the extinction coefficient used to calculate the available light in the water column. The brief description of the primary production module in the present manuscript was only intended to explicit the numerical method used to integrate the light profile in the water column. This short section has now been expended to clarify what processes are included in the calculation of primary production in our model.

"The primary production dynamics, which ***takes into account the combined effects of nutrients***

***limitation and light attenuation in the water column induced by its background turbidity and SPM***

***concentration,*** requires vertical resolution of the photic depth***. The latter*** is calculated according to the method described in Vanderborght et al. (2007). This method assumes an exponential decrease of the light in the water column (Platt et al., 1980), which is solved using a Gamma function."

It is true however that our model only accounts for one phytoplankton group and that limitation as well as potential future improvements are discussed in the 'Biogeochemical Model' paragraph of section '3.4 Scope of applicability and model limitations'.

"Although the reaction network of C-GEM accounts for all processes that control estuarine $FCO_2$

(Borges and Abril, 2012; Cai, 2011), several, potentially important processes, such as benthic-pelagic exchange processes, phosphorous sorption/desorption and mineral precipitation, a more complex representation of the local phytoplankton community, grazing by higher trophic levels, or multiple reactive organic carbon pools are not included. Although these processes are difficult to constrain and their importance for $FCO_2$ is uncertain, the lack of their explicit representations induces uncertainties in Cfilt."

L312 World Ocean Atlas, 2009: Reference missing

The data from the world ocean Atlas come from Antonov et al., 2010 and Locarini et al., 2010 for the temperature and salinity, respectively. This has been clarified in the manuscript:

"Transient physical forcings are calculated for each season and grid cell using monthly mean values of water temperature ***(World Ocean Atlas: Antonov et al. 2010; Locarini et al., 2010)***…"

L467 UNH/GRDC Database: give a reference

The following reference was added for the UNH/GRDC Database:

***"GRDC: Global Freshwater Fluxes into the World Oceans / Online provided by Global Runoff Data***

***Centre. 2014 ed. Koblenz: Federal Institute of Hydrology (BfG), 2014."***

L468 Fekete et al 2000: Reference missing

The Fekete et al 2000 reference was removed and replaced by GRDC (2014) to refer to the
UNH/GRDC Database (see comment above).
L473 is it July 2003 or July 2013? Compare L487
The validation performed for the Delaware Bay relies on a pH longitudinal profile and boundary
conditions all sampled in 2003 (Sharp et al., 2009; Sharp, 2010) as stated in the manuscript.
However, for comparison's sake, a reference is also made to the recent work of Joesoef et al. (2015)
when discussing the $pCO_2$ profile because this study reports $pCO_2$ measurements performed in the
same estuary during the same month (July) but another year (2013). The fact that Joesoef et al.
(2015)'s data were not sampled the same year as the one simulated by the model was already
mentioned in the text, but the following sentences have been modified to make this fact clearer and
avoid potential confusion:
"***Overall, the longitudinal pCO_2 profile of the Delaware estuary is characterized by*** values close to
equilibrium with the atmosphere in the widest section of the Delaware Bay (close to the estuarine
mouth ***and throughout the 40 first kilometers of the system***) and values above 1200 µatm at
***kilometer 150 and beyond, where characteristic*** salinities ***are*** below 5. ***Although the profile***
***presented here is simulated using boundary conditions representative of July 2003 and no pCO_2***
***data were available for validation for this period, a recent study by Joesoef et al. (2015) reports a***
***similar longitudinal pCO_2 profile in July 2013.***"
L481 I think you exchanged the display of pH upstream and downstream in the tables, as they do not
fit to Fig. 7
There was indeed in mistake in table S6 displaying the pH values used at the downstream boundary
condition, which are calculated from DIC and Alkalinity assuming $CO_2$ equilibrium between coastal
waters and the atmosphere (see section 2.4.4 of the manuscript). Table S6 was updated with the
proper values. Table S5, which provides the upstream boundary conditions, was correct however.
L490 Compare pCO2 along the salinity gradient, or change text here.
Panel c of figure 7 reports simulated $pCO_2$ values for the Delaware estuary against the distance from
the mouth of the system. When discussing this profile, we now refer the distance from the estuarine
mouth rather than to salinity to identify the different sections of the estuary:
"***Overall, the longitudinal pCO_2 profile of the Delaware estuary is characterized by*** values close to
equilibrium with the atmosphere in the widest section of the Delaware Bay (***near*** the estuarine
mouth ***and throughout the 40 first kilometers of the system***) and values above 1200 µatm at
***kilometer 150 and beyond, where characteristic*** salinities ***are*** below 5."
L500 "Although .." Improve grammar.
The sentence was rewritten as:
"Although ***significant*** discrepancies ***are observed*** at the level of individual systems, the model
captures remarkably well the overall ***behaviors of estuaries along the East coast of the US in term***
***of intensity of*** $CO_2$ evasion rate."
L501 is it really a trend?
The use of the word tend was not really adequate in this sentence and it was removed in the
updated manuscript (see comment above).
L502 I see only 6 systems
If Florida Bay is removed from table 2, as suggested by the reviewer in his third to last comment,
there indeed will be only 6 systems with observed emission rates < 5 mol C m$^{-2}$ yr$^{-1}$.
L503 I see only 5 systems

Indeed, the reviewer is correct; there are only 5 systems with observed emission rates > 10 mol C m$^{-2}$ yr$^{-1}$. The sentence was corrected in the manuscript.

"The model simulates low $CO_2$ efflux (< 5 mol C m$^{-2}$ yr$^{-1}$) for the **6** systems were such conditions have been observed, while the **5** systems for which the $CO_2$ evasion exceeds 10 mol C m$^{-2}$ yr$^{-1}$ are the same in the observations and in the model runs."

L507 "local observations"
Corrected

L544 "B0"
Corrected

L468 Savenije 2000: Reference missing
The reference was intended to be Savenije (2005), which is in the reference list. This mistake was corrected in the updated manuscript.
"A larger ratio of estuarine width **B0** and convergence length b corresponds to a more funnel shaped estuary while a low ratio corresponds to a more prismatic geometry (Savenije, **2005**; Volta et al., 2014)."

L564 "2013b"
Corrected

L567 Zeebe and Wolf-Gladrow 2001: Reference missing
The reference has been added to the reference list:
*"Zeebe, R. E. and Wolf-Gladrow, D. (Eds.): CO2 in seawater: equilibrium, kinetics, isotopes, Elsevier, Amsterdam, 2001."*

L717 "discuss"
Corrected

L721 "watersheds"
Corrected

L723 "contained by" -> "covered by"
Corrected

L730 (b, B0, H)
Corrected

L772 omit "is"
Corrected

L789 Antonov et al., 2014: Reference missing
Although the database has been updated in 2014, the recommended reference is Antonov et al., 2010, which is in our reference list. The text was updated accordingly:
"At the lower boundary condition, direct observations for nutrients and oxygen are extracted from databases such as the World Ocean Atlas (Antonov et al., **2010**)."

L802 "Chlorophyll-a"
Corrected

L802 omit "and" -> "For DIC and alkalinity boundary conditions .."
Corrected
L804 "are extracted"
Corrected
L812-815 Improve grammar.
The sentence was rewritten as:
"The generic nature of the applied model approach *renders a direct validation of model results on*
*the basis of local and instantaneous observational data (e.g. longitudinal profiles) difficult.* In
particular the application*s* of seasonally/annually averaged or model-deduced boundary conditions,
which are likely not representative of these long-term average conditions*, do not lend themselves*
*well to comparison with punctual measurements.*"
L841 Mayer and Eyre, 2012: Reference missing
The proper citation was Ma**h**er and Eyre, 2012. This typo was corrected in the manuscript and
Maher and Eyre (2012) is in the reference list.
L897 "suggests"
Corrected
L967 Abril 2002 does not fit to L39
Abril (2002) is not cited in L39, Borges and Abril (2012) is.
L998 Billen et al: Not called
The reference above was removed from the reference list.
L1062 Hartmann 2012 does not fit to L165
Hartmann (2012) is not cited in L165, Hartmann et al. (2009) is.
L1176 Paerl et al: Not called
L1262 Thieu et al: Not called
L1282 Van der Burgh: Not called
The three references above were removed from the reference list.
L1306 Tab. 2 omit Florida Bay as this was not modeled.
Florida Bay has been removed from table 2 and the corresponding reference (Dufore et al., 2012)
has been removed from the reference list.
L1353 In the text you discuss Fig. 7b as Delaware (L486).
This is indeed a mistake, the $pCO_2$ profile for the Delaware estuary is represented in panel c. The text
has been updated accordingly.
"Both processes lead to a well-developed $pCO_2$ increase in this area (Fig. *7c*)."
L1378 –FCO2 should be also at the y-axis
The caption is wrong and not the axis. Instead of referring to NEM and $-FCO_2$, the caption should
refer to –NEM and $FCO_2$. It has been updated accordingly in the manuscript.

[revised manuscript text omitted]